# Multi-domain Riemannian Graph Gluing for Building Graph Foundation Models

**Li Sun**[1]*, **Zhenhao Huang**[2], **Silei Chen**[2], **Lanxu Yang**[2], **Junda Ye**[1],
**Sen Su**[1], **Philip S. Yu**[3]
[1]Beijing University of Posts and Telecommunications, Beijing 100876, China
[2]North China Electric Power University, Beijing 102206, China
[3]University of Illinois Chicago, IL, USA

## Abstract

Multi-domain graph pre-training integrates knowledge from diverse domains to enhance performance in the target domains, which is crucial for building graph foundation models. Despite initial success, existing solutions often fall short of answering a fundamental question: *how is knowledge integrated or transferred across domains?* This theoretical limitation motivates us to rethink the consistency and transferability between model pre-training and domain adaptation. In this paper, we propose a fresh Riemannian geometry perspective, whose core idea is to merge any graph dataset into a unified, smooth **Riemannian manifold**, enabling a systematic understanding of knowledge integration and transfer. To achieve this, our key contribution is the theoretical establishment of neural manifold gluing, which first characterizes local geometry using an adaptive orthogonal frame and then "glues" the local pieces together into a coherent whole. Building on this theory, we present the **GraphGlue** framework, which supports batched pre-training with EMA prototyping and provides a transferability measure based on geometric consistence. Extensive experiments demonstrate its superior performance across diverse graph domains. Moreover, we empirically validated **GraphGlue**'s geometric scaling law, showing that larger quantities of datasets improve model transferability by producing a smoother manifold. **Codes** are available at `https://github.com/RiemannGraph/GraphGlue`.

## 1 Introduction

Foundation models have revolutionized the representation learning in natural language processing Bommasani et al. (2021); Brown et al. (2020); Devlin et al. (2019) and computer vision Dosovitskiy et al. (2020) by integrating multi-domain knowledge during pre-training and transferring it to target domains. Graph-structured data are ubiquitous non-Euclidean structures in real-world applications, ranging from social network analysis Zhou et al. (2020); Sharma et al. (2024) to molecular design Guo et al. (2022); Wang et al. (2023). Hence, recent efforts have been made to replicate the success of the foundation model in the field of graphs, achieving multi-domain pre-training and cross-domain transfer learning for graphs.

Multi-domain graph pre-training is challenging given the significant semantic heterogeneity across different domains, such as social networks and biological molecules. In the literature, one line of work extracts multi-domain knowledge via Large Language Models (LLMs), leveraging the well-pretrained textual semantics but remaining limited to text-attributed graphs Zhu et al. (2025); Xia et al. (2024); Tang et al. (2024); Ren et al. (2024); Chen et al. (2024). However, many real graphs lack explicit textual attributes. Moreover, textual annotation is labor-intensive and may introduce hallucinations through LLM generation.

Rather than being tied to textual information, multi-domain pre-training for text-free graphs has garnered increasing attention recently. A series of methods seek to learn shared or invariant knowledge during pre-training using graph codebooks Wang et al. (2024); Sun et al. (2025b); Jiang et al. (2024);

---

*Corresponding Author: Li Sun, lsun@bupt.edu.cn

Bo et al. (2025), motifs Sun et al. (2025b), computation trees Wang et al. (2024; 2025c), etc. Meanwhile, advanced adaptation techniques are introduced to improve the downstream tasks, e.g., domain tokens Yu et al. (2025a); Jiao et al. (2025); Yuan et al. (2025); Wang et al. (2025a) and in-context learning Huang et al. (2023); Liu et al. (2024). While existing solutions have achieved encouraging results, a fundamental question remains inadequately addressed: **how is knowledge integrated or transferred across domains?** The theoretical underpinnings in this context remain underexplored. Though Wang et al. (2024); Zhang et al. (2024); Ruiz et al. (2020) give similarity measures across different domains, they do not frame model pre-training and domain adaptation within a consistent framework. This gap limits its ability to assess transfer difficulty, especially for the unseen graphs. Thus, we are motivated to rethink the consistency and transferability to target domains.

In this paper, we propose a fresh differential geometry perspective, whose core is the integration of any graph dataset into **a unified, smooth Riemannian manifold**, providing a rigorous foundation for systematically analyzing knowledge integration and transfer. To achieve this, we introduce a new theory – **Neural Manifold Gluing**, whose intuitive idea is to first characterize the local geometry, and then "glue" these local pieces together into a coherent whole. Specifically, we propose a sparse perturbation and an adaptive orthogonal frame to learn the local geometry. Gluing local pieces is achieved through metric compatibility along the edges (Theorem 4.5) and triangle triviality with respect to the concept of holonomy (Theorem 4.8). Finally, we smooth the manifold by controlling the change ratio of volume elements (Theorem 4.9), enhancing knowledge transport along the manifold.

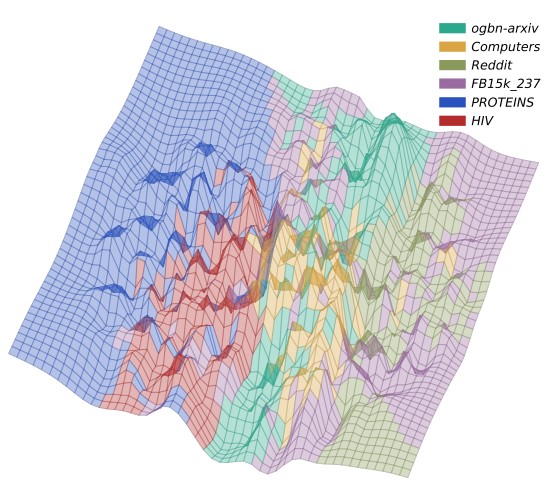

Figure 1: An illustration of manifold gluing. The domains are distinguished by colors.

Building on the theory established above, we design a pre-training-adaptation framework named **GRAPHGLUE**, which extends local geometry to the global scale. During pre-training, we incorporate an Exponential Moving Average (EMA) prototyping before gluing, which distinguishes domain semantics through different locations on the manifold and efficiently handles large-scale graphs in a batched manner. In the adaptation phase, GRAPHGLUE employs learnable prompts and a Riemannian Mixture-of-Experts, while gluing target domains to the pre-trained manifold, ensuring geometric consistency. A Geometric Transfer Metric (GTM) is naturally defined by metric compatibility to quantify transfer difficulty. Moreover, GRAPHGLUE exhibits a **geometric scaling law**: larger quantities of graph datasets produce a smoother manifold, thereby improving model transferability.

In summary, key contributions are listed as follows. **1. Problem.** We investigate the theoretical underpinnings of multi-domain graph pre-training, and study a foundational problem of how knowledge is integrated and transferred across different domains. **2. Theory.** We introduce a fresh differential geometry perspective for systematically understanding knowledge transfer, and propose the theory of neural manifold gluing, which consistently integrates multi-domain graphs into a unified, smooth Riemannian manifold via "gluing". **3. Methodology.** We propose a GRAPHGLUE framework based on the above theory, which supports batched pre-training for large-scale graphs and incorporates a natural metric to quantify its transferability. **4. Experiment.** We evaluate GRAPHGLUE in cross-domain transfer learning and empirically demonstrate its geometric scaling law.

## 2   RELATED WORK

**Graph Foundation Models**   Graph Foundation Models (GFMs) aim to provide pre-trainable, general-purpose deep learning architectures for graphs Wang et al. (2025b); Liu et al. (2025). Recently, the capabilities of Large Language Models (LLMs) have extended to text-attributed graphs

Zhu et al. (2025); Xia et al. (2024); Tang et al. (2024); Ren et al. (2024); Chen et al. (2024). Also, GFMs have been developed for various specialized domains, such as knowledge graphs Huang et al. (2025); Luo et al. (2025), recommender systems Wu et al. (2025), and molecular graphs Xia et al. (2023); Sypetkowski et al. (2024). Given the prevalence of text-free graphs, recent efforts have focused on building general-purpose models via multi-domain pre-training Zhao et al. (2025).

**Multi-domain Graph Pre-training**  In graph pre-training, Graph Neural Networks (GNNs) are trained by self-supervised learning—either generative Hou et al. (2022) or contrastive Veličković et al. (2019); Qiu et al. (2020). In light of the semantic heterogeneity across different domains, several methods have been proposed to learn shared or invariant knowledge Yuan et al. (2025); Chen et al. (2025); Wang et al. (2025a). Despite the encouraging results, the theoretical foundations of how knowledge is integrated and transferred remain underexplored.

**Graph Fine-tuning and Prompt Learning**  The alignment of pre-trained models with downstream tasks necessitates an adaptation phase, which is roughly categorized into two paradigms: 1) Graph fine-tuning adapts the model behavior using limited target-domain data Sun et al. (2024d), and recent advances introduce parameter-efficient fine-tuning methods such as low-rank adaptation Yang et al. (2025b). 2) Graph prompting keeps pre-trained parameters frozen and enhances performance by inserting learnable prompt vectors Yu et al. (2025a); Liu et al. (2023); Sun et al. (2022b); Fang et al. (2023). Yet, how to quantify the transfer effort to target domains remains an open issue.

**Riemannian Graph Representation Learning**  Most existing Riemannian models are tailored to specific tasks Chami et al. (2019); Grover et al. (2025); Bachmann et al. (2020); Gu et al. (2019). Recently, Sun et al. (2025b) design a new GNN backbone on the product manifold for GFM. In contrast, our focus lies on developing a framework for multi-domain pre-training, and on constructing a general manifold, rather specific ones. (Full related work is provided in Appendix E.)

## 3 Notations and Preliminaries

This part briefly reviews the key concepts of Riemannian manifold, frame and holonomy, and then states multi-domain pre-training where we reconsider its consistency and transferability from a fresh differential geometry perspective. Important notations are summarized in Appendix A.

**Riemannian Geometry**  Riemannian geometry provides an elegant framework for studying graphs and structures. A Riemannian manifold $(\mathcal{M}, \mathbf{G})$ with dimension $M$ is a smooth manifold $\mathcal{M}$ endowed with a Riemannian metric tensor $\mathbf{G}$. Each point $\boldsymbol{p} \in \mathcal{M}$ ties to a tangent space $\mathcal{T}_{\boldsymbol{p}}\mathcal{M}$, and its volume element is the determinant of the Riemannian metric tensor, denoted as $|\boldsymbol{G}(\boldsymbol{p})|$. The coordinate chart of tangent space is denoted as $(U, x^1, ..., x^M)$. Ricci curvature $\mathrm{Ric}(X, Y)$ governs the change ratio of volume elements along the geodesic. The concept of holonomy describes the changes of a tangent vector traversing a closed curve. Rigorous elaborations are in Appendix C.

**Cartan's Method of Moving Frame**  This renowned method Tron et al. (2024) offers a principled way to study manifold geometry with a frame. Though Élie Cartan laid the mathematical principle, its deep learning methodology remains largely unexplored. Our work seeks to bridge this gap.

**Multi-domain Graph Pre-training**  In this context, a deep learning architecture is first pre-trained on different source domains and then adapted to a target domain. A graph is described as $\mathcal{G} = (\mathcal{V}, \mathcal{E})$ with a feature matrix $\boldsymbol{X} \in \mathbb{R}^{|\mathcal{V}| \times F}$, where $\mathcal{V}$ and $\mathcal{E}$ denote the node set and edge set, respectively. We consider a collection of $K$ graphs $\mathbb{S} = \{\mathcal{S}^1, \mathcal{S}^2, \cdots, \mathcal{S}^K\}$ from $L$ domains $\mathbb{D} = \{\mathcal{D}^1, \mathcal{D}^2, \cdots, \mathcal{D}^L\}$. A model $f_{\boldsymbol{\Theta}}(\mathrm{GNN}(\cdot))$ is pre-trained on the graph dataset $\mathbb{G}$ with an encoder $\mathrm{GNN}(\cdot)$, after which the pre-trained parameters $\{\boldsymbol{\Theta}_f^{\star}, \boldsymbol{\Theta}_{\mathrm{GNN}}^{\star}\}$ are frozen. The encoder is implemented with popular graph neural networks such as GCN Kipf & Welling (2017). Given a graph $\mathcal{G}^t$ of the target domain $\mathcal{D}^t$, the pre-trained model can generate informative representations for $\mathcal{G}^t$ with slight adaptation. Note that the target domain can be seen $\mathcal{D}^t \in \mathbb{D}$ or unseen $\mathcal{D}^t \notin \mathbb{D}$ during pre-training. Unlike existing solutions, *our goal is to design a transferable graph model with a principled interpretation*.

## 4 THEORY: CONSTRUCTING A UNIFIED, SMOOTH MANIFOLD

Existing solutions often lack a principled framework to interpret how knowledge is integrated or transferred across domains. To fill this gap, we introduce a differential geometry perspective for multi-domain graph pre-training. The core of our approach is the construction of a **pre-trainable, unified, and smooth Riemannian manifold**, which provides a rigorous foundation for systematically analyzing knowledge integration and transfer. In the literature, Riemannian graph representation learning primarily studies the specific manifolds, e.g., hyperbolic spaces Chami et al. (2019); Yang et al. (2025a), spherical spaces Liu et al. (2022), and product manifolds Gu et al. (2019). However, constructing a general manifold underlying multi-domain graphs remains unexplored.

To achieve this, we establish a novel theory – **neural manifold gluing**, whose intuitive idea is to first characterize the local geometry, and then "glue" these local pieces together to form a unified, smooth Riemannian manifold. Derivations and proofs of our establishment are provided in Appendix B.

### 4.1 LEARNING LOCAL GEOMETRY WITH ADAPTIVE ORTHOGONAL FRAME

In a Riemannian manifold, the local geometry at a given point is characterized by its tangent space. Going beyond the classic Cartan's method Tron et al. (2024), we present a deep learning approach to infer the basis of the tangent space. Specifically, we introduce a $(k, M)$-sparse perturbation, mimicking the directional derivative $D_{\boldsymbol{v}} f = \lim_{t \to 0} \frac{f(\boldsymbol{p}+t\boldsymbol{v})-f(\boldsymbol{p})}{t}$, to generate a set of tangent vectors at the given point, after which an adaptive orthogonal frame is applied to form the basis of the tangent space. Note that the perturbation is attached with a parametric $f_{\text{GNN}}$ in our establishment.

**Definition 4.1** $((k, M)$-**sparse perturbation**). *Given a graph perturbation set that consists of $M$ nodes $\mathbb{P} = \{p_i\}$ with parameters $\{\boldsymbol{p}_i\}$, for $\mathcal{G} = (\mathcal{V}, \mathcal{E})$, the perturbed graph is denoted as $\hat{\mathcal{G}} = (\hat{\mathcal{V}}, \hat{\mathcal{E}}) := \mathcal{G} \oplus \mathbb{P} = \left( \mathcal{V} \cup \{p_m\}_{m=1}^M, \mathcal{E} \cup \{(v_{i_m}, p_m)\}_{i_m=1, m=1}^{k, M} \right)$, where $(v_i, p_m)$ is a edge weighted by an attentive function $h(\boldsymbol{x}_i, \boldsymbol{p}_m)$, $v_{i_m}$ are $k$ nodes selected based on top-k $h(\boldsymbol{x}_i, \boldsymbol{p}_m)$.*

**Definition 4.2** (**Adaptive Orthogonal Frame, AOF**). *With tangent vectors generated by the above perturbation and a graph encoder $f_{\text{GNN}}$, after QR-decomposition with length recovery, the adaptive orthogonal frame is $\{\boldsymbol{w}_m : \boldsymbol{z}^{(i)} \mapsto \boldsymbol{w}_m^{(i)} \in \mathbb{R}^d\}_{m=1}^M$ for every representation $\boldsymbol{z}^{(i)}$. There exists a dual frame $\{\boldsymbol{\theta}^m\}$ such that $\boldsymbol{\theta}^m(\boldsymbol{w}_l) = \delta_{ml}$, where $\delta_{ml}$ is the Kronecker delta.*

We show that the aforementioned length recovery of the basis is important, since the length of the tangent vector, describing the space deformation, is upper-bounded by the perturbation. In fact, the angles and lengths of the basis vectors reflect how the space is stretched and twisted, respectively.

**Theorem 4.3** (**Upper bound of Tangent Vector Length, Appendix B.1**). *Given a connected $\mathcal{G}$ with $N$ nodes, the adjacency matrix $\boldsymbol{A}$, the Laplacian $\boldsymbol{L}$, and the feature matrix of perturbation nodes $\boldsymbol{P}$, apply $(k, M)$-sparse perturbation to $\mathcal{G}$, suppose $\frac{kM}{N} = \varepsilon$, where $\varepsilon > 0$ is small, and added edge weights satisfy $\sum_l h(v_i, p_l) = 1$. Then, the upper bound $\|\boldsymbol{w}_m^{\boldsymbol{p}}\| \leq (1 + \varepsilon)\|\boldsymbol{P}\|$ holds, where $\boldsymbol{w}_m^{\boldsymbol{p}}$ is the component of $\boldsymbol{w}_m$ determined by perturbation.*

Thus, the local metric at each point is derived through the basis vectors of its tangent space. In particular, given the representation of $\mathcal{G}_i$ as $\boldsymbol{z}_i \in \mathbb{R}^d$, the coordinates $U_i$ in a neighborhood around $\boldsymbol{z}_i$, and the learned dual frame $\boldsymbol{\theta}^m$, the local metric tensor $\mathbf{G}_i$ on $U_i$ takes the form of $\mathbf{G}_i(\boldsymbol{w}_m^{(i)}, \boldsymbol{w}_l^{(i)}) = g_{ml}(\boldsymbol{z}_i)(\boldsymbol{\theta}^m \otimes \boldsymbol{\theta}^l)$, where $g_{ml}(\boldsymbol{z}_i) = \langle \boldsymbol{w}_m^{(i)}, \boldsymbol{w}_l^{(i)} \rangle$. Equivalently, the matrix form of $\mathbf{G}_i$ is written as

$$\boldsymbol{G}_i = \boldsymbol{W}^{(i)\top} \boldsymbol{W}^{(i)} = \text{diag}(\|\boldsymbol{w}_1\|^2, ..., \|\boldsymbol{w}_M\|^2), \tag{1}$$

with the basis of tangent space formulated as $\boldsymbol{W}^{(i)} = [\boldsymbol{w}_1^{(i)}, ..., \boldsymbol{w}_M^{(i)}] \in \mathbb{R}^{d \times M}$. The inner product w.r.t. $\boldsymbol{G}_i$ is given as $\boldsymbol{G}_i(\boldsymbol{u}, \boldsymbol{v}) := \boldsymbol{u}^\top \boldsymbol{G}_i \boldsymbol{v}$ for tangent vectors $\boldsymbol{u}, \boldsymbol{v} \in T_{\boldsymbol{z}^{(i)}} U_i$.

### 4.2 GLUING LOCAL PIECES TO FORM A SMOOTH MANIFOLD

Given a set of isolated Riemannian manifolds $\{\mathcal{M}^{(i)} = (\boldsymbol{z}^{(i)}, T_{\boldsymbol{z}^{(i)}} U_i, \boldsymbol{G}_i)\}_{i=1}^N$, we are devoted to gluing them together to construct a unified, smooth Riemannian manifold with a global metric. In a nutshell, These local pieces are connected through the edges and triangles with the concept of holonomy, after which the constructed manifold is smoothed by controlling the Ricci curvature.

**Gluing.** We begin with the *compatibility of metric* along edges, which is necessary for the existence of a global metric. According to Edelsbrunner & Harer; Chung, the gluing boundary can be defined by the adjacency in graph topology. To preserve compatibility, we perform a tangent translation along an edge $(i, j) \in \mathcal{E}$, referred to as edge tangent translation, to transform the local metrics. We show that it ensures metric compatibility along an edge, and is proven to induce a global metric. In addition, its computational complexity is reduced to $\mathcal{O}(M)$ with the QR-decomposition above.

**Definition 4.4** (**Edge Tangent Translation**). *Given an edge* $(i, j) \in \mathcal{E}$, *the tangent spaces of its two endpoints* $T_{\boldsymbol{z}^{(i)}} U_i$ *and* $T_{\boldsymbol{z}^{(j)}} U_j$, *and the Riemannian metric of* $T_{\boldsymbol{z}^{(i)}} U_i$ *denoted as* $\boldsymbol{G}_i$, *the edge tangent translation is defined as a linear map* $\boldsymbol{P}^{(i,j)} : T_{\boldsymbol{z}^{(i)}} U_i \to T_{\boldsymbol{z}^{(j)}} U_j$ *on edge* $(i, j) \in \mathcal{E}$ *as*

$$\boldsymbol{P}^{(i,j)} = \boldsymbol{G}_j^{-1/2} \left( \boldsymbol{G}_j^{1/2} \boldsymbol{G}_i \boldsymbol{G}_j^{1/2} \right)^{1/2} \boldsymbol{G}_j^{-1/2}. \tag{2}$$

**Theorem 4.5** (**Tangent Edge Translation as Isometry, Appendix B.2**). *The tangent edge translation in Definition 4.4 is the optimal solution of*

$$\min_{\boldsymbol{P} \in GL(M)} \left\| \boldsymbol{P}^\top \boldsymbol{G}_j \boldsymbol{P} - \boldsymbol{G}_i \right\|_F^2, \tag{3}$$

*where* GL *denotes the general linear group, such that* $\boldsymbol{G}_j(\boldsymbol{P}^{(i,j)}\boldsymbol{u}, \boldsymbol{P}^{(i,j)}\boldsymbol{v}) = \boldsymbol{G}_i(\boldsymbol{u}, \boldsymbol{v})$, *which induces an isometry* $\phi^{(i,j)}$ *between manifold boundaries* $\partial U_i$ *and* $\partial U_j$.

**Theorem 4.6** (**Existence of Global Metric, Appendix B.3**). *Let* $(\{\boldsymbol{G}_i\}_{i=1}^N, \{\boldsymbol{P}^{(i,j)}\}_{(i,j)\in\mathcal{E}})$ *be local metrics and tangent edge translations. There exists a unique global continuous metric* $\boldsymbol{G}$ *on* $\left(\bigcup_\phi\right)_{i=1}^N U_i$ *such that the restriction of* $\boldsymbol{G}|_{U_i} = \boldsymbol{G}_i$ *for all* $i$.

The edge tangent translations connect gluing boundaries in accordance to Theorem 4.5 and 4.6. However, when gluing along higher-order motifs, such as triangles and cycles, some offsets may occur when going round trips, so that gluing boundaries are not well aligned. In other words, although the glued manifold is connected, it is not yet continuous everywhere. To address this issue, we introduce the *concept of holonomy*, describing how the tangent vector changes when traversing along a closed curve, and define a holonomy map to measure the changes. We show that, when the holonomy map of triangles is trivial, the offset at the gluing boundaries is eliminated.

**Definition 4.7** (**Holonomy Map and Holonomy Loss**). *Let* $\mathcal{Z}_1(\mathcal{G})$ *denote the real vector space of 1-cycles on graph* $\mathcal{G} = (\mathcal{V}, \mathcal{E})$ *under symmetric difference. For any cycle* $\mathcal{C} = (i_0, i_1, \ldots, i_L = i_0)$, *its* ***holonomy map*** *is defined as the composition of transport maps along the path,*

$$\boldsymbol{H}(\mathcal{C}) := \prod_{\ell=0}^{L-1} \boldsymbol{P}^{(i_\ell, i_{\ell+1})} \in \mathrm{GL}(M). \tag{4}$$

*The collection* $\mathbf{P} := \{\boldsymbol{P}^{(i,j)}\}$ *is said to be* ***trivial*** *if* $\boldsymbol{H}(\mathcal{C})$ *is the identity map for* $\forall \mathcal{C} \in \mathcal{Z}_1(\mathcal{G})$. *Given the set of all triangles* $\mathcal{A}_{ijk} = ((v_i, v_j), (v_j, v_k))$, *the corresponding holonomy loss is formulated as*

$$\mathcal{L}_{holo}(\mathcal{G}) = \frac{1}{|\mathcal{A}|} \sum_{\mathcal{A}_{ijk}} \|\boldsymbol{P}^{(k,i)} \boldsymbol{P}^{(j,k)} \boldsymbol{P}^{(i,j)} - \boldsymbol{I}\|_F^2. \tag{5}$$

**Theorem 4.8** (**Triangle Triviality, Appendix B.4**). *If every edge belongs to at least one triangle, and* $\boldsymbol{H}(\mathcal{T}) = \boldsymbol{I}$ *for all triangular cycles* $\mathcal{T}$ *in* $\mathcal{G}$, *then* $\boldsymbol{H}(\mathcal{C}) = \boldsymbol{I}$ *for all cycles* $\mathcal{C} \in \mathcal{Z}_1(\mathcal{G})$.

**Smoothing.** So far, the glued manifold has achieved $C^1$ continuity, but $C^2$ continuity is required to yield a smooth global metric and to eliminate "fold" that hinders knowledge transport along the manifold. To bridge this gap, we visit the *concept of Ricci curvature*, a kind of $C^2$ continuity on the manifold, which governs the rate of changes of the volume element along the geodesic. Nevertheless, calculating Ricci curvature is rather expensive Petersen (2016); Ollivier (2007). Instead, we propose an alternative of volume change ratio between two endpoints, which is shown to sufficiently determine whether the geodesic is "convex" or "concave".

**Theorem 4.9** (**Ricci Curvature Estimation, Appendix B.5**). *Given a graph* $\mathcal{G} = (\mathcal{V}, \mathcal{E})$ *and an edge* $(i, j) \in \mathcal{E}$, *let* $\boldsymbol{z}^{(i)}, \boldsymbol{z}^{(j)} \in \mathcal{M}$ *be the corresponding embedded points, and* $\gamma : [0, 1] \to \mathcal{M}$ *be the unit-speed geodesic connecting them, i.e.,* $\gamma(0) = \boldsymbol{z}^{(i)}$, $\gamma(1) = \boldsymbol{z}^{(j)}$. *The sign of the Ricci curvature along* $\dot{\gamma}$ *can be estimated by the ratio of metric determinants:*

$$r(\boldsymbol{z}^{(i)}, \boldsymbol{z}^{(j)}) := \frac{\det \boldsymbol{G}_i}{\det \boldsymbol{G}_j} \approx 1 - \frac{1}{3} Ric(\dot{\gamma}). \tag{6}$$

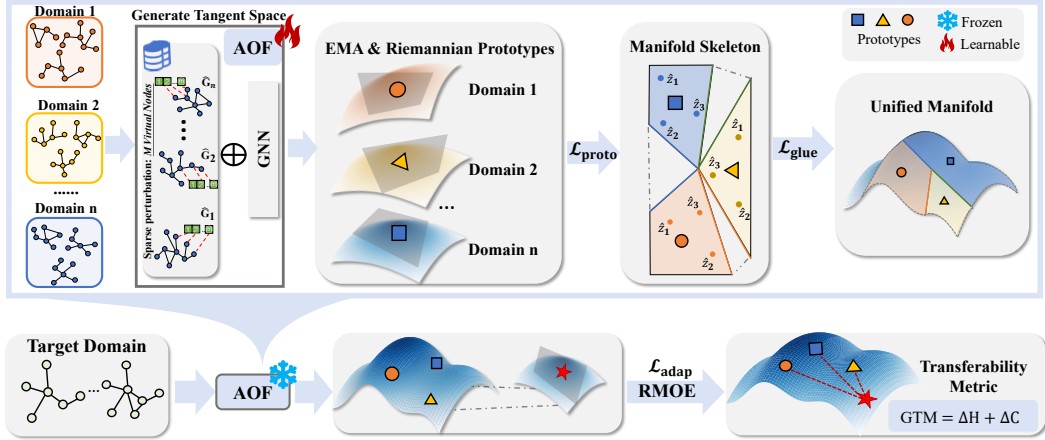

Figure 2: An Illustration of GRAPHGLUE Framework.

Accordingly, the volume element $\sqrt{\det \boldsymbol{G}_i}$ varies smoothly along the path of length $k$, implying that the Ricci curvature changes continuously along that path, referred to as Log-Determinant $k$-order smoothness. Thus, we can investigate the $k$-order smoothness with a scalar field of volume element, and formulate a Ricci curvature loss which encourages the glued manifold to be smooth.

**Definition 4.10** ($k$-order Smoothness and Curvature Loss). *Define $g_i = \frac{1}{2} \log \det \boldsymbol{G}_i$ as a scalar field over $\mathcal{G}$, representing the logarithmic volume density at node $v_i$. We say the manifold structure exhibits* **log-determinant smoothness** *if $\boldsymbol{g} \in \mathbb{R}^{|\mathcal{V}|}$ minimizes the graph Dirichlet energy: $\mathcal{E}_{Dir}[\boldsymbol{g}] = \|\boldsymbol{L}^k \boldsymbol{g}\|^2$, where $\boldsymbol{L}$ is the (normalized) Laplacian of $\mathcal{G}$. In light of computational efficiency in practice, we define the curvature loss function of $2$-order smoothness as follows,*

$$\mathcal{L}_{Curv}(\mathcal{G}) = \frac{1}{|\mathcal{A}|} \sum_{\mathcal{A}_{ijk}} |\log(r_{ij}) - \log(r_{jk})|^2 \tag{7}$$

**Geometric Scaling Law** Consequently, any graph datasets are merged into a unified, smooth Riemannian manifold, allowing us to study knowledge transfer within the framework of differential geometry. As the quantities of graphs increase, $(\mathcal{F}, \boldsymbol{G}, \boldsymbol{P})$ approximates an ideal manifold, and thus we deduce a geometric scaling law that larger quantities of datasets improve model transferability with a smoother manifold, which is empirically validated in Sec. 6.2.

**Theorem 4.11** (Gluing into a Smooth Manifold, Appendix B.6). *For any graph dataset $\mathbb{G}$, if $\boldsymbol{G}$ is log-determinant $\infty$-order smooth, and $\boldsymbol{P}$ is trivial with induced metric-preserving diffeomorphism $\phi$, then $(\mathcal{F}, \boldsymbol{G}, \boldsymbol{P})$ glues to a smooth Riemannian manifold $(\mathcal{F}, \boldsymbol{G})$, where $\mathcal{F} = (\bigcup_\phi)_{i=1}^N U_i$.*

## 5 GRAPHGLUE: GEOMETRIC MULTI-DOMAIN GRAPH PRE-TRAINING

Building on our theory of neural manifold gluing, we present a novel pretraining-adaptation framework, **GRAPHGLUE**, as illustrated in Fig. 2. The pre-training first learns the local geometry and then glues these local pieces together as introduced in Sec. 4. Moreover, before gluing, an *Exponential Moving Average (EMA) prototyping* is proposed to distinguish domain semantics through different locations on the manifold, while enabling batched pre-training to efficiently handle large-scale graphs. Then, we leverage prompt adaptation and a Riemannian Mixture-of-Experts (MoE), while gluing the target domain to the pre-trained manifold for geometric consistency. A *Geometric Transfer Metric (GTM)* is naturally induced to measure the transfer difficulty. The overall procedure is summarized in Algorithm 1.

### 5.1 PRE-TRAINING WITH EMA PROTOTYPING

For multi-domain source graphs $\mathbb{S} = \{\mathcal{S}_1, ..., \mathcal{S}_K\}$, we associate each graph with a Riemannian prototype, which is given as a tuple of global location and Riemannian metrics, $(\boldsymbol{z}^{\mathcal{S}_k}, \log \boldsymbol{G}^{\mathcal{S}_k} =$

$\left( \frac{1}{|\mathcal{S}_k|} \sum_{\mathcal{G} \in \mathcal{S}_k} \boldsymbol{z}^{\mathcal{G}}, \frac{1}{|\mathcal{S}_k|} \sum_{\mathcal{G} \in \mathcal{S}_k} \log \boldsymbol{G}(\boldsymbol{z}^{\mathcal{G}}) \right)$. The challenges of Riemannian prototyping are dual: computation efficiency for large-scale graphs, and semantics distinction across different domains. To address the first challenge, we develop an EMA for Riemannian prototyping. For each batch, we perform the following updating rules,

$$\boldsymbol{z}^{\mathcal{S}_k} \leftarrow \beta \boldsymbol{z}^{\mathcal{S}_k} + (1 - \beta) \frac{1}{|\mathcal{B}_k|} \sum_{\mathcal{G} \in \mathcal{B}_k} \boldsymbol{z}^{\mathcal{G}} \tag{8}$$

$$\log \boldsymbol{G}^{\mathcal{S}_k} \leftarrow \beta \log \boldsymbol{G}^{\mathcal{S}_k} + (1 - \beta) \frac{1}{|\mathcal{B}_k|} \sum_{\mathcal{G} \in \mathcal{B}_k} \log \boldsymbol{G}(\boldsymbol{z}^{\mathcal{G}}), \tag{9}$$

where $\beta \in (0, 1)$ is a momentum coefficient, and $\log$ means matrix logarithm. This EMA update ensures that $(\boldsymbol{z}^{\mathcal{S}_k}, \log \boldsymbol{G}^{\mathcal{S}_k})$ gradually converge to the stable average value throughout pre-training Morales-Brotons et al. (2024); Izmailov et al. (2019). Note that the metric matrix belongs to a symmetric positive-definite manifold, and we utilize the log update, different from the traditional ones. To address the second challenge, we incorporate a sample-prototype contrastive loss that encourages graph prototypes to be well separated on the manifold, distinguishing domain semantics.

$$\mathcal{L}_{\text{proto}}(\mathcal{G}) = -\frac{1}{K} \sum_{k=1}^{K} \log \frac{\exp(\text{sim}(\boldsymbol{z}^{\mathcal{G}}, \boldsymbol{z}^{\mathcal{S}_k})/\tau)}{\sum_{j=1}^{K} \exp(\text{sim}(\boldsymbol{z}^{\mathcal{G}}, \boldsymbol{z}^{\mathcal{S}_j})/\tau)}. \tag{10}$$

## 5.2 Consistent Adaptation & Quantifiable Transferability

GRAPHGLUE employs prompt adaptation and Riemannian MoE to generate representations, while we emphasize geometric consistency between the pre-trained manifold and target graphs by "gluing". To be specific, for a target sample $\mathcal{G}^{\mathcal{T}}$, we **first** infer the global coordinates and local metric through prompting. With the coordinates $\boldsymbol{z}^{\mathcal{T}}$, local metric $\boldsymbol{G}_{\boldsymbol{z}}$ and basis vectors of the tangent space $\boldsymbol{W}^{\mathcal{T}} = [\boldsymbol{w}_1^{\mathcal{T}}, \ldots, \boldsymbol{w}_M^{\mathcal{T}}]$ given by the pre-trained model, we introduce a learnable prompt matrix $\boldsymbol{Q} \in \mathbb{R}^{d \times d}$. The global coordinates is adapted as $\boldsymbol{z}^{\text{adapt}} = \boldsymbol{Q} \boldsymbol{z}^{\mathcal{T}}$. Note that the metric adaptation is challenging owing to the orthogonal requirement of basis vectors. Thus, instead of prompting the pre-trained local metric, we apply the prompt matrix $\boldsymbol{Q}$ to $\boldsymbol{W}^{\mathcal{T}}$, and the adapted local metric is derived as $\boldsymbol{G}^{\text{adapt}} = \text{diag}\left( \|\boldsymbol{Q} \boldsymbol{w}_1^{\mathcal{T}}\|^2, \ldots, \|\boldsymbol{Q} \boldsymbol{w}_M^{\mathcal{T}}\|^2 \right)$, where $\boldsymbol{w}^{\mathcal{T}}$ are the basis vectors undergoing the proposed adaptive orthogonal frame. **Second**, to ensure consistency, we glue the target sample to the pre-trained Riemannian manifold $\mathcal{F}$, where Riemannian prototypes are treated as the anchors to align the target. In particular, we construct a transfer graph $\mathcal{G}_0$ by connecting the target to its $k$-nearest prototypes, and apply $\mathcal{L}_{\text{holo}}(\mathcal{G}_0)$ and $\mathcal{L}_{\text{curv}}(\mathcal{G}_0)$ proposed in Sec. 4, penalizing non-trivial holonomy and abrupt volume changes, respectively. **Third**, we present a Riemmanian MoE where each Riemannian prototype $(\boldsymbol{z}^{\mathcal{S}_k}, \log \boldsymbol{G}^{\mathcal{S}_k})$ serves as an expert and its weight is given by a gating function $\beta_k = \boldsymbol{g}_k(\boldsymbol{z}^{\text{adapt}}, \boldsymbol{G}^{\text{adapt}})$. This MoE generates $\log \boldsymbol{G}^{\text{align}} = \sum_{k=1}^{K} \beta_k \log \boldsymbol{G}^{\mathcal{S}_k}$. summarized from the experts. Accordingly, we obtain the final representation $\boldsymbol{z}_{\text{task}} = \left[ \boldsymbol{z}^{\mathcal{T}}; \log \boldsymbol{G}^{\text{adapt}}; \log \boldsymbol{G}^{\text{align}} \right]$, where $\boldsymbol{z}_{\text{task}} \in \mathbb{R}^{d+2M}$ since $\log \boldsymbol{G}^{\text{adapt}}, \log \boldsymbol{G}^{\text{align}}$ are both diagonal matrix that can be vectorized. The overall adaptation loss is given as

$$\mathcal{L}_{\text{adap}} = \mathcal{L}_{\text{task}}(\boldsymbol{z}_{\text{task}}; \boldsymbol{y}_{\text{task}}) + \lambda \mathcal{L}_{\text{glue}}, \qquad \mathcal{L}_{\text{glue}} = \mathcal{L}_{\text{holo}}(\mathcal{G}_0) + \mathcal{L}_{\text{curv}}(\mathcal{G}_0), \tag{11}$$

where $\lambda$ balances task-specific learning with consistency, and $\boldsymbol{y}_{\text{task}}$ is the label of downstream task.

**On Transferability** Within the framework of differential geometry, we are able to systematically analyze knowledge transfer across different domains, and transfer effort of GRAPHGLUE is naturally measured by the geometric compatibility. We introduce *Geometric Transfer Metric* (GTM) which is defined as the minimal geometric deformation required to merge the target $\mathcal{G}^{\mathcal{T}}$ into the pre-trained manifold $\mathcal{F}$ without disrupting its learned local geometry. GTM is computed along with the adaptation and decomposes into two *interpretable* components as follows,

$$\text{GTM}(\mathcal{G}^{\mathcal{T}}; \mathcal{S}) = \Delta H + \Delta C, \qquad \Delta H = \mathcal{L}_{\text{holo}}(\mathcal{G}_0), \qquad \Delta C = \mathcal{L}_{\text{curv}}(\mathcal{G}_0). \tag{12}$$

1. **Holonomy disagreement** $\Delta H$. It measures how the holonomy map deviates from identity along paths connecting the target to its nearest prototype, interpreted as the "twisting" induced by $\mathcal{G}^{\mathcal{T}}$.

2. **Curvature disagreement** $\Delta C$. It is computed as the discrepancy between the volume element $\sqrt{\det \boldsymbol{G}_i}$, indicating the dismatch with respect to Ricci curvature according to Theorem 4.9. The natural interpretation is given as the "bending" or abrupt change in local volume.

Accordingly, a low GTM means that the target is seamlessly integrated $\mathcal{F}$ with trivial deformation, implying high transferability; in contrast, a high value shows that the target is geometrically alien, thus requiring significant effort to fit the geometry of $\mathcal{F}$. Different from similarity measures between source and target domains Wang et al. (2024), GTM examines the geometric consistency from GRAPHGLUE itself, and provides an interpretable assessment of transfer difficulty.

**Further Insight** The generalization error is related to the smoothness of the model objective Bartlett et al. (2017); Scaman & Virmaux (2018). In fact, GRAPHGLUE controls the smoothness by inducing a smooth global metric. Specifically, $\mathcal{L}_{\text{holo}}$ guarantees the topological continuity of gluing boundaries, while $\mathcal{L}_{\text{curv}}$ achieves $k$-order smooth by log-determinant smoothness in Definition 4.10, similar to Czarnecki et al. (2017). The complexity analysis is provided in Appendix D.2, D.3.

## 6 EXPERIMENTS

We conduct experiments on six representative domains to evaluate cross-domain transfer learning performance. Also, we examine the transferability measure (GTM), geometric scaling law, the effect of incorporating graphs of distinct semantics, and the geometric interpretation. Ablation study, hyperparameter sensitivity and performance on heterophilic graphs are in Appendix G.2, G.3, G.4.

### 6.1 EXPERIMENTAL SETUPS

**Datasets & Baselines** We carefully select 6 representative benchmark datasets, covering various domains: an academic citation network `Arxiv`, a product co-purchase graph `Computers`, a social network `Reddit`, a knowledge graph `FB15k_237`, and benchmarks on bioinformatics `PROTEINS` and chemoinformatics `HIV`. We compare GRAPHGLUE against baselines from 3 main categories: (1) Supervised GNNs: GCN Kipf & Welling (2017), GraphSAGE Hamilton et al. (2017), and GIN Xu et al. (2019). (2) Self-Supervised GNNs: DGI Veličković et al. (2019), GraphMAE Hou et al. (2022), and GCC Qiu et al. (2020). (3) Graph Foundation Models: PRODIG Huang et al. (2023), GFT Wang et al. (2024), RAGraph Jiang et al. (2024), SAMGPT Yu et al. (2025a), GCOPE Zhao et al. (2024), and MDGFM Wang et al. (2025a). Detailed descriptions are specified in Appendix F.

**Evaluation Protocol** Our evaluation adopts a leave-one-out cross-domain setup, where models are pre-trained on five source datasets and fine-tuned on a single held-out target dataset. We use a few-shot fine-tuning setting, leveraging $k$ labeled samples per class ($k \in \{1, 5\}$) from the target task for adaptation. The remaining target data is randomly split into 10% for validation and 90% for testing. We evaluate performance on three tasks: node/edge classification measured by ACC and graph classification measured by AUC. All reported results are the average of 10 independent runs.

### 6.2 RESULTS AND DISCUSSION

**Main Results on Cross-domain Transfer Learning** As shown in Table 1, the empirical results demonstrate the superior effectiveness of GRAPHGLUE in challenging few-shot scenarios. For instance, in the 1-shot setting, it outperforms the strongest baselines on `Computers` and `Reddit` by significant margins of 4.9% and 2.3%, respectively. This strong performance is often maintained as more data becomes available. In the 5-shot setting on the `Reddit` dataset, GRAPHGLUE achieves 85.0% ACC, exceeding the runner-up by 4.6%. These results suggest that the geometric construction of GRAPHGLUE enhances the model performance, and we will demonstrate additional benefits of the constructed smooth manifold in the following parts.

Ablation study on the effectiveness of proposed $\mathcal{L}_{\text{curv}}$ and $\mathcal{L}_{\text{holo}}$ are provided in Appendix G, showing that both gluing via holonomy and smoothing via Ricci curvature are important to downstream tasks.

**On Transferability Measure** This part shows how the proposed measure of GMT aligns with the transfer effort of the pre-trained model. To this end, we pre-train the model in `Arxiv`, `Reddit`, `FB15k_237`, `PROTEINS`, and `HIV` datasets, then conduct transfer settings on `Computers` with 2000 epochs. In this case, holonomy loss vanishes rapidly during training, and thus we investigate the curvature loss in Figure 3, where $x$-axis is the training epoch. We plot the test task loss of cross-entropy for the classification task on the $y$-axis on the left. In the top of Figure 3, we find that, as

Table 1: Performance of cross-domain transfer on various downstream tasks, reported as mean $\pm$ std over 10 runs. The highest result is **bolded**, and the runner-up is underlined.

| Model | Node Classification | | | | | | Link Classification | | Graph Classification | |
|---|---|---|---|---|---|---|---|---|---|---|
| | Arxiv | | Computers | | Reddit | | FB15k_237 | | PROTEINS | |
| | 1-shot | 5-shot | 1-shot | 5-shot | 1-shot | 5-shot | 1-shot | 5-shot | 1-shot | 5-shot |
| GCN | $12.6_{\pm1.7}$ | $27.6_{\pm2.1}$ | $33.8_{\pm3.8}$ | $65.7_{\pm4.2}$ | $11.1_{\pm2.1}$ | $28.3_{\pm1.0}$ | $32.1_{\pm2.3}$ | $52.4_{\pm1.8}$ | $50.1_{\pm13.0}$ | $55.0_{\pm9.9}$ |
| GraphSAGE | $14.6_{\pm3.7}$ | $26.1_{\pm2.2}$ | $35.4_{\pm8.2}$ | $66.7_{\pm4.4}$ | $14.6_{\pm2.3}$ | $22.2_{\pm1.1}$ | $35.7_{\pm2.1}$ | $58.9_{\pm1.5}$ | $58.9_{\pm2.7}$ | $60.4_{\pm1.3}$ |
| GIN | $11.2_{\pm2.0}$ | $26.0_{\pm2.4}$ | $44.7_{\pm6.0}$ | $69.5_{\pm3.5}$ | $18.5_{\pm1.8}$ | $29.0_{\pm1.6}$ | $38.2_{\pm2.5}$ | $63.7_{\pm1.7}$ | $54.2_{\pm13.5}$ | $58.8_{\pm5.0}$ |
| GCC | $12.6_{\pm2.0}$ | $26.8_{\pm2.1}$ | $34.8_{\pm6.1}$ | $62.6_{\pm3.1}$ | $54.7_{\pm5.6}$ | $65.2_{\pm1.5}$ | $47.8_{\pm1.9}$ | $73.6_{\pm1.2}$ | $59.2_{\pm7.9}$ | $64.2_{\pm3.0}$ |
| DGI | $13.3_{\pm3.3}$ | $27.1_{\pm2.3}$ | $35.2_{\pm7.5}$ | $61.0_{\pm3.2}$ | $60.0_{\pm4.8}$ | $62.7_{\pm2.2}$ | $42.5_{\pm2.0}$ | $68.3_{\pm1.4}$ | $53.1_{\pm8.4}$ | $53.3_{\pm6.2}$ |
| GraphMAE | $12.6_{\pm1.7}$ | $27.6_{\pm2.1}$ | $33.8_{\pm3.8}$ | $65.7_{\pm4.2}$ | $11.1_{\pm2.1}$ | $28.3_{\pm1.0}$ | $51.3_{\pm1.8}$ | $77.2_{\pm1.0}$ | $\mathbf{60.1_{\pm13.0}}$ | $65.0_{\pm9.9}$ |
| PRODIGY | $\underline{28.4_{\pm2.2}}$ | $33.6_{\pm2.8}$ | $45.3_{\pm4.1}$ | $52.7_{\pm3.6}$ | $35.6_{\pm3.2}$ | $42.3_{\pm2.9}$ | $53.5_{\pm1.0}$ | $72.1_{\pm6.9}$ | $48.9_{\pm5.4}$ | $55.2_{\pm4.7}$ |
| GFT | $26.5_{\pm2.4}$ | $36.7_{\pm1.9}$ | $\underline{54.6_{\pm4.0}}$ | $69.1_{\pm3.5}$ | $58.8_{\pm2.5}$ | $66.2_{\pm1.4}$ | $58.0_{\pm1.3}$ | $79.1_{\pm1.6}$ | $55.4_{\pm5.8}$ | $62.1_{\pm3.5}$ |
| RAGraph | $18.7_{\pm2.5}$ | $32.3_{\pm1.7}$ | $46.2_{\pm4.3}$ | $62.3_{\pm3.7}$ | $52.5_{\pm3.4}$ | $63.0_{\pm1.3}$ | $52.1_{\pm3.0}$ | $64.5_{\pm2.5}$ | $51.4_{\pm5.1}$ | $58.6_{\pm2.8}$ |
| SAMGPT | $24.1_{\pm3.8}$ | $34.4_{\pm2.2}$ | $47.6_{\pm7.4}$ | $60.8_{\pm3.6}$ | $62.8_{\pm4.2}$ | $75.1_{\pm1.6}$ | $57.4_{\pm2.4}$ | $77.6_{\pm2.7}$ | $52.4_{\pm3.1}$ | $59.1_{\pm2.6}$ |
| GCOPE | $26.5_{\pm5.5}$ | $\mathbf{39.1_{\pm1.9}}$ | $54.5_{\pm9.1}$ | $\underline{72.2_{\pm2.8}}$ | $62.7_{\pm4.5}$ | $\underline{80.4_{\pm0.7}}$ | $\underline{58.2_{\pm2.6}}$ | $\underline{79.3_{\pm2.2}}$ | $55.1_{\pm3.5}$ | $64.8_{\pm2.4}$ |
| MDGFM | $26.0_{\pm2.4}$ | $32.2_{\pm1.7}$ | $46.6_{\pm8.4}$ | $64.0_{\pm5.3}$ | $\underline{64.8_{\pm3.3}}$ | $76.5_{\pm1.7}$ | $56.1_{\pm1.6}$ | $77.6_{\pm2.0}$ | $53.4_{\pm5.3}$ | $57.7_{\pm3.4}$ |
| GRAPHGLUE | $\mathbf{28.8_{\pm5.2}}$ | $\underline{37.0_{\pm2.3}}$ | $\mathbf{59.5_{\pm7.0}}$ | $\mathbf{73.2_{\pm0.7}}$ | $\mathbf{67.1_{\pm3.3}}$ | $\mathbf{85.0_{\pm1.1}}$ | $\mathbf{59.7_{\pm5.2}}$ | $\mathbf{81.5_{\pm2.3}}$ | $\underline{59.8_{\pm4.8}}$ | $\mathbf{65.3_{\pm2.4}}$ |

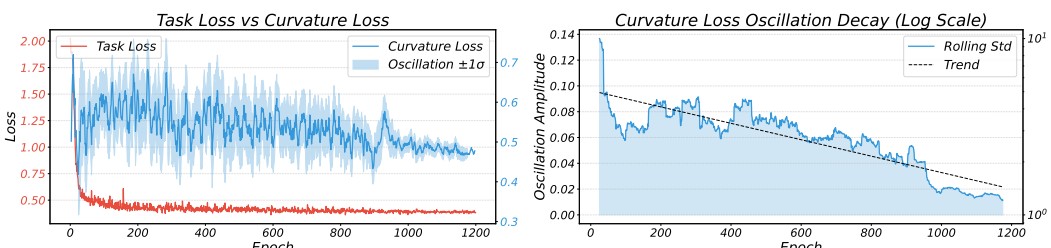

Figure 3: GTM vs Test Task Loss.

curvature loss decreases and converges, the test task loss exhibits the same pattern, and it suggests that GMT measures the effort of training the pre-trained model in transfer setting. Moreover, at the bottom of Figure 3, it shows another feature of curvature loss that the convergence of its oscillation amplitude implies the convergence of the test task loss, which meets the theory in Keskar et al. (2017); Czarnecki et al. (2017).

**Case Study** We conduct an interesting case study to examine the effect of including semantically distinct data during pre-training. To this end, we incrementally expand a `Reddit`-only pre-training with the distinct `PROTEINS` and `HIV` datasets, and consistently evaluate on `Reddit` under the 1-shot setting. As shown in Figure 4, GRAPHGLUE achieves a steady improvement with the inclusion of each dataset. In contrast, GCOPE suffers from negative transfer and results in possible performance decline. This result provides evidence that GRAPHGLUE can effectively incorporate knowledge from even vastly different domains to enhance its capabilities.

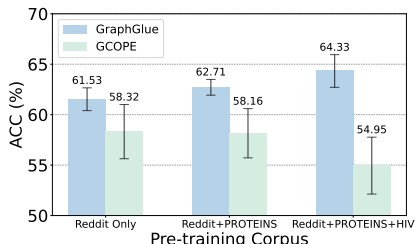

Figure 4: Effect of including distinct domains during pre-training.

**On Geometric Scaling Law** We validate the geometric scaling law by enlarging the quantities of pre-training datasets. Specifically, we show the few-shot performance on `Computers` and `Reddit` in Figure 5, where the original datasets are same as that in Figure 3, denoted as $+0$, and we incrementally incorporate `Pubmed`, `Photo`, `FacebookPagePage`, `WordNet18RR`, `MUATG` and `Lipophilicity` in order, referred to as $+1, +2, +3, +4, +5$ and $+6$, respectively. In the 1-shot setting, average accuracy rises steadily while transfer loss drops consistently, both well-fitted by logarithmic functions (blue curves), and thus it exhibits clear scaling laws. 5-shot performance remains more stable (red curves), with only marginal gains in accuracy and a slight reduction in loss. The insight is that, under extreme data scarcity (1-shot), the performance is highly sensitive to

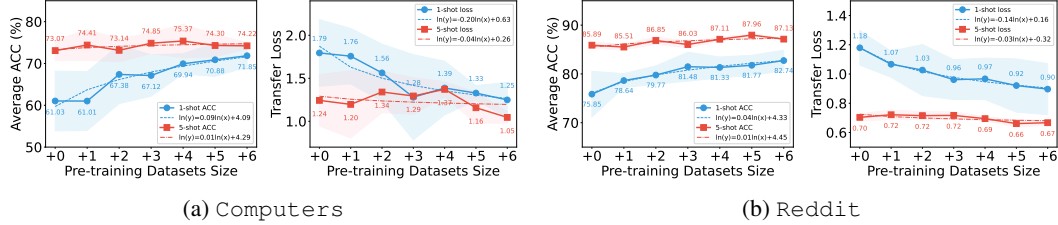

Figure 5: Geometric scaling law on (a) `Computers` and (b) `Reddit` datasets.

the pre-trained model's capacity, the expressive power of the learned manifold, while more labeled samples restrain such scaling effect. The observed logarithmic scaling supports our claim on the scaling law.

**Visualization & Geometric Interpretation**   To illustrate our intuition, we visualize a 3D per-trained manifold on the 6 datasets in Figure 6, where the configuration is detailed in Appendix G. We observe that the datasets—`Reddit` (social network), `Arxiv` (citation network), `Computers` (e-commerce network), and `FB15k_237` (knowledge graph)—exhibit substantial semantic overlap while retaining the difference. Their corresponding regions on the manifold lie in close proximity, sometimes intermingling owing to shared semantics, yet remain distinguishable. The two chemistry-related datasets (`PROTEINS` and `HIV`) are well-separated from the others on the learned manifold. That is, the proposed neural manifold gluing captures the complicated domain semantics. Also, the smoothness is generally ensured, facilitating knowledge transport along the manifold. The visualization underscores our framework's ability to unify

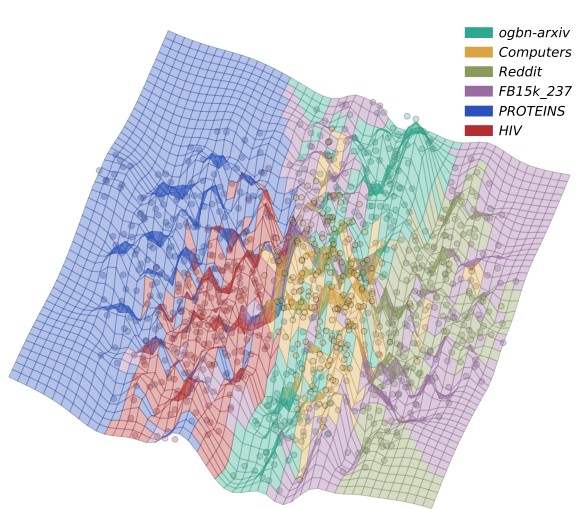

Figure 6: Visualization of the pre-trained manifold from 6 datasets.

diverse domains into a coherent geometric structure, which forms the foundation for effective cross-domain transfer.

## 7 CONCLUSION

This work present the first multi-domain graph pre-training framework through the lens of Riemannian geometry, enabling the merging of arbitrary graph datasets into a unified, smooth Riemannian manifold and facilitating a principled understanding of knowledge transfer across different graphs. The theoretical contribution lies in the establishment of neural manifold gluing, which "glues" the local pieces together into a coherent whole. Building on this theory, we introduce the GRAPHGLUE framework, supporting the batched pre-training and providing a means to measure its transferability. Furthermore, we empirically validate the geometric scaling law of GRAPHGLUE.

## ACKNOWLEDGEMENT

This work is supported in part by NSFC under grants 62202164. Philip S. Yu is supported in part by NSF under grants III-2106758, and POSE-2346158.

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

APPENDIX: TABLE OF CONTENT

# A NOTATIONS

Table 2: Notation and Description

| Notation | Description |
|---|---|
| $\mathcal{M}$ | A smooth manifold |
| $\mathbf{G}$ | A Riemannian metric tensor |
| $\boldsymbol{p}$ | A point in $\mathcal{M}$ |
| $\mathcal{T}_{\boldsymbol{p}}\mathcal{M}$ | The tangent space of point p on $\mathcal{M}$ |
| $|\boldsymbol{G}(\boldsymbol{p})|$ | The volume element at $\boldsymbol{p}$ |
| $(U, x^1, ..., x^M)$ | A coordinate chart of tangent space $\mathcal{T}_{\boldsymbol{p}}\mathcal{M}$ |
| $\{\frac{\partial}{\partial x^1}, ..., \frac{\partial}{\partial x^M}\}, \{\partial_i\}$ | The standard frame of $\mathcal{T}\mathcal{M}$ |
| $\text{Ric}(X, Y)$ | Ricci curvature for vector fields $X, Y$ |
| $\mathcal{G} = (\mathcal{V}, \mathcal{E})$ | A graph with node set $\mathcal{V}$ and edge set $\mathcal{E}$ |
| $\boldsymbol{X} \in \mathbb{R}^{|\mathcal{V}| \times F}$ | A feature matrix with node set $\mathcal{V}$ |
| $\mathcal{G}^t$ | The graph of target domain $\mathcal{D}^t$ |
| $\mathbb{G} = \{\mathcal{G}^1, \mathcal{G}^2, \cdots, \mathcal{G}^K\}$ | A collection of $K$ graphs from $L$ domains $\mathbb{D}$ |
| $\mathbb{D} = \{\mathcal{D}^1, \mathcal{D}^2, \cdots, \mathcal{D}^L\}$ | $L$ domains |
| $f_{\boldsymbol{\Theta}}(\text{GNN}(\cdot))$ | A pretrained model on the graph dataset $\mathbb{G}$ with an encoder $\text{GNN}(\cdot)$ |
| $\{\boldsymbol{\Theta}_f^{\star}, \boldsymbol{\Theta}_{\text{GNN}}^{\star}\}$ | The pre-training parameters |
| $f_{\text{GNN}} : \mathbb{G} \to \mathbb{R}^d$ | A differentiable encoder from manifold $\mathbb{G}$ into the Euclidean space |
| $\boldsymbol{w}_m$ | The set of tangent vectors of graph $\mathcal{G}$ |
| $D_{\boldsymbol{v}}f$ | The directional derivative $\mathcal{G}$ |
| $\boldsymbol{H}(\mathcal{C})$ | The holonomy map of cycle $\mathcal{C}$ |
| $\mathcal{E}_{\text{Dir}}[g]$ | The graph Dirichlet energy of function $g$ |
| $\mathcal{S}_i(\mathcal{V}_i, \mathcal{E}_i)$ | The $h$-hop neighborhood centered at $v_i$ with node set $\mathcal{V}_i$ and edge set $\mathcal{E}_i$ within this subgraph |
| $\hat{\mathcal{G}}_m = (\hat{\mathcal{V}}_m, \hat{\mathcal{E}}_m)$ | The augmented graph |
| $\mathbb{P}[M, d]$ | The Adaptive Orthogonal Frame |
| $U_i$ | A neighborhood around $\boldsymbol{z}^{(i)}$ |
| $\boldsymbol{W}^{(i)}$ | The basis of $T_{\boldsymbol{z}^{(i)}}U_i$ generated from AFB |
| $\boldsymbol{G}_i(\boldsymbol{u}, \boldsymbol{v})$ | The local Riemannian metric defined on $U_i$ |
| $\mathcal{S}_{++}^{M \times M}$ | The set of positive-definite matrix |
| $\mathbb{S} = \{\mathcal{S}_1, ..., \mathcal{S}_K\}$ | The source graph datasets |
| $(\boldsymbol{z}^{\mathcal{S}_k}, \log \boldsymbol{G}^{\mathcal{S}_k})$ | the Riemannian prototypes for each source graph dataset |
| $\mathcal{L}_{\text{proto}}(\mathcal{G})$ | The prototype-level contrastive loss |
| $\boldsymbol{P}^{(i,j)}$ | The tangent edge translation |
| $\mathcal{A}$ | The set of all triangles $\mathcal{A}_{ijk} = ((v_i, v_j), (v_j, v_k))$ |
| $\mathcal{L}_{\text{hol}}(\mathcal{G})$ | The holonomy loss |
| $r(\boldsymbol{z}^{(i)}, \boldsymbol{z}^{(j)})$ | The overall sign of the Ricci curvature along the geodesic $\gamma(t)$ between $\boldsymbol{z}_i$ and $\boldsymbol{z}_j$ |
| $\mathcal{L}_{\text{Curv}}(\mathcal{G})$ | The curvature loss regularizing the change of curvature by controlling the volume change ratio |
| $\mathcal{G}^{\mathcal{T}} = (\mathcal{V}^{\mathcal{T}}, \mathcal{E}^{\mathcal{T}})$ | An unseen graph |
| $\boldsymbol{W}^{\text{adapt}}, \boldsymbol{G}^{\text{adapt}}$ | The adaptive tangent vectors and adaptive metric |
| $\boldsymbol{Q}$ | The prompt matrix |
| $\log \boldsymbol{G}^{\text{align}}$ | The aligned log-metric to give a $K$-dimensional weighted vector |
| $\mathcal{L}_{\text{adap}}$ | The adaptation loss |
| $\lambda$ | The balance coefficient of task loss and gluing loss |
| $\Delta H$ | Holonomy disagreement |
| $\Delta C$ | Curvature disagreement |

## B   PROOFS

### B.1   PROOF OF THEOREM 4.3

**Theorem 4.3** (Upper bound of Tangent Vector Length) *Given a connected $\mathcal{G}$ with $N$ nodes, $\boldsymbol{A}$, $\boldsymbol{L}$ the adjacency matrix and Laplacian of $\mathcal{G}$, and $\boldsymbol{P}$ the feature matrix of perturbation nodes. Apply $(k, M)$-sparse perturbation to $\mathcal{G}$, suppose $\frac{kM}{N} = \varepsilon$, where $\varepsilon > 0$ is small, and added edge weights satisfy $\sum_l h(v_i, p_l) = 1$. Then,*

$$\|\boldsymbol{w}_m^{\boldsymbol{p}}\| \leq (1 + \varepsilon)\|\boldsymbol{P}\|$$

*holds, where $\boldsymbol{w}_m^{\boldsymbol{p}}$ is the component of $\boldsymbol{w}_m$ determined by perturbation.*

*Proof.* We denote the weighted matrix $\boldsymbol{H} \in \mathbb{R}^{N \times M}$ that consists of $h(v_i, p_l)$, the row summation $\boldsymbol{r_H} = \boldsymbol{H}\boldsymbol{1}_M$. Then, the perturbed adjacency matrix $\hat{\boldsymbol{A}}$ is

$$\hat{\boldsymbol{A}} = \left[ \begin{array}{cc} \boldsymbol{A} & \boldsymbol{H} \\ \boldsymbol{H}^\top & \boldsymbol{I}_M \end{array} \right] \in \mathbb{R}^{(N+M) \times (N+M)}.$$

The perturbed Laplacian is

$$\hat{\boldsymbol{L}} = \left[ \begin{array}{cc} \boldsymbol{L} + \operatorname{diag}(\boldsymbol{r_H}) & -\boldsymbol{H} \\ -\boldsymbol{H}^\top & \boldsymbol{I}_M \end{array} \right] \in \mathbb{R}^{(N+M) \times (N+M)}. \tag{13}$$

Let the $d$-dimensional graph signal $\boldsymbol{F} \in \mathbb{R}^{(N+M) \times d}$ in heat diffusion on a perturbed graph $\hat{\mathcal{G}}$ be

$$\boldsymbol{F}(t) = \exp(-t\hat{\boldsymbol{L}})\boldsymbol{F}(0), t > 0, \tag{14}$$

$$\boldsymbol{F}(0) = \left[ \begin{array}{c} \boldsymbol{X} \\ \boldsymbol{P} \end{array} \right] \in \mathbb{R}^{(N+M) \times d}. \tag{15}$$

By the linearity of the heat equation, we can divide $\boldsymbol{F}(t)$ into two parts:

$$\boldsymbol{F}(t) = \exp(-t\hat{\boldsymbol{L}}) \left[ \begin{array}{c} \boldsymbol{X} \\ \boldsymbol{0} \end{array} \right] + \exp(-t\hat{\boldsymbol{L}}) \left[ \begin{array}{c} \boldsymbol{0} \\ \boldsymbol{P} \end{array} \right]. \tag{16}$$

We denote

$$\boldsymbol{F}_{\text{base}}(t) = \exp(-t\hat{\boldsymbol{L}}) \left[ \begin{array}{c} \boldsymbol{X} \\ \boldsymbol{0} \end{array} \right], \tag{17}$$

$$\boldsymbol{F}_{\text{pert}}(t) = \exp(-t\hat{\boldsymbol{L}}) \left[ \begin{array}{c} \boldsymbol{0} \\ \boldsymbol{P} \end{array} \right]. \tag{18}$$

We are only concerned about $\boldsymbol{F}_{\text{pert}}(t)$ since it reflects how perturbation $\boldsymbol{P}$ affects other nodes. Observe from the construction of $\hat{\boldsymbol{L}}$, the affected nodes are non-zero elements in $\boldsymbol{r_H}$. Let $\mathbb{S} = \operatorname{supp} \boldsymbol{r_H} \subset \{1, ..., N\}$, that $S := |\mathbb{S}| \leq kM$. We can extract the corresponding part of $\hat{\boldsymbol{L}}$:

$$\boldsymbol{L}_{\text{local}} = \left[ \begin{array}{cc} \boldsymbol{L}_{\mathbb{S}} & -\boldsymbol{H}_{\mathbb{S}} \\ -\boldsymbol{H}_{\mathbb{S}}^\top & \boldsymbol{I}_M \end{array} \right] \in \mathbb{R}^{(S+M) \times (S+M))}, \tag{19}$$

where

$$\boldsymbol{L}_{\mathbb{S}} = (\boldsymbol{L} + \operatorname{diag}(\boldsymbol{r_H}))_{[\mathbb{S}, \mathbb{S}]},$$
$$\boldsymbol{H}_{\mathbb{S}} = \boldsymbol{H}_{[\mathbb{S}, :]}.$$

Then we have

$$\boldsymbol{F}_{\text{pert}}(t) = \left[ \begin{array}{c} \boldsymbol{F}_{\text{pert}, N}(t) \\ \boldsymbol{F}_{\text{pert}, M}(t) \end{array} \right] = \left[ \begin{array}{c} \boldsymbol{S}\left( \exp(-t\boldsymbol{L}_{\text{local}}) \left[ \begin{array}{c} \boldsymbol{0}_S \\ \boldsymbol{P} \end{array} \right] \right)_{[1:S]} \\ \left( \exp(-t\boldsymbol{L}_{\text{local}}) \left[ \begin{array}{c} \boldsymbol{0}_S \\ \boldsymbol{P} \end{array} \right] \right)_{[S+1:S+M]} \end{array} \right], \tag{20}$$

where $\boldsymbol{S} \in \mathbb{R}^{N \times S}$ is an index projection matrix such that $(\boldsymbol{S})[i, j] = 1$ if $i = \mathbb{S}[j]$. To simplify the notation, let

$$\boldsymbol{K}(t) = \exp(-t\boldsymbol{L}_{\text{local}}) = \left[ \begin{array}{cc} \boldsymbol{K}_{SS}(t) & \boldsymbol{K}_{SM}(t) \\ \boldsymbol{K}_{MS}(t) & \boldsymbol{K}_{MM}(t) \end{array} \right] \in \mathbb{R}^{(S+M) \times (S+M))}. \tag{21}$$

We can find

$$\boldsymbol{F}_{\text{pert},N}(t) = \boldsymbol{S}\boldsymbol{K}_{SM}(t)\boldsymbol{P}, \tag{22}$$
$$\boldsymbol{F}_{\text{pert},M}(t) = \boldsymbol{K}_{MM}(t)\boldsymbol{P}. \tag{23}$$

As return to Eq. (16), we obtain

$$\boldsymbol{F}_N(t) = \boldsymbol{F}_{\text{base,N}}(t) + \boldsymbol{S}\boldsymbol{K}_{SM}(t)\boldsymbol{P}, \tag{24}$$
$$\boldsymbol{F}_M(t) = \boldsymbol{F}_{\text{base,M}}(t) + \boldsymbol{K}_{MM}(t)\boldsymbol{P}, \tag{25}$$

where $\boldsymbol{F}_{\text{base,N}}(t) = \boldsymbol{F}_{\text{base}}(t)_{[:N]}, \boldsymbol{F}_{\text{base,M}}(t) = \boldsymbol{F}_{\text{base}}(t)_{[N+1:N+M]}$.

We simply consider the global mean pooling operation that we obtain

$$\boldsymbol{z}(t) = \frac{1}{N}\boldsymbol{1}_N^\top \boldsymbol{F}_N(t) \in \mathbb{R}^d, \tag{26}$$

$$\boldsymbol{w}_m(t) = \boldsymbol{f}_m(t) - \boldsymbol{z}(t) \in \mathbb{R}^d, \tag{27}$$

where $\boldsymbol{f}_m(t) = \boldsymbol{F}_M(t)_{[m]}$, the $m$-th row of $\boldsymbol{F}_M$.

Similarly, we can still divide $\boldsymbol{w}_m(t)$ into two parts as

$$\boldsymbol{w}_m(t) = \boldsymbol{f}_m^{\boldsymbol{p}}(t) - \boldsymbol{z}^{\boldsymbol{p}}(t) + (\text{term affects by } \boldsymbol{X}), \tag{28}$$

where $\boldsymbol{f}_m^{\boldsymbol{p}}(t)$ is the $m$-th row of $\boldsymbol{K}_{MM}(t)\boldsymbol{P}$, and

$$\boldsymbol{z}^{\boldsymbol{p}}(t) = \frac{1}{N}\boldsymbol{1}_N^\top \boldsymbol{S}\boldsymbol{K}_{SM}(t)\boldsymbol{P} = \frac{1}{N}(\boldsymbol{1}_S^\top \boldsymbol{K}_{SM}(t))\boldsymbol{P}. \tag{29}$$

Let $\boldsymbol{a}(t) = \frac{1}{N}\boldsymbol{K}_{SM}^\top(t)\boldsymbol{1}_S \in \mathbb{R}^M$, then we have $\boldsymbol{z}^{\boldsymbol{p}}(t) = \boldsymbol{a}^\top(t)\boldsymbol{P}$. We denote $\boldsymbol{w}_m^{\boldsymbol{p}}(t) = \boldsymbol{f}_m^{\boldsymbol{p}}(t) - \boldsymbol{z}^{\boldsymbol{p}}(t)$, then

$$\boldsymbol{w}_m^{\boldsymbol{p}}(t) = [\boldsymbol{K}_{MM}(t)\boldsymbol{P}]_{[m]} - \frac{1}{N}\boldsymbol{a}^\top(t)\boldsymbol{P} = \left[ \boldsymbol{K}_{MM}(t)_{[m,:]} - \boldsymbol{a}^\top(t) \right] \boldsymbol{P}. \tag{30}$$

Let $\boldsymbol{b}(t) = \boldsymbol{K}_{MM}(t)_{[m,:]} - \boldsymbol{a}(t)$, we have $\boldsymbol{w}_m^{\boldsymbol{p}}(t) = \boldsymbol{b}^\top(t)\boldsymbol{P}$.

Since $\frac{kM}{N} = \varepsilon$, $S \le kM = \varepsilon N$, we obtain

$$\|\boldsymbol{a}(t)\| = \|\frac{1}{N}(\boldsymbol{1}_S^\top \boldsymbol{K}_{SM}(t))\| \le \frac{S}{N} \max_{i,j} |\boldsymbol{K}_{SM}(t)_{[i,j]}| \le \frac{S}{N} \le \varepsilon, \tag{31}$$

which means

$$\|\boldsymbol{w}_m^{\boldsymbol{p}}(t)\| \le (\|\boldsymbol{K}_{MM}(t)_{[m,:]}\| + \|\boldsymbol{a}(t)\|)\|\boldsymbol{P}\| \le (1 + \varepsilon)\|\boldsymbol{P}\|, \tag{32}$$

since each element is finite, and the diagonal elements of the heat kernel matrix are near 1 while the other elements are less than 1. Then, we complete the proof. $\square$

## B.2 Proof of Theorem 4.5

**Theorem 4.5** (Edge Tangent Translation as Isometry) *The tangent edge translation in Definition 4.4 is the optimal solution of*

$$\min_{\boldsymbol{P} \in GL(M)} \left\| \boldsymbol{P}^\top \boldsymbol{G}_j \boldsymbol{P} - \boldsymbol{G}_i \right\|_F^2, \tag{33}$$

*where* GL *denotes the general linear group, such that* $\boldsymbol{G}_j(\boldsymbol{P}^{(i,j)}\boldsymbol{u}, \boldsymbol{P}^{(i,j)}\boldsymbol{v}) = \boldsymbol{G}_i(\boldsymbol{u}, \boldsymbol{v})$, *which induces an isometry* $\phi^{(i,j)}$ *between* $\partial U_i$ *and* $\partial U_j$.

*Proof.* We prove that the tangent edge translation $\boldsymbol{P}^{(i,j)} = \boldsymbol{G}_j^{-1/2} \left( \boldsymbol{G}_j^{1/2} \boldsymbol{G}_i \boldsymbol{G}_j^{1/2} \right)^{1/2} \boldsymbol{G}_j^{-1/2}$ uniquely minimizes $\|\boldsymbol{P}^\top \boldsymbol{G}_j \boldsymbol{P} - \boldsymbol{G}_i\|_F^2$ over $\boldsymbol{P} \in \mathrm{GL}(M)$ and induces an isometry.

Let $\boldsymbol{Q} = \boldsymbol{G}_j^{1/2} \boldsymbol{P}$. Then $\boldsymbol{P}^\top \boldsymbol{G}_j \boldsymbol{P} = \boldsymbol{Q}^\top \boldsymbol{Q}$, and the objective becomes:

$$\min_{\boldsymbol{Q} \in \mathrm{GL}(M)} \|\boldsymbol{Q}^\top \boldsymbol{Q} - \boldsymbol{G}_i\|_F^2. \tag{34}$$

The minimum is achieved when $\boldsymbol{Q}^\top \boldsymbol{Q} = \boldsymbol{G}_i$, since the Frobenius norm is strictly convex over SPD matrices. Thus, $\boldsymbol{Q} = \boldsymbol{G}_i^{1/2} \boldsymbol{R}$ for orthogonal $\boldsymbol{R}$, and minimal norm occurs at $\boldsymbol{R} = \boldsymbol{I}$, giving $\boldsymbol{Q}^* = \boldsymbol{G}_i^{1/2}$.

From $\boldsymbol{Q} = \boldsymbol{G}_j^{1/2} \boldsymbol{P}$, we get candidate $\boldsymbol{P}_0 = \boldsymbol{G}_j^{-1/2} \boldsymbol{G}_i^{1/2}$. However, this is not symmetric in $\boldsymbol{G}_i, \boldsymbol{G}_j$ unless they commute. To ensure geometric consistency and symmetry, we instead use the *metric geometric mean*:

$$\boldsymbol{P}^{(i,j)} = \boldsymbol{G}_j^{-1/2} \left( \boldsymbol{G}_j^{1/2} \boldsymbol{G}_i \boldsymbol{G}_j^{1/2} \right)^{1/2} \boldsymbol{G}_j^{-1/2}. \tag{35}$$

Then, we compute

$$\boldsymbol{P}^{(i,j)\top} \boldsymbol{G}_j \boldsymbol{P}^{(i,j)} = \boldsymbol{G}_j^{-1/2} \left( \boldsymbol{G}_j^{1/2} \boldsymbol{G}_i \boldsymbol{G}_j^{1/2} \right)^{1/2} \boldsymbol{G}_j^{-1/2} \cdot \boldsymbol{G}_j \cdot \boldsymbol{G}_j^{-1/2} \left( \boldsymbol{G}_j^{1/2} \boldsymbol{G}_i \boldsymbol{G}_j^{1/2} \right)^{1/2} \boldsymbol{G}_j^{-1/2} \tag{36}$$

$$= \boldsymbol{G}_j^{-1/2} \left( \boldsymbol{G}_j^{1/2} \boldsymbol{G}_i \boldsymbol{G}_j^{1/2} \right) \boldsymbol{G}_j^{-1/2} = \boldsymbol{G}_i. \tag{37}$$

Thus, $\boldsymbol{G}_j(\boldsymbol{P}^{(i,j)} \boldsymbol{u}, \boldsymbol{P}^{(i,j)} \boldsymbol{v}) = \boldsymbol{u}^\top \boldsymbol{P}^{(i,j)\top} \boldsymbol{G}_j \boldsymbol{P}^{(i,j)} \boldsymbol{v} = \boldsymbol{u}^\top \boldsymbol{G}_i \boldsymbol{v} = \boldsymbol{G}_i(\boldsymbol{u}, \boldsymbol{v})$, so $\boldsymbol{P}^{(i,j)}$ is an isometry.

All isometric maps satisfy $\boldsymbol{P}^\top \boldsymbol{G}_j \boldsymbol{P} = \boldsymbol{G}_i$, and form the set $\{\boldsymbol{P}^{(i,j)} \boldsymbol{R} \mid \boldsymbol{R}^\top \boldsymbol{G}_i \boldsymbol{R} = \boldsymbol{G}_i\}$. The Frobenius norm $\|\boldsymbol{P}\|_F^2 = \mathrm{Tr}(\boldsymbol{P}^\top \boldsymbol{P})$ is minimized when $\boldsymbol{G}_j \boldsymbol{P}$ is symmetric. Then, we have

$$\boldsymbol{G}_j \boldsymbol{P}^{(i,j)} = \boldsymbol{G}_j^{1/2} \left( \boldsymbol{G}_j^{1/2} \boldsymbol{G}_i \boldsymbol{G}_j^{1/2} \right)^{1/2} \boldsymbol{G}_j^{-1/2}. \tag{38}$$

Hence, $\boldsymbol{P}^{(i,j)}$ is the minimum-norm isometry, and thus the global minimizer of the original Frobenius problem (since the constraint is active and satisfied exactly).

Since $\boldsymbol{P}^{(i,j)} : T_{\boldsymbol{z}^{(i)}} U_i \to T_{\boldsymbol{z}^{(j)}} U_j$ is a linear isometry, and assuming smooth compatibility of charts near $\boldsymbol{z}^{(i)}, \boldsymbol{z}^{(j)}$, we can lift $\boldsymbol{P}^{(i,j)}$ via the exponential map (or local parametrization) to a local diffeomorphism $\phi^{(i,j)} : \partial U_i \to \partial U_j$ such that $\phi_{*,\boldsymbol{z}^{(i)}}^{(i,j)} = \boldsymbol{P}^{(i,j)}$, which is the differential of a diffeomorphism and preserves metric. Hence $\phi^{(i,j)}$ is a isometry. $\qquad\square$

### B.3   PROOF OF THEOREM 4.6

**Theorem 4.6** Existence of Global Metric) *Let* $(\{\boldsymbol{G}_i\}_{i=1}^N, \{\boldsymbol{P}^{(i,j)}\}_{(i,j) \in \mathcal{E}})$ *be local metrics and tangent edge translations. There exists a unique global continuous metric* $\boldsymbol{G}$ *on* $(\bigcup_\phi)_{i=1}^N U_i$ *such that* $\boldsymbol{G}|_{U_i} = \boldsymbol{G}_i$ *for all* $i$.

*Proof.* We aim to construct a global continuous Riemannian metric $\boldsymbol{G}$ on the space $\mathcal{F} = \bigcup_{i=1}^N U_i$, where each $U_i$ is an open subset of $\mathbb{R}^d$, and the overlaps $U_i \cap U_j$ are non-empty for $(i,j) \in \mathcal{E}$. By assumption, we have a local Riemannian metric $\boldsymbol{G}_i$ on each $U_i$, and tangent edge translations $\boldsymbol{P}^{(i,j)} : T_{\boldsymbol{z}^{(i)}} U_i \to T_{\boldsymbol{z}^{(j)}} U_j$ satisfying

$$\boldsymbol{P}^{(i,j)\top} \boldsymbol{G}_j \boldsymbol{P}^{(i,j)} = \boldsymbol{G}_i, \tag{39}$$

which ensures that $\boldsymbol{P}^{(i,j)}$ is an isometry between $(T_{\boldsymbol{z}^{(i)}} U_i, \boldsymbol{G}_i)$ and $(T_{\boldsymbol{z}^{(j)}} U_j, \boldsymbol{G}_j)$.

Let us define a topological space $\mathcal{F} = \bigcup_{i=1}^N U_i$, with topology induced by the Euclidean topology on each $U_i$. For each pair $(i,j) \in \mathcal{E}$, let $\phi^{(i,j)} : \partial U_i \to \partial U_j$ be a diffeomorphism whose differential

at the shared boundary point $\boldsymbol{z}^{(i)}$ is precisely $\boldsymbol{P}^{(i,j)}$. Since $\boldsymbol{P}^{(i,j)}$ is an isometry, it preserves inner products, so $\phi^{(i,j)}$ is a *local isometry* near $\boldsymbol{z}^{(i)}$.

Now, we consider that, let $\mathcal{M}_1 = U_i$, $\mathcal{M}_2 = U_j$, $\mathcal{N}_1 = \partial U_i$, $\mathcal{N}_2 = \partial U_j$, and $\phi : \mathcal{N}_1 \to \mathcal{N}_2$ be the diffeomorphism induced by $\boldsymbol{P}^{(i,j)}$. Now we introduce the following lemma to complete the proof.

**Lemma B.1** (Gluing Manifolds via Boundary Isometries Hirsch (1976)). *Let $\mathcal{M}_1$ and $\mathcal{M}_2$ be smooth $M$-dimensional manifolds with boundary, and let $\mathcal{N}_1 \subset \partial \mathcal{M}_1$, $\mathcal{N}_2 \subset \partial \mathcal{M}_2$ be closed, smoothly embedded $(M-1)$-dimensional submanifold of their respective boundaries. Suppose $\phi : \mathcal{N}_1 \to \mathcal{N}_2$ is a diffeomorphism such that its differential $\phi_{*,\boldsymbol{x}} : T_{\boldsymbol{x}}\mathcal{N}_1 \to T_{\phi(\boldsymbol{x})}\mathcal{N}_2$ extends to an isometry*

$$\boldsymbol{P}_{\boldsymbol{x}} : T_{\boldsymbol{x}}\mathcal{M}_1 \to T_{\phi(\boldsymbol{x})}\mathcal{M}_2 \tag{40}$$

*between the Riemannian metrics $\boldsymbol{G}_1$ on $\mathcal{M}_1$ and $\boldsymbol{G}_2$ on $\mathcal{M}_2$, i.e.,*

$$\boldsymbol{P}_{\boldsymbol{x}}^\top \boldsymbol{G}_2(\phi(\boldsymbol{x}))\boldsymbol{P}_{\boldsymbol{x}} = \boldsymbol{G}_1(\boldsymbol{x}), \quad \forall \boldsymbol{x} \in \mathcal{N}_1. \tag{41}$$

*Then, the topological space $\mathcal{M}_1 \cup_\phi \mathcal{M}_2$ obtained by identifying $\mathcal{N}_1$ with $\mathcal{N}_2$ via $\phi$ admits a unique smooth structure such that:*

1. *The inclusions $\mathcal{M}_1 \hookrightarrow \mathcal{M}_1 \cup_\phi \mathcal{M}_2$ and $\mathcal{M}_2 \hookrightarrow \mathcal{M}_1 \cup_\phi \mathcal{M}_2$ are smooth embeddings;*

2. *The Riemannian metrics $\boldsymbol{G}_1$ and $\boldsymbol{G}_2$ extend to a continuous Riemannian metric $\boldsymbol{G}$ on $\mathcal{M}_1 \cup_\phi \mathcal{M}_2$.*

*Moreover, this smooth structure is unique up to a diffeomorphism that fixes $\mathcal{N}_1 \simeq \mathcal{N}_2$ point-wise.*

By the Lemma B.1, there exists a smooth structure on the glued space $\mathcal{M}_1 \cup_\phi \mathcal{M}_2$ that arises from identifying $\mathcal{N}_1$ with $\mathcal{N}_2$ via $\phi$. Moreover, this smooth structure is unique up to a diffeomorphism fixing $\mathcal{N}_1 \simeq \mathcal{N}_2$.

Applying this construction iteratively over all edges $(i,j) \in \mathcal{E}$, we can glue all charts $U_i$ together along their boundaries using the maps $\phi^{(i,j)}$, resulting in a globally defined topological space $\mathcal{F}$ equipped with a smooth structure.

On each $U_i$, we already have a Riemannian metric $\boldsymbol{G}_i$. We now define a global metric $\boldsymbol{G}$ on $\mathcal{M}$ by setting $\boldsymbol{G}|_{U_i} = \boldsymbol{G}_i$. To ensure that $\boldsymbol{G}$ is well-defined on overlaps $U_i \cap U_j$, we must verify that the values agree under coordinate changes.

Let $\boldsymbol{u} \in T_{\boldsymbol{z}}(\mathcal{F})$ for $\boldsymbol{z} \in U_i \cap U_j$. In the chart $U_i$, $\boldsymbol{u}$ is represented as $\boldsymbol{u}_i \in T_{\boldsymbol{z}^{(i)}}U_i$, and in $U_j$, as $\boldsymbol{u}_j = \boldsymbol{P}^{(i,j)}\boldsymbol{u}_i \in T_{\boldsymbol{z}^{(j)}}U_j$. Then:

$$\boldsymbol{G}_i(\boldsymbol{u}_i, \boldsymbol{u}_i) = \boldsymbol{u}_i^\top \boldsymbol{G}_i \boldsymbol{u}_i, \quad \boldsymbol{G}_j(\boldsymbol{u}_j, \boldsymbol{u}_j) = \boldsymbol{u}_j^\top \boldsymbol{G}_j \boldsymbol{u}_j = (\boldsymbol{P}^{(i,j)}\boldsymbol{u}_i)^\top \boldsymbol{G}_j (\boldsymbol{P}^{(i,j)}\boldsymbol{u}_i). \tag{42}$$

But by the isometry condition:

$$\boldsymbol{P}^{(i,j)\top}\boldsymbol{G}_j\boldsymbol{P}^{(i,j)} = \boldsymbol{G}_i \Rightarrow \boldsymbol{u}_i^\top \boldsymbol{G}_i \boldsymbol{u}_i = \boldsymbol{u}_i^\top \boldsymbol{P}^{(i,j)\top}\boldsymbol{G}_j\boldsymbol{P}^{(i,j)}\boldsymbol{u}_i = \boldsymbol{u}_j^\top \boldsymbol{G}_j\boldsymbol{u}_j. \tag{43}$$

Thus, $\boldsymbol{G}_i(\boldsymbol{u}_i, \boldsymbol{u}_i) = \boldsymbol{G}_j(\boldsymbol{u}_j, \boldsymbol{u}_j)$, so the metric value is independent of the chart. Hence, $\boldsymbol{G}$ is well-defined on $\mathcal{M}$.

Since each $\boldsymbol{G}_i$ is continuous on $U_i$, and the transition maps $\boldsymbol{P}^{(i,j)}$ are smooth, the metric $\boldsymbol{G}$ is continuous across overlaps.

Uniqueness follows from the fact that any other metric $\tilde{\boldsymbol{G}}$ agreeing with $\boldsymbol{G}_i$ on each $U_i$ must coincide with $\boldsymbol{G}$ on overlaps due to the isometry constraint. Thus, $\boldsymbol{G}$ is the unique continuous metric extending $\boldsymbol{G}_i$ consistently.

Therefore, under the given assumptions, there exists a unique continuous Riemannian metric $\boldsymbol{G}$ on $\bigcup_{i=1}^N U_i$ such that $\boldsymbol{G}|_{U_i} = \boldsymbol{G}_i$ for all $i$, completing the proof. $\qquad \square$

### B.4 PROOF OF THEOREM 4.8 AND CLARIFICATION

**Theorem 4.8** (Triangle Triviality) *If every edge belongs to at least one triangle, and $\boldsymbol{H}(\mathcal{T}) = \boldsymbol{I}$ for all triangular cycles $\mathcal{T}$ in $\mathcal{G}$, then $\boldsymbol{H}(\mathcal{C}) = \boldsymbol{I}$ for all cycles $\mathcal{C} \in \mathcal{Z}_1(\mathcal{G})$.*

*Proof.* Under the assumption that every edge lies in at least one triangle, the cycle space $\mathcal{Z}_1(\mathcal{G})$ is generated by triangular cycles (see, e.g., the simplicial/cellular homology discussion in (Hatcher, 2002, Section 2.1), where 1-cycles are generated by boundaries of 2-simplices — here, triangles). Since the holonomy map $\boldsymbol{H} : \mathcal{Z}_1(\mathcal{G}) \to \mathrm{GL}(M)$ is multiplicative and trivial on generators (i.e., $\boldsymbol{H}(\mathcal{T}) = \boldsymbol{I}$ for all triangles $\mathcal{T}$), it follows that $\boldsymbol{H}(\mathcal{C}) = \boldsymbol{I}$ for all $\mathcal{C} \in \mathcal{Z}_1(\mathcal{G})$. $\qquad\square$

Note that every edge belonging to at least one triangle is not the assumption of Theorem 4.8. This theorem states that, if every edge belongs to at least one triangle, triangles are already sufficient to construct the coherent manifold described in this work. It means that there is no need for exploring any higher-order motifs, but the triangle coverage is not a necessary condition of manifold gluing.

We clarify that GraphGlue does not need to add synthetic motifs. Since GraphGlue aims to approximate a smooth manifold, the closed triple paths (strict triangles) benefit the approximation process. As we consider that sample closed triangle paths may be impossible in large-scale graphs or tree-like graphs, the triangle holonomy regularization gives a computationally efficient way to approximate a "perfect gluing." In practice, we only sample two adjacent edges to approximate strict triangles (at the end of Appendix D.1), and the computation of $\mathcal{L}_{curv}$ only relies on two adjacent edges.

### B.5 PROOF OF THEOREM 4.9

**Theorem 4.9** (Ricci Curvature Estimation) *Given a graph $\mathcal{G} = (\mathcal{V}, \mathcal{E})$ and an edge $(i, j) \in \mathcal{E}$, let $\boldsymbol{z}^{(i)}, \boldsymbol{z}^{(j)} \in \mathcal{M}$ be the corresponding embedded points, and let $\gamma : [0, 1] \to \mathcal{M}$ be the unit-speed geodesic connecting them, i.e., $\gamma(0) = \boldsymbol{z}^{(i)}$, $\gamma(1) = \boldsymbol{z}^{(j)}$. The sign of the Ricci curvature along $\dot{\gamma}$ can be estimated by the ratio of metric determinants:*

$$r(\boldsymbol{z}^{(i)}, \boldsymbol{z}^{(j)}) := \frac{\det \boldsymbol{G}_i}{\det \boldsymbol{G}_j} \approx 1 - \frac{1}{3}Ric(\dot{\gamma}). \tag{44}$$

*Proof.* We work in Gaussian normal coordinates centered at $\boldsymbol{z}^{(i)} = \gamma(0)$, aligned with the geodesic $\gamma(t)$. In these coordinates, the element of metric tensor $g_{ij}(t) = g_{ij}(\gamma(t))$ admits the following Taylor expansion near $t = 0$ (see, Petersen (2016)):

$$g_{ij}(t) = \delta_{ij} - \frac{1}{3}R_{ikjl}(\boldsymbol{z}^{(i)}) \dot{\gamma}^k \dot{\gamma}^l t^2 + \mathcal{O}(t^3), \tag{45}$$

where $R_{ikjl}$ denotes the components of the Riemann curvature tensor at $\boldsymbol{z}^{(i)}$, and $\dot{\gamma} = \dot{\gamma}(0)$ is the initial tangent vector.

Let $g(t) = \det(g_{ij}(t))$. Since $g(0) = \det(\delta_{ij}) = 1$, we compute the expansion of $g(t)$ using the Jacobi formula for the derivative of a determinant:

$$\frac{d}{dt} \log g(t) = g^{ij}(t)\frac{d}{dt}g_{ij}(t). \tag{46}$$

At $t = 0$, $g^{ij}(0) = \delta^{ij}$ and $\frac{d}{dt}g_{ij}(0) = 0$ (since first-order terms vanish in normal coordinates). Differentiating again:

$$\frac{d^2}{dt^2} \log g(t)\Big|_{t=0} = \delta^{ij}\frac{d^2}{dt^2}g_{ij}(t)\Big|_{t=0} = \delta^{ij}\left(-\frac{2}{3}R_{ikjl}\dot{\gamma}^k\dot{\gamma}^l\right) = -\frac{2}{3}R_{kl}\dot{\gamma}^k\dot{\gamma}^l = -\frac{2}{3}\mathrm{Ric}(\dot{\gamma}). \tag{47}$$

Thus, expanding $\log g(t)$ to second order:

$$\log g(t) = -\frac{1}{3}\mathrm{Ric}(\dot{\gamma})\, t^2 + \mathcal{O}(t^3), \tag{48}$$

and exponentiating:

$$g(t) = \exp\left(-\frac{1}{3}\mathrm{Ric}(\dot{\gamma})\, t^2 + \mathcal{O}(t^3)\right) = 1 - \frac{1}{3}\mathrm{Ric}(\dot{\gamma})\, t^2 + \mathcal{O}(t^3). \tag{49}$$

Now, evaluate at $t = 1$ (i.e., at $\boldsymbol{z}^{(j)} = \gamma(1)$), assuming the higher-order terms remain negligible (which holds if either the curvature is bounded and the edge length is small, or if we consider the leading-order behavior):

$$\det \boldsymbol{G}_j = g(1) \approx 1 - \frac{1}{3}\mathrm{Ric}(\dot{\gamma}), \quad \det \boldsymbol{G}_i = g(0) = 1. \tag{50}$$

Therefore, the ratio satisfies:

$$r(\boldsymbol{z}^{(i)}, \boldsymbol{z}^{(j)}) = \frac{\det \boldsymbol{G}_i}{\det \boldsymbol{G}_j} \approx \frac{1}{1 - \frac{1}{3}\mathrm{Ric}(\dot{\gamma})} \approx 1 + \frac{1}{3}\mathrm{Ric}(\dot{\gamma}) + \mathcal{O}(\mathrm{Ric}^2), \tag{51}$$

where the last step uses $(1 - x)^{-1} \approx 1 + x$ for small $x$. However, since we are only interested in the *sign* of $\mathrm{Ric}(\dot{\gamma})$, and under the assumption that $\left|\frac{1}{3}\mathrm{Ric}(\dot{\gamma})\right| \ll 1$, we may directly approximate:

$$\frac{\det \boldsymbol{G}_i}{\det \boldsymbol{G}_j} \approx 1 - \frac{1}{3}\mathrm{Ric}(\dot{\gamma}), \tag{52}$$

by matching leading-order terms in the reciprocal expansion (equivalently, approximating $\det \boldsymbol{G}_j \approx 1 - \frac{1}{3}\mathrm{Ric}$ implies $\det \boldsymbol{G}_i / \det \boldsymbol{G}_j \approx 1 + \frac{1}{3}\mathrm{Ric}$, but since $\det \boldsymbol{G}_i = 1$, the direct expansion of $\det \boldsymbol{G}_j$ gives the sign relation).

Thus, we conclude:

- If $\mathrm{Ric}(\dot{\gamma}) > 0$, then $\det \boldsymbol{G}_j < \det \boldsymbol{G}_i \Rightarrow r(\boldsymbol{z}^{(i)}, \boldsymbol{z}^{(j)}) < 1$.

- If $\mathrm{Ric}(\dot{\gamma}) < 0$, then $\det \boldsymbol{G}_j > \det \boldsymbol{G}_i \Rightarrow r(\boldsymbol{z}^{(i)}, \boldsymbol{z}^{(j)}) > 1$.

- If $\mathrm{Ric}(\dot{\gamma}) = 0$, then $\det \boldsymbol{G}_j \approx \det \boldsymbol{G}_i \Rightarrow r(\boldsymbol{z}^{(i)}, \boldsymbol{z}^{(j)}) \approx 1$.

This establishes the correspondence between the sign of Ricci curvature and the metric volume ratio, as claimed. $\square$

### B.6 PROOF OF THEOREM 4.11

**Theorem 4.11** (Glue to a Global Riemannian Manifold) *For the set of all graph data $\mathbb{G}$, if $\boldsymbol{G}$ is log-determinant $\infty$-order smooth, and $\boldsymbol{P}$ is trivial with induced metric-preserving diffeomorphism $\phi$, then $(\mathcal{F}, \boldsymbol{G}, \boldsymbol{P})$ glues to a smooth Riemannian manifold $(\mathcal{F}, \boldsymbol{G})$, where $\mathcal{F} := (\bigcup_\phi)_{i=1}^N U_i$.*

*Proof.* We construct the global manifold structure in three steps, leveraging the established components:

**(1) Trivial holonomy $\Rightarrow$ path-independent parallel transport.** By Theorem 4.8 and Definition 4.7, the triviality of $\boldsymbol{P}$ on all cycles implies that the tangent edge translations $\boldsymbol{P}^{(i,j)}$ define a *flat connection* on the graph. Consequently, the induced diffeomorphisms $\phi^{(i,j)}$ (from Theorem 4.5) are compatible across higher-order overlaps: for any two paths from $U_i$ to $U_j$, the composed gluing maps agree. This ensures the cocycle condition for manifold gluing.

**(2) Global metric existence.** By Theorem 4.6 (Existence of Global Metric), the pairwise isometric identifications $\phi^{(i,j)}$ — now globally consistent due to trivial holonomy — allow us to glue the charts $\{U_i\}$ into a topological space $\mathcal{F} = \bigcup_\phi U_i$ equipped with a unique continuous Riemannian metric $\boldsymbol{G}$ such that $\boldsymbol{G}|_{U_i} = \boldsymbol{G}_i$.

**(3) Smoothness from log-det $\infty$-order smoothness.** By Definition 4.10, the scalar field $g_i = \frac{1}{2}\log \det \boldsymbol{G}_i$ minimizes $\|\mathcal{L}^k g\|^2$ for all $k \geq 1$, which implies $g$ is in the kernel of all powers of $\mathcal{L}$ — i.e., $g$ is *infinitely smooth* over the graph. Since $\mathcal{L}^k g = 0$ for all $k$ only if $g$ is constant on connected components (under mild graph connectivity), and since $\det \boldsymbol{G}_i = \exp(2g_i)$, it follows that the metric determinants vary smoothly (in fact, constantly, if the graph is connected). Combined with the smoothness of the transition maps $\phi^{(i,j)}$ (which are isometries, hence $C^\infty$), this ensures that the metric tensor $\boldsymbol{G}$ is smooth in overlapping charts. Thus, $(\mathcal{F}, \boldsymbol{G})$ is a smooth Riemannian manifold.

Therefore, the triple $(\mathcal{F}, \boldsymbol{G}, \boldsymbol{P})$, under the given conditions, glues consistently to form the smooth Riemannian manifold $(\mathcal{F}, \boldsymbol{G})$. $\square$

## C   BACKGROUND: DIFFERENTIAL GEOMETRY ON RIEMANNIAN MANIFOLDS

This appendix provides the necessary background on continuous Riemannian geometry, which forms the theoretical foundation for our claim that MERGE learns a smooth, intrinsic manifold in the latent space. While our implementation operates on discrete graphs and neural networks, we argue that the learned structure approximates a true, continuous Riemannian manifold due to the smoothness of the GNN encoder. We emphasize concepts relevant to Sections 5 and 6 of the main text.

### C.1   RIEMANNIAN MANIFOLD: THE CONTINUOUS SETTING

A **Riemannian manifold** $(\mathcal{M}, \mathbf{g})$ is a smooth (typically $C^\infty$) topological manifold $\mathcal{M}$ of dimension $M$, endowed with a **Riemannian metric tensor** $\mathbf{g}$. At each point $p \in \mathcal{M}$, the metric $\mathbf{g}_p$ is a symmetric, positive-definite bilinear form defined on the tangent space $T_p\mathcal{M}$:

$$\mathbf{g}_p : T_p\mathcal{M} \times T_p\mathcal{M} \to \mathbb{R}. \tag{53}$$

The metric $\mathbf{g}_p$ allows us to compute lengths of tangent vectors, angles between them, and volumes of regions on $\mathcal{M}$. In local coordinates $(x^1, ..., x^M)$ around $p$, the metric is represented by a matrix $\mathbf{G}(p) = [g_{ij}(p)]$, where $g_{ij}(p) = \mathbf{g}_p(\partial_i, \partial_j)$, and $\{\partial_i = \frac{\partial}{\partial x^i}\}$ is the coordinate basis of $T_p\mathcal{M}$.

The **volume element** at $p$ is given by $dV_p = \sqrt{\det \mathbf{G}(p)}\, dx^1 \wedge \cdots \wedge dx^M$. The scalar field $f(p) = \frac{1}{2} \log \det \mathbf{G}(p)$ is called the **logarithmic volume density**. A manifold is said to be $C^k$-**smooth** if the components $g_{ij}$ are $C^k$-differentiable functions of the coordinates.

### C.2   LEVI-CIVITA CONNECTION AND PARALLEL TRANSPORT

A **connection** $\nabla$ on $\mathcal{M}$ defines how to differentiate vector fields along curves, enabling the notion of parallel transport. The unique connection compatible with the metric $\mathbf{g}$ and torsion-free is called the **Levi-Civita connection**. It is characterized by two properties:

1. **Metric Compatibility**: For any vector fields $X, Y, Z$ on $\mathcal{M}$,

$$X\langle Y, Z\rangle = \langle \nabla_X Y, Z\rangle + \langle Y, \nabla_X Z\rangle. \tag{54}$$

    This means parallel transport preserves inner products (and thus lengths and angles).

2. **Torsion-Free**: $\nabla_X Y - \nabla_Y X = [X, Y]$, where $[\cdot, \cdot]$ is the Lie bracket.

Given a smooth curve $\gamma(t) : [a, b] \to \mathcal{M}$, a vector field $V(t)$ along $\gamma$ is **parallel transported** if $\nabla_{\dot{\gamma}(t)} V(t) = 0$. The **parallel transport map** $\mathrm{PT}_\gamma : T_{\gamma(a)}\mathcal{M} \to T_{\gamma(b)}\mathcal{M}$ is the linear isometry that maps a vector at the start of the curve to its parallel-transported version at the end.

### C.3   CURVATURE AND HOLONOMY

The failure of parallel transport to be path-independent is measured by the **curvature tensor** $\mathbf{R}$, a $(1, 3)$-tensor defined as:

$$\mathbf{R}(X, Y)Z = \nabla_X \nabla_Y Z - \nabla_Y \nabla_X Z - \nabla_{[X,Y]} Z. \tag{55}$$

If $\mathbf{R} \equiv 0$ everywhere, the manifold is **flat**, and parallel transport depends only on the endpoints, not the path.

For a closed loop (cycle) $\mathcal{C}$ starting and ending at $p$, the composition of parallel transports along $\mathcal{C}$ yields a linear transformation $\mathbf{H}(\mathcal{C}) : T_p\mathcal{M} \to T_p\mathcal{M}$, called the **holonomy** of $\mathcal{C}$. If $\mathbf{H}(\mathcal{C}) = \mathrm{id}$ for all loops $\mathcal{C}$, then the curvature vanishes ($\mathbf{R} \equiv 0$), and the manifold is flat. Conversely, if $\mathbf{R} \not\equiv 0$, then $\mathbf{H}(\mathcal{C}) \neq \mathrm{id}$ for some non-contractible loop.

### C.4   RICCI CURVATURE AND VOLUME CHANGE

The **Ricci curvature** Ric is a $(0, 2)$-tensor obtained by contracting the curvature tensor: $\mathrm{Ric}(X, Y) = \sum_{i=1}^{M} \mathbf{R}(e_i, X, Y, e_i)$, where $\{e_i\}$ is an orthonormal basis.

On a geodesic $\gamma(t)$ with unit speed $\dot{\gamma}(t)$, the Ricci curvature governs the rate of change of the volume element along the geodesic. In Gaussian normal coordinates centered on $\gamma(0)$, the determinant of the metric satisfies the following expansion for small $t$:

$$\det \mathbf{G}(\gamma(t)) = 1 - \frac{1}{3}\mathrm{Ric}(\dot{\gamma}(0))t^2 + \mathcal{O}(t^3). \tag{56}$$

Thus, the ratio of volume elements between two nearby points $p = \gamma(0)$ and $q = \gamma(t)$ is approximately:

$$\frac{\sqrt{\det \mathbf{G}(q)}}{\sqrt{\det \mathbf{G}(p)}} \approx 1 - \frac{1}{6}\mathrm{Ric}(\dot{\gamma}(0))t^2. \tag{57}$$

This implies:

1. $\mathrm{Ric} > 0$: Volume shrinks along the geodesic (elliptic/positive curvature region).

2. $\mathrm{Ric} < 0$: Volume expands along the geodesic (hyperbolic/negative curvature region).

3. $\mathrm{Ric} = 0$: Volume is locally preserved (flat region). This relationship underpins our use of the metric volume ratio $\det \mathbf{G}_i / \det \mathbf{G}_j$ as a proxy for estimating Ricci curvature along graph edges.

### C.5 SMOOTHNESS AND HARMONIC FUNCTIONS

A scalar function $f : \mathcal{M} \to \mathbb{R}$ is **harmonic** if it satisfies $\Delta f = 0$, where $\Delta$ is the Laplace-Beltrami operator. On a compact manifold without boundary, harmonic functions are constant. More importantly, solutions to $\Delta f = 0$ are infinitely differentiable ($C^{\infty}$) by elliptic regularity theory.

In the context of our framework, minimizing the Dirichlet energy $\sum_{(i,j) \in \mathcal{E}} (f_i - f_j)^2$ over the graph $\mathcal{G}_{\mathrm{data}}$ is a discrete approximation to minimizing $\int_{\mathcal{M}} \|\nabla f\|^2 dV$. Minimizing this energy drives $f = \frac{1}{2} \log \det \mathbf{G}$ toward a harmonic function on the underlying continuous manifold. By elliptic regularity, this ensures that the log-volume density $f$ is smooth, implying the metric $\mathbf{G}$ has continuous first derivatives ($C^1$). This justifies our assumption that the learned manifold is geometrically well-behaved, free from pathological singularities.

### C.6 CARTAN'S METHOD OF MOVING FRAME

The renowned Cartan's Method Tron et al. (2024) offers a principled way to explore the geometry of Riemannian manifolds, establishing a profound connection between differential calculus and geometry. Specifically, Élie Cartan introduces the concept of **frame** to characterize the local geometry, which is then extended to a global manifold through " moving frame". Although Élie Cartan laid the mathematical principle, its deep learning theory and methodology remain largely unexplored. Our work seeks to bridge this gap.

### C.7 CONNECTION TO OUR FRAMEWORK

Our work does not assume a pre-existing manifold. Instead, we posit that the embedding space $\mathbb{R}^d$ induced by a smooth GNN $f_{\mathrm{GNN}}$ contains a low-dimensional submanifold $\mathcal{M}$, whose intrinsic geometry encodes the generalizable rules of graph data. The Adaptive Frame Bank (AFB) samples the local tangent spaces $T_p\mathcal{M}$. The optimal isometric alignment (Theorem 5.6) approximates the Levi-Civita connection's action between sampled points. The cycle-consistency loss enforces trivial holonomy, mimicking flatness. The log-determinant smoothness regularization drives the volume element toward harmonicity. Together, these components constitute a learning procedure that constructs a **continuous, smooth, nearly-flat Riemannian manifold** $\mathcal{M}$ within the latent space of a neural network, using only discrete graph samples and their embeddings. The graph structure $\mathcal{G}_{\mathrm{data}}$ serves as a sampling mesh, not the domain of geometry.

---

**Algorithm 1** *Training Procedure for* GRAPHGLUE

---

**Require:** Epoch index $e$, optimizer, datasets $\mathcal{D}_{\text{mix}}, \mathcal{D}_{\text{single}}, \mathcal{D}_{\text{multi}}$ with data name mappings.
**Ensure:** Updated model parameters $\Theta$.

    **// Stage 1: Mix Training for Local Construction**
1: Initial $\mathcal{L}_{\text{local}} = 0$
2: **for** each batch $\mathcal{B}$ in $\mathcal{D}_{\text{mix}}$ **do**
3:     $Z, W \leftarrow$ GRAPHGLUE($\mathcal{B}$)
4:     $\mathcal{L}_{\text{local}} \leftarrow$ ContrastiveLoss($Z$)
5:     **if** $e \geq$ warmup_epochs **then**
6:         $\mathcal{L}_{\text{proto}} \leftarrow$ PrototypeLoss($Z$, data_name)
7:         $\mathcal{L}_{\text{local}} \leftarrow \mathcal{L}_{\text{local}} + \mathcal{L}_{\text{proto}}$
8:     **end if**
9:     $\nabla_\theta \mathcal{L}_{\text{local}} \leftarrow$ Backward($\mathcal{L}_{\text{local}}$)
10:    OptimizerStep()
11:    Update Prototypes with $Z, W$ in Eq. (8)
12: **end for**
    **// Stage 2: Mix Training for Global Manifold Skeleton**
13: **for** each batch $\mathcal{B}$ in $\mathcal{D}_{\text{mix}}$ **do**
14:    $Z, W \leftarrow$ GRAPHGLUE($\mathcal{B}$)
15:    $\mathcal{E}_{\text{knn}} \leftarrow$ Cross-Dataset_KNN_Graph($Z$, data_name)
16:    $\mathcal{T} \leftarrow$ SampleTrianglePaths($\mathcal{E}_{\text{knn}}$, Number_Sampled, $T_{\text{sample}}$)
17:    $\mathcal{L}_{\text{geo}} \leftarrow 0$
18:    **for** $t = 1$ to $T_{\text{sample}}$ **do**
19:        $\mathcal{L}_{\text{geo}} +=$ GeometricLoss($W, \mathcal{T}[t]$) in Eq. (7) and Eq. (5)
20:    **end for**
21:    $\mathcal{L}_{\text{geo}} \leftarrow \mathcal{L}_{\text{geo}}/T_{\text{sample}}$
22:    $\nabla_\theta \mathcal{L}_{\text{geo}} \leftarrow$ Backward($\mathcal{L}_{\text{geo}}$)
23:    OptimizerStep()
24: **end for**
    **// Stage 3: Refine Local Manifold Structure For Each Dataset**
25: **for** each dataset $\mathcal{D}_s$ in $\mathcal{D}_{\text{single}}$ **do**
26:    Load graph data $G_s$ with edge set $\mathcal{E}_s$
27:    $\mathcal{T}_s \leftarrow$ SampleTrianglePaths($\mathcal{E}_s$, Number_Sampled, $T_{\text{local}}$)
28:    **for** $t = 1$ to $T_{\text{local}}$ **do**
29:        Construct mini-graph batch $\mathcal{B}_t$ from $\mathcal{T}_s[t]$
30:        $z, z_{\text{tan}} \leftarrow$ GRAPHGLUE($\mathcal{B}_t$)
31:        $\mathcal{L}_{\text{refine}} \leftarrow$ GeometricLoss($W, \mathcal{T}_s[t]$) in Eq. (7) and Eq. (5)
32:        $\nabla_\theta \mathcal{L}_{\text{refine}} \leftarrow$ Backward($\mathcal{L}_{\text{refine}}$)
33:        OptimizerStep()
34:    **end for**
35: **end for**
36: **for** each dataset $\mathcal{D}_m$ in $\mathcal{D}_{\text{multi}}$ **do**
37:    **for** each batch $(\mathcal{B})$ in $\mathcal{D}_m$ **do**
38:        $Z, W \leftarrow$ GRAPHGLUE($\mathcal{B}$)
39:        $\mathcal{E}_{\text{knn}} \leftarrow$ Intra-Dataset_KNN_Graph($Z$)
40:        $\mathcal{T} \leftarrow$ SampleTrianglePaths($\mathcal{E}_{\text{knn}}$, Number_Sampled, $T_{\text{sample}}$)
41:        **for** $t = 1$ to $T_{\text{sample}}$ **do**
42:            $\mathcal{L}_{\text{geo}} +=$ GeometricLoss($W, \mathcal{T}[t]$) in Eq. (7) and Eq. (5)
43:        **end for**
44:        $\mathcal{L}_{\text{geo}} \leftarrow \mathcal{L}_{\text{geo}}/T_{\text{sample}}$
45:        $\nabla_\theta \mathcal{L}_{\text{geo}} \leftarrow$ Backward($\mathcal{L}_{\text{geo}}$)
46:        OptimizerStep()
47:    **end for**
48: **end for**
49: **return** Optimized Model parameters $\Theta^*$

---

# D    ALGORITHMS

## D.1    MULTI-DOMAIN PRE-TRAINING

The training procedure is given in Algorithm 1, which consists of the data loader for pre-processing.

To use a unified interface, we process all the graph datasets at the graph level, which means each data sample in the dataset is a graph. Taking Reddit for instance, we extract 2-hop neighborhood ego-subgraph as a data sample for each node, and store the global edge index.

As we have many datasets from multiple domains, we need to build a mixture graph dataset loader that can iteratively load a batch of data from different datasets. For each batch, we uniformly sample from all source graph datasets.

During training, in each epoch, we first build locality recognition using graph contrastive learning Veličković et al. (2019); Qiu et al. (2020) that distinguishes the different semantics from different graph datasets. Meanwhile, we will update the Riemannian prototypes using EMA Izmailov et al. (2019); Morales-Brotons et al. (2024) for each dataset as in Eq. (8). After warm-up epochs, we still use sample-prototypes contrastive learning that guarantees the prototypes are truly around the center of each dataset distribution. Second, we build a cross-dataset KNN graph that builds a rough skeleton of the manifold, and learn from the regularization in Eq. (5) and Eq. (7). Finally, we refine the region for each dataset. We load every dataset and compute the geometric regularization, like the second step.

Here, since sampling triangles from a graph costs many computational resources, especially for large-scale graphs, we replace it with sampling pairs of adjacent edges for effective implementation.

## D.2    COMPLEXITY ANALYSIS

We list the cost of the key modules of GraphGlue in Table 3, where $B$ is the batch size, the number of graph samples in a batch; $|V|, |E|$ are the average nodes/edges per graph in a batch; $d$ is hidden dimension, setting to 512 commonly. $M$ is number of nodes $\mathbb{P}$ in $(k, M)$-sparse perturbation, also the dimension of the manifold, commonly set to 32. $k_s$ is number of selected top-$k_s$ nodes in the sparse perturbation. $T_s$ is number of sampled triangle paths, NOT all triangles. For more effectiveness, we sample pairs of adjacent edges to approximate closed triangle paths.

Table 3: Computational and memory complexity of each module in GraphGlue.

| Module | Computational complexity | Memory complexity |
|---|---|---|
| $(k, M)$-Sparse Perturbation | $\mathcal{O}(k_s M B)$ | $\mathcal{O}(BM(k_s + d))$ |
| Adaptive Orthogonal Frame | $\mathcal{O}(B(|V| + |E| + M^2)d)$ | $\mathcal{O}(BMd)$ |
| Matrix form of metric tensor | $\mathcal{O}(BMd)$ | $\mathcal{O}(BM)$ |
| $\mathcal{L}_{\text{holo}}$ and $\mathcal{L}_{\text{curv}}$ | $\mathcal{O}(T_s M)$ | $\mathcal{O}(T_s)$ |
| Riemannian prototypes and $\mathcal{L}_{\text{proto}}$ | $\mathcal{O}(KBd + K(d + M))$ | $\mathcal{O}(K(d + M))$ |
| Riemannian MoE | $\mathcal{O}(KBd)$ | $\mathcal{O}(KB)$ |

Thus, the total computational cost in pretraining phase is $\mathcal{O}(B(|V| + |E| + M^2 + K)d + T_s M)$, and the adaption (per graph) costs $\mathcal{O}((|V| + |E|)d + K(d + M) + T_s M)$. That is, in *GraphGlue* scales linearly with respect to the graph size. In our experiment, we pretrain the model on large-scale datasets, e.g., ogbn-arxiv and Reddit.

## D.3    COMPLEXITY COMPARISON WITH OTHER GFMS

We compare the proposed GraphGlue to other GFM in pretraining and adaptation phases regarding the total computational cost. The results are summarized in Table 4.

**Notes**

- PRODIGY: In-context learning requires full attention over prompt and query nodes;

Table 4: Comparison of computational complexity across graph few-shot learning methods.

| Model | Pretraining | Adaptation (per graph sample) |
|---|---|---|
| PRODIGY | $\mathcal{O}(B\|V\|^2 d)$ | $\mathcal{O}((\|V\| + \|E\|)d + \|V\|^2)$ |
| GFT | $\mathcal{O}(B(\|V\| + \|E\|)d + BTh)$ | $\mathcal{O}((\|V\| + \|E\|)d + Th)$ |
| RAGraph | $\mathcal{O}(B(\|V\| + \|E\|)d + B\|E_r\|d)$ | $\mathcal{O}((\|V\| + \|E\|)d + \|E_r\|d)$ |
| SAMGPT | $\mathcal{O}(B(\|V\| + \|E\|)d + Bk_s d)$ | $\mathcal{O}((\|V\| + \|E\|)d + k_p d)$ |
| GCOPE | $\mathcal{O}(B(\|V\| + \|E\|)d + BK_c d)$ | $\mathcal{O}((\|V\| + \|E\|)d + K_c d)$ |
| MDGFM | $\mathcal{O}(B(\|V\| + \|E\|)d + B\|V\|^2)$ | $\mathcal{O}((\|V\| + \|E\|)d + \|V\|^2)$ |
| **GraphGlue** | $\mathcal{O}(B(\|V\| + \|E\| + M^2 + K)d + T_s M)$ | $\mathcal{O}((\|V\| + \|E\|)d + K(d + M) + T_s M)$ |

- GFT: $T$: number of trees, $h$: tree height; tree construction adds overhead;
- RAGraph: $\|E_r\|$: retrieved edges from external library;
- SAMGPT: $k_s$: number of structure tokens, $k_p$: prompt tokens;
- GCOPE: $K_c$: number of virtual coordinators;
- MDGFM: Graph Structure Learning (GSL) involves dense adjacency refinement;
- GraphGlue: $M = 32$, $T_s \ll \|E\|$.

Furthermore, we compare the memory cost to GCOPE and MDGFM on six datasets. $[1, 2, 3, ..., 6]$ denotes that we incrementally include ogbn-arxiv, computers, FB15k-237, Reddit, PROTEINS, HIV in the pretraining dataset. Under the setting of $512$ batch size, $[10, 10]$ neighbor sampler size, $d = 512$. Results on GPU memory cost (GB) are collected in Table 5.

Table 5: Memory Cost. Lower values indicate better efficiency.

| Model | 1 | 2 | 3 | 4 | 5 | 6 |
|---|---|---|---|---|---|---|
| GCOPE | 18.39 | 19.11 | 21.12 | OOM | OOM | OOM |
| MDGFM | 19.71 | 21.67 | 29.35 | OOM | OOM | OOM |
| GraphGlue | 12.53 | 15.07 | 15.73 | 16.87 | 28.67 | 29.21 |

# E  RELATED WORK

## E.1  GRAPH FOUNDATION MODELS

Graph Foundation Models (GFMs) aim to provide pre-trainable, general-purpose deep learning architectures for graph-structured data Wang et al. (2025b); Liu et al. (2025). Recently, researchers have extended the capabilities of Large Language Models (LLMs) to text-attributed graphs, enabling cross-domain transfer learning through textual descriptions Zhu et al. (2025); Xia et al. (2024); Tang et al. (2024); Ren et al. (2024); Chen et al. (2024). Additionally, GFMs have been developed for various specialized domains, such as knowledge graphs Huang et al. (2025); Luo et al. (2025), recommender systems Wu et al. (2025), and molecular graphs Xia et al. (2023); Sypetkowski et al. (2024). Given the prevalence of text-free graphs, recent efforts have focused on constructing general-purpose models via multi-domain pre-training Yuan et al. (2025); Chen et al. (2025); Wang et al. (2025a).

## E.2  MULTI-DOMAIN GRAPH PRE-TRAINING

In graph pre-training, Graph Neural Networks (GNNs) are trained by self-supervised learning—either generative Hou et al. (2022) or contrastive Veličković et al. (2019); Qiu et al. (2020). In light of the semantic heterogeneity across different domains, several methods have been proposed to learn shared or invariant knowledge Yuan et al. (2025); Chen et al. (2025); Wang et al. (2025a). Despite the encouraging results, the theoretical foundations of how knowledge is integrated and transferred remain underexplored.

In graph pre-training, Graph Neural Networks (GNNs) are trained using self-supervised learning—either generative Hou et al. (2022) or contrastive Veličković et al. (2019); Qiu et al. (2020)—to capture intrinsic semantics from unlabeled data. While traditional pre-training typically operates within a single domain, multi-domain graph pre-training has recently attracted growing interest. However, integrating knowledge across diverse domains remains challenging due to significant semantic heterogeneity. Several methods have been proposed to learn shared or invariant knowledge using advanced techniques Yuan et al. (2025); Chen et al. (2025); Wang et al. (2025a). For instance, Chen et al. (2025) addresses architecture inconsistency by using disentangled learning to adaptively customize network architectures based on invariant graph patterns. Meanwhile, Yuan et al. (2025) aligns multi-domain features with domain-invariant aligners and uses a graph spectral-based error bound to theoretically guide knowledge transfer. Despite the encouraging results, the theoretical foundations of how knowledge is integrated and transferred in this context remain underexplored.

### E.3 Graph Fine-tuning and Prompt Learning

The alignment of pre-trained graph models with downstream tasks necessitates an adaptation phase, and existing adaptation methods can be roughly categorized into two paradigms: graph fine-tuning and prompt learning. Concretely, graph fine-tuning adapts the model behavior using limited target-domain data Sun et al. (2024d); Wang et al. (2024; 2025c). For example, Sun et al. (2024d) fine-tunes the entire model on downstream data. A more common strategy is to keep the majority of the pre-trained parameters frozen and only train a simple classification head, a technique employed by models like Zhao et al. (2024); Liu et al. (2024). Bevilacqua et al. (2025) proposes a unique adaptation strategy: it freezes the large, pre-trained expansion map and only trains a smaller reduction map and a task-specific head for the new task. And recent advances introducing parameter-efficient fine-tuning methods such as low-rank adaptation Yang et al. (2025b). On the contrary, graph prompting keeps the pre-trained parameters frozen and enhances performance by inserting learnable prompt vectors Yu et al. (2025a); Liu et al. (2023); Yu et al. (2024a;b; 2025b). For instance, Liu et al. (2023) unifies tasks under a subgraph similarity template and employs a learnable vector to guide the READOUT function. Other approaches generate more adaptive prompts, such as PRONOG Yu et al. (2025b), which uses a conditional network to create node-specific prompts for non-homophilic graphs, and PRODIGY Huang et al. (2023), which formulates a novel prompt graph for in-context learning. To tackle more complex scenarios, several works have developed dual-prompting mechanisms. Jiao et al. (2025); Yu et al. (2024a) introduce a feature prompt and a heterogeneity prompt to bridge the gap between homogeneous and heterogeneous graphs. Yu et al. (2024c) leverages a composed prompt for task-specific knowledge and an open prompt for global knowledge from multiple pre-training tasks. Similarly, Yu et al. (2025a) designs holistic and specific structural prompts for cross-domain adaptation. Yet, how to quantify the transfer effort to the target domain remains an open issue.

### E.4 Riemannian Graph Representation Learning

In recent years, Riemannian manifolds have emerged as a promising alternative to traditional Euclidean spaces in graph representation learning Sun et al. (2026b; 2025a; 2024a;b; 2026a; 2023b;a; 2024c). Most existing Riemannian models are tailored to specific tasks Grover et al. (2025), and often leverage the particular manifolds, such as the hyperbolic space Chami et al. (2019); Yang et al. (2025a), the spherical space Liu et al. (2022), the symmetric positive definite manifold Ju & Guan (2024), and their products Gu et al. (2019); Bachmann et al. (2020); Sun et al. (2022a) and quotients Xiong et al. (2022). Very recently, Sun et al. (2025b) introduces a structural vocabulary and designs a new GNN backbone on the product manifold for general-purpose graph foundation model. In contrast to backbone architecture design, our focus lies in developing a framework for multi-domain pre-training and characterizing a general manifold that underlies diverse graphs.

## F Empirical Details

### F.1 Dataset Description

This section provides a detailed description of the 12 benchmark datasets used in our experiments. For a summary of their statistics, please refer to Table 6.

Table 6: Statistics of 12 datasets used in our experiment.

| Domain | Dataset | Task | # Graphs | Avg. #Nodes | Avg. #Edges | # Classes |
|---|---|---|---|---|---|---|
| Citation | PubMed | Node | 1 | 19,717 | 88,648 | 3 |
| | Arxiv | Node | 1 | 169,343 | 1,166,243 | 40 |
| Co-purchase | Computers | Node | 1 | 13,752 | 491,722 | 10 |
| | Photo | Node | 1 | 7,650 | 238,162 | 8 |
| Social Network | Reddit | Node | 1 | 232,965 | 114,615,892 | 41 |
| | FacebookPagePage | Node | 1 | 22,470 | 342,004 | 4 |
| Knowledge Graph | FB15K_237 | Edge | 1 | 14,541 | 310,116 | 237 |
| | WordNet18RR | Edge | 1 | 40,943 | 93,003 | 11 |
| Bioinformatics | PROTEINS | Graph | 1,113 | 39.1 | 145.6 | 2 |
| | MUTAG | Graph | 188 | 17.9 | 39.6 | 2 |
| Molecule | HIV | Graph | 41,127 | 25.5 | 27.5 | 2 |
| | Lipophilicity | Graph | 4,200 | 27.0 | 59.0 | 2 |

Our experiments utilize a diverse set of 12 benchmark datasets. For citation networks, we include PubMed, where nodes represent scientific publications, and the task is to classify their category, as well as Arxiv, a large-scale network for academic paper classification. In the co-purchase domain, both Computers and Photo are sourced from Amazon; in these graphs, nodes are products, edges signify frequent co-purchases, and the task is to predict product categories. For social networks, Reddit is constructed from posts, with the goal of predicting a post's community, while FacebookPagePage consists of official pages with edges as mutual likes, and the task is to classify the page's category. Our knowledge graph datasets include WordNet18RR, where the task is to classify the semantic relation between synsets, and FB15K_237, used to predict the relation type between entities. Finally, for graph-level classification, we use several benchmarks: PROTEINS and MUTAG are bioinformatics datasets for binary classification, with the latter predicting compound mutagenicity; similarly, HIV and Lipophilicity are molecular datasets for binary classification tasks that predict molecular properties.

## F.2 BASELINES

We evaluate our model against a comprehensive set of baselines from three main categories: Supervised GNNs, Self-Supervised GNNs, and Graph Foundation Models.

**Supervised GNNs** This category includes foundational GNNs that are trained from scratch in a supervised manner for a specific downstream task.

- GCN Kipf & Welling (2017) is a widely used GNN model that generates node representations by aggregating information from local node neighborhoods. It employs a mean-pooling approach for neighborhood aggregation to integrate information from adjacent nodes.
- GraphSAGE Hamilton et al. (2017) is an inductive representation learning framework designed for large graphs. It utilizes a mean-pooling propagation rule and often employs a neighborhood sampling approach to scale efficiently to large-scale graphs.
- GIN Xu et al. (2019) is a state-of-the-art GNN that is commonly used as a powerful supervised baseline, particularly for graph classification tasks.

**Self-Supervised GNNs** These methods first pre-train a GNN encoder on unlabeled graph data using self-supervised objectives and are then fine-tuned for downstream tasks. They represent the predominant pre-training paradigm in graph machine learning.

- DGI Veličković et al. (2019) learns node representations by maximizing the mutual information between local patch representations and a global graph summary vector. Its contrastive objective is notably not based on random walks.
- GraphMAE Hou et al. (2022) operates by masking a portion of node features and training a GNN-based architecture to reconstruct them. It utilizes a scaled cosine error for reconstruction to improve training robustness.

- GCC Qiu et al. (2020) is a self-supervised pre-training framework designed to capture transferable structural representations across multiple networks. Its pre-training task is subgraph instance discrimination, using contrastive learning to distinguish between augmented views of a node's local subgraph and those from other nodes.

**Graph Foundation Models** This group comprises recent, large-scale models pre-trained on diverse datasets and fine-tuned for strong generalization. They are the most direct competitors to our work and represent the current state-of-the-art.

- PRODIGY Huang et al. (2023) enables in-context learning over graphs by formulating tasks with a novel prompt graph representation. This structure connects prompt examples with queries, allowing the model to perform new tasks without updating its parameters.

- GFT Wang et al. (2024) rethinks transferable patterns as computation trees derived from the GNN message-passing process. It uses a tree reconstruction task for pre-training and unifies downstream tasks as tree classification.

- RAGraph Jiang et al. (2024) is a retrieval-augmented framework that improves GNN generalization by retrieving knowledge from an external library of toy graphs. The retrieved information is injected into the target graph using a message-passing prompt mechanism to enhance performance.

- SAMGPT Yu et al. (2025a) is a text-free graph foundation model for multi-domain pre-training and cross-domain adaptation. It uses learnable structure tokens to harmonize structural differences across domains during pre-training and dual prompts to adapt knowledge to new target domains.

- GCOPE Zhao et al. (2024) mitigates negative transfer during cross-domain pre-training by introducing coordinators, which are virtual nodes that act as bridges between disparate graph datasets. This approach helps create a unified representation from multiple graphs.

- MDGFM Wang et al. (2025a) focuses on achieving robust knowledge transfer through topology alignment. It employs a Graph Structure Learning (GSL) module to refine graph structures, reduce noise, and learn domain-invariant knowledge.

### F.3 IMPLEMENTATION NOTES

Our primary framework is a leave-one-out cross-domain evaluation. We pre-train models on five source datasets and evaluate them on a single held-out target dataset. This protocol applies to Self-Supervised GNNs and Graph Foundation Models. In contrast, supervised GNNs are not pre-trained and are instead trained directly from scratch on the target task. All downstream evaluations use a few-shot fine-tuning setting. For the pre-trained models, we use only k labeled samples per class from the target task for fine-tuning. In our experiments, k is set to 1 and 5. After setting aside these training samples, the remaining data is randomly split into a validation set (10%) and a test set (90%). We evaluate performance across three downstream tasks: node classification, Link Classification, and graph classification. For node and Link Classification, we use Accuracy (ACC) as the evaluation metric. For graph classification, we use Area Under the Curve (AUC). To ensure robust results, the final reported score for each experiment is the average over 10 runs with different random data splits.

For pretraining, we extract the 2-hop ego-graph with 10 neighbors each hop for single graph datasets and adopt a 2-layer GCN Kipf & Welling (2017) as backbone model. The dimension of the manifold, or the number of virtual nodes in $(k, M)$-sparse perturbation, is set to $M = 32$ with $k = 15$. For the KNN construction for mixed data training in Algorithm 1 and multi-graph datasets training, we also set $k = 15$. The dropout rate is $0.1$ and the learning rate is $1e^{-4}$. The model input dimension is 128. For different datasets, we unify the input dimension by random projection or SVD. For the knowledge graph datasets, we use Node2Vec Grover & Leskovec (2016) to get the node embeddings. The hidden dimension is $512$. The temperature in contrastive learning is $1.0$. The optimizer is Adam Kingma & Ba (2015), with a cosine annealing schedule Loshchilov & Hutter (2017).

Table 7 to Table 12 show the hyperparameters in few-shot transferring: the learning rate $lr$, dropout rate $drop$, the KNN number $k$ between prototypes and target graph data points, and the balance coefficient $\lambda$. We adopt the classifier head with only a linear layer for the node or graph classification

task. For the link classification task, we simply adopt a bilinear layer as a classifier. The gated function for Riemannian MoE is an MLP with 2 layers.

Table 7: Hyper-parameters for 1-shot and 5-shot cross-domain transfer on `Arxiv`.

|  | $lr$ | $drop$ | $k$ | $\lambda$ |
|---|---|---|---|---|
| 1-shot | $1e^{-3}$ | 0.1 | 3 | 1.0 |
| 5-shot | $1e^{-3}$ | 0.15 | 3 | 1.0 |

Table 8: Hyper-parameters for 1-shot and 5-shot cross-domain transfer on `Computers`.

|  | $lr$ | $drop$ | $k$ | $\lambda$ |
|---|---|---|---|---|
| 1-shot | $1e^{-3}$ | 0.2 | 3 | 1.0 |
| 5-shot | $1e^{-3}$ | 0.2 | 3 | 1.0 |

Table 9: Hyper-parameters for 1-shot and 5-shot cross-domain transfer on `Reddit`.

|  | $lr$ | $drop$ | $k$ | $\lambda$ |
|---|---|---|---|---|
| 1-shot | $1e^{-3}$ | 0.1 | 3 | 0.1 |
| 5-shot | $1e^{-3}$ | 0.15 | 3 | 0.1 |

Table 10: Hyper-parameters for 1-shot and 5-shot cross-domain transfer on `FB15k_237`.

|  | $lr$ | $drop$ | $k$ | $\lambda$ |
|---|---|---|---|---|
| 1-shot | $1e^{-4}$ | 0.5 | 3 | 0.5 |
| 5-shot | $1e^{-4}$ | 0.5 | 3 | 0.5 |

Table 11: Hyper-parameters for 1-shot and 5-shot cross-domain transfer on `PROTEINS`.

|  | $lr$ | $drop$ | $k$ | $\lambda$ |
|---|---|---|---|---|
| 1-shot | $1e^{-3}$ | 0.1 | 1 | 2.0 |
| 5-shot | $1e^{-3}$ | 0.15 | 1 | 2.0 |

Table 12: Hyper-parameters for 1-shot and 5-shot cross-domain transfer on `HIV`.

|  | $lr$ | $drop$ | $k$ | $\lambda$ |
|---|---|---|---|---|
| 1-shot | $1e^{-3}$ | 0.1 | 2 | 2.0 |
| 5-shot | $1e^{-3}$ | 0.15 | 2 | 2.0 |

# G    ADDITIONAL RESULTS

## G.1    SUPPLEMENTARY RESULTS

We provide additional empirical results to further validate our framework. We present comprehensive results for both cross-domain transfer (Table 18 and 19) and intra-domain transfer (Table 20 and 21) in few-shot settings. Furthermore, an ablation study in Table 22 demonstrates the effectiveness of the key components of our model. We also include an additional visualization of the pre-trained manifold in Figure 8, where we first project the 512-dimensional embeddings into 3-D using t-SNE van der Maaten & Hinton (2008) and then apply RBF interpolation Wright & Fornberg (2006) to generate a smooth surface that approximates the learned global Riemannian manifold.

## G.2    COMPREHENSIVE ABLATION STUDY

We conduct a further ablation study to verify the effectiveness of EMA, prototype loss and Riemannian MoE. Specifically, we introduce 3 variants of GraphGlue, described as follows:

- "w/o EMA" means that we replace EMA with the common average of a batch of embeddings;

- "w/o $\mathcal{L}_{proto}$" means pretraining without prototype loss;

- "w/o Riemannian MoE" means that during adaption, Riemannian MoE module is replaced by a typical prompting scheme.

In Table 13, both results on 1-shot setting and 5-shot setting demonstrate the effectiveness of the proposed components.

Table 13: Ablation study of GraphGlue's key components.

| | Variants | Arxiv | Computers | Reddit | FB15k_237 | PROTEINS | HIV |
|---|---|---|---|---|---|---|---|
| **1-shot** | w/o EMA | 15.46±1.41 | 30.84±9.50 | 7.43±2.37 | 35.90±17.40 | 58.48±2.56 | 52.62±2.41 |
| | w/o L_proto | 9.57±4.56 | 31.24±10.36 | 37.90±9.34 | 43.59±8.13 | 58.49±2.60 | 54.10±2.76 |
| | GRAPHGLUE | **29.73±2.56** | **61.03±7.13** | **68.42±4.68** | **60.89±2.11** | **69.12±4.19** | **58.53±8.20** |
| **5-shot** | w/o EMA | 16.08±2.13 | 34.90±7.77 | 11.53±2.26 | 40.21±12.07 | 59.85±4.53 | 53.87±2.43 |
| | w/o L_proto | 15.26±9.91 | 36.63±11.84 | 46.13±14.14 | 64.56±16.42 | 61.64±6.28 | 55.50±2.70 |
| | GRAPHGLUE | **39.98±1.67** | **74.15±2.38** | **84.89±0.68** | **79.52±1.75** | **73.94±2.38** | **62.18±2.50** |

## G.3 HYPERPARAMETER SENSITIVITY ANALYSIS

For AOF, we investigate the hyperparameter sensitivity on the neighborhood size $k$ and the number of nodes $M$ in $(k, M)$-sparse perturbation. Results are shown in Table 14 and 15.

Table 14: 1-shot results under different settings.

(a) Analysis on $k$ ($M = 32$).

| $k$ | Arxiv | Computers | Reddit | FB15k_237 | PROTEINS | HIV |
|---|---|---|---|---|---|---|
| 2 | 18.64±3.10 | 49.80±12.41 | 66.04±1.91 | 31.63±6.19 | 57.80±3.05 | 53.07±3.02 |
| 5 | 17.16±3.08 | 43.76±10.78 | 48.57±6.76 | 21.10±2.47 | 59.49±2.62 | 52.04±3.01 |
| 10 | 15.29±2.69 | 46.21±11.10 | 61.03±2.89 | 45.51±15.83 | 58.10±3.41 | 54.42±3.16 |
| 15 | **29.73±2.56** | **61.03±7.13** | **68.42±4.68** | **60.89±2.11** | **69.12±4.19** | **58.53±8.20** |
| 30 | 20.23±2.83 | 55.39±10.15 | 49.57±4.80 | 38.85±19.62 | 54.36±5.80 | 54.49±3.33 |
| 60 | 18.20±2.63 | 51.88±10.22 | 75.76±3.00 | 31.83±16.41 | 58.24±3.34 | 51.64±3.44 |

(b) Analysis on $M$ ($k = 15$).

| $M$ | Arxiv | Computers | Reddit | FB15k_237 | PROTEINS | HIV |
|---|---|---|---|---|---|---|
| 4 | 19.89±3.79 | 42.66±11.17 | 60.57±4.48 | 45.16±5.85 | 56.99±4.64 | 52.58±3.68 |
| 8 | 22.64±2.51 | 46.24±9.12 | 61.73±5.02 | 54.80±9.57 | 58.77±1.92 | 52.32±2.94 |
| 16 | 27.84±1.47 | 56.92±14.86 | 62.73±1.97 | 57.09±7.44 | 55.34±5.68 | 54.18±3.84 |
| 32 | **29.73±2.56** | **61.03±7.13** | **68.42±4.68** | **60.89±2.11** | **69.12±4.19** | **58.53±8.20** |

Table 15: 5-shot results under different settings.

(a) on $k$ ($M = 32$).

| $k$ | Arxiv | Computers | Reddit | FB15k_237 | PROTEINS | HIV |
|---|---|---|---|---|---|---|
| 2 | 29.49±3.14 | 70.82±3.14 | 80.46±1.21 | 36.24±5.11 | 59.42±2.03 | 56.10±2.76 |
| 5 | 23.95±7.88 | 67.74±4.02 | 67.41±8.74 | 39.26±14.50 | 60.45±3.05 | 54.76±2.33 |
| 10 | 25.86±7.17 | 70.59±18.03 | 80.49±0.59 | 48.01±10.53 | 60.86±3.08 | 55.69±4.23 |
| 15 | **39.98±1.67** | **74.15±2.38** | **84.89±0.68** | **79.52±1.75** | **73.94±2.38** | **62.18±2.50** |
| 30 | 33.00±1.63 | 57.65±29.35 | 64.57±6.84 | 42.88±16.13 | 62.09±2.45 | 54.63±3.01 |
| 60 | 32.59±1.57 | 68.08±1.66 | 85.24±0.42 | 45.42±8.09 | 60.40±1.35 | 54.01±3.69 |

(b) Analysis on $M$ ($k = 15$).

| $M$ | Arxiv | Computers | Reddit | FB15k_237 | PROTEINS | HIV |
|---|---|---|---|---|---|---|
| 4 | 32.62±3.40 | 64.66±13.87 | 70.86±2.92 | 60.63±4.80 | 57.38±4.76 | 56.27±1.62 |
| 8 | 34.92±1.50 | 72.25±1.99 | 78.82±1.79 | 63.18±14.37 | 59.70±1.92 | 54.64±2.35 |
| 16 | 37.78±5.79 | 74.22±13.95 | 74.92±2.41 | 70.11±11.89 | 60.62±3.66 | 57.55±2.65 |
| 32 | **39.98±1.67** | **74.15±2.38** | **84.89±0.68** | **79.52±1.75** | **73.94±2.38** | **62.18±2.50** |

### G.4 RESULTS ON HETEROPHILIC GRAPHS

We demonstrate the performance of GraphGlue on several benchmarking heterophilic graphs (Amazon-ratings, Roman-empire, Texas and Wisconsin). The results are in Table 16 and 17.

Table 16: Performance under different shot settings with pretrained on ogbn-arxiv, Reddit, Computers, FB15k_237, PROTEINS, and HIV.

|        | Method    | Amazon-Ratings   | Roman-empire      | Texas             | Wisconsin          |
|--------|-----------|------------------|-------------------|-------------------|--------------------|
| 1-shot | GCOPE     | 28.65±5.82       | 11.44±1.91        | 33.19±6.62        | 31.22±6.85         |
|        | MDGFM     | 29.53±3.45       | 14.51±2.08        | 34.63±10.70       | 35.11±10.53        |
|        | GraphGlue | **31.16±3.56**   | **16.23±3.00**    | **35.16±20.43**   | **37.95±10.91**    |
| 5-shot | GCOPE     | 30.06±5.11       | 16.00±1.29        | 36.31±10.14       | 38.21±2.96         |
|        | MDGFM     | 30.42±3.80       | 17.15±1.66        | 48.33±6.36        | 47.46±4.86         |
|        | GraphGlue | **32.17±2.91**   | **18.50±1.07**    | **50.88±11.93**   | **49.71±8.00**     |

Table 17: Performance under different shot settings with pretrained on 8 datasets (including Amazon-Ratings and Roman-Empire).

|        | Method    | Amazon-Ratings   | Roman-empire      | Texas             | Wisconsin          |
|--------|-----------|------------------|-------------------|-------------------|--------------------|
| 1-shot | GCOPE     | 29.03±4.17       | 13.14±2.36        | 33.82±7.39        | 30.08±5.13         |
|        | MDGFM     | 27.01±2.98       | 14.11±2.14        | 36.02±9.54        | 33.28±8.76         |
|        | GraphGlue | **34.12±2.57**   | **18.19±2.51**    | **38.65±13.88**   | **40.17±10.09**    |
| 5-shot | GCOPE     | 32.83±3.26       | 16.98±1.40        | 41.33±9.85        | 43.74±3.19         |
|        | MDGFM     | 32.54±3.75       | 16.77±1.92        | 48.10±7.26        | 46.62±4.16         |
|        | GraphGlue | **36.26±3.09**   | **20.67±1.34**    | **52.60±6.07**    | **51.49±7.97**     |

### G.5 VISUALIZATION OF MANIFOLD GLUING

Manifold gluing aims to glue local pieces into one smooth surface, whose process is described as follows.

- First, we construct local geometry on each patch using $(k, M)$-sparse perturbation–like drawing a coordinate grid;
- Then, when two graphs share similar structures, we "glue" their grids together along overlapping regions, ensuring no stretching or twisting (via the isometry of Def. 4.4 and holonomy of Eq. 5);
- Finally, we smooth the entire surface so that curvature changes gradually–forming a unified manifold where knowledge flows naturally across domains.

In addition, we visualize a toy example of the aforementioned process in Figure 7.

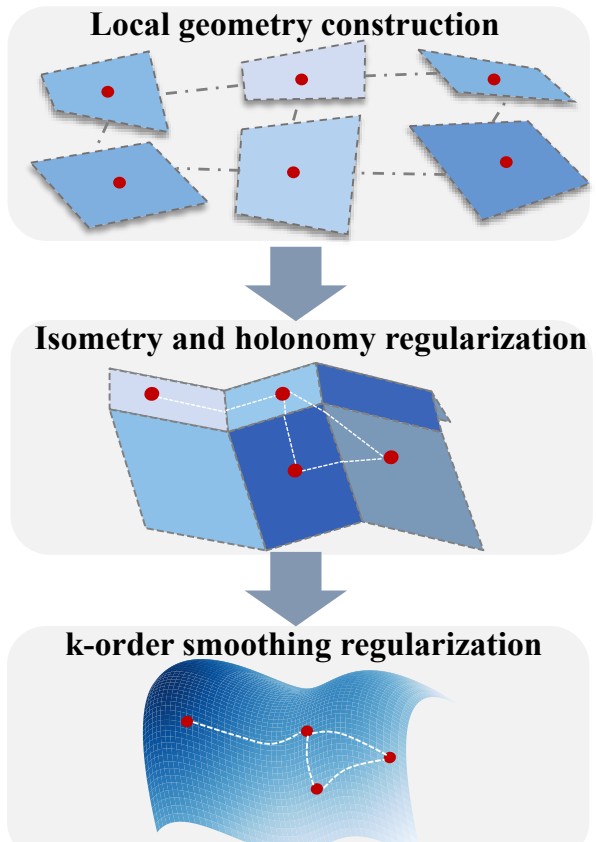

Figure 7: Visualization of the pre-trained manifold from 6 datasets.

Table 18: Performance of cross-domain transfer on various downstream tasks in the 1-shot setting, reported as mean ± std over 10 runs. The highest result is **bolded**, and the runner-up is underlined.

| Model | Node Classification | | | Link Classification | Graph Classification | |
|---|---|---|---|---|---|---|
| | Arxiv | Computers | Reddit | FB15k_237 | PROTEINS | HIV |
| GCN | $12.61_{\pm1.75}$ | $33.89_{\pm3.86}$ | $11.15_{\pm2.14}$ | $32.11_{\pm2.37}$ | $50.11_{\pm13.07}$ | $52.56_{\pm5.39}$ |
| GraphSAGE | $14.68_{\pm3.76}$ | $35.47_{\pm8.29}$ | $14.69_{\pm2.31}$ | $35.74_{\pm2.19}$ | $58.99_{\pm2.79}$ | $56.78_{\pm3.75}$ |
| GIN | $11.20_{\pm2.03}$ | $44.77_{\pm6.02}$ | $18.53_{\pm1.89}$ | $38.25_{\pm2.55}$ | $54.22_{\pm13.50}$ | $52.63_{\pm7.47}$ |
| GCC | $12.65_{\pm2.08}$ | $34.82_{\pm6.13}$ | $54.78_{\pm5.64}$ | $47.84_{\pm1.95}$ | $59.20_{\pm7.97}$ | $52.63_{\pm3.63}$ |
| DGI | $13.32_{\pm3.35}$ | $35.26_{\pm7.58}$ | $60.08_{\pm4.80}$ | $42.50_{\pm2.03}$ | $53.18_{\pm8.44}$ | $52.80_{\pm7.53}$ |
| GraphMAE | $12.61_{\pm1.75}$ | $33.89_{\pm3.86}$ | $11.15_{\pm2.14}$ | $51.34_{\pm1.87}$ | $\mathbf{60.11}_{\pm13.07}$ | $52.78_{\pm6.72}$ |
| PRODIGY | $28.45_{\pm2.20}$ | $45.32_{\pm4.10}$ | $35.67_{\pm3.20}$ | $53.50_{\pm1.02}$ | $48.90_{\pm5.40}$ | $41.78_{\pm4.50}$ |
| GFT | $26.59_{\pm2.45}$ | $54.65_{\pm4.08}$ | $58.87_{\pm2.53}$ | $58.07_{\pm1.39}$ | $55.41_{\pm5.87}$ | $58.94_{\pm6.32}$ |
| RAGraph | $18.71_{\pm2.58}$ | $46.21_{\pm4.37}$ | $52.56_{\pm3.48}$ | $52.18_{\pm3.04}$ | $51.42_{\pm5.18}$ | $54.26_{\pm3.51}$ |
| SAMGPT | $24.15_{\pm3.81}$ | $47.61_{\pm7.42}$ | $62.85_{\pm4.22}$ | $57.44_{\pm2.46}$ | $52.42_{\pm3.15}$ | $55.48_{\pm3.26}$ |
| GCOPE | $26.52_{\pm5.56}$ | $54.55_{\pm9.14}$ | $62.76_{\pm4.52}$ | $58.25_{\pm2.67}$ | $55.19_{\pm3.59}$ | $58.93_{\pm2.60}$ |
| MDGFM | $26.05_{\pm2.40}$ | $46.68_{\pm8.43}$ | $64.88_{\pm3.31}$ | $56.11_{\pm1.68}$ | $53.41_{\pm5.34}$ | $51.46_{\pm2.85}$ |
| GRAPHGLUE | $\mathbf{28.88}_{\pm5.22}$ | $\mathbf{59.50}_{\pm7.05}$ | $\mathbf{67.12}_{\pm3.39}$ | $\mathbf{59.75}_{\pm5.27}$ | $59.87_{\pm4.85}$ | $\mathbf{60.22}_{\pm3.09}$ |

Table 19: Performance of cross-domain transfer on various downstream tasks in the 5-shot setting, reported as mean ± std over 10 runs. The highest result is **bolded**, and the runner-up is underlined.

| Model | Node Classification | | | Link Classification | Graph Classification | |
|---|---|---|---|---|---|---|
| | Arxiv | Computers | Reddit | FB15k_237 | PROTEINS | HIV |
| GCN | $27.68_{\pm2.13}$ | $65.78_{\pm4.20}$ | $28.36_{\pm1.01}$ | $52.43_{\pm1.87}$ | $55.04_{\pm9.98}$ | $47.81_{\pm3.91}$ |
| GraphSAGE | $26.18_{\pm2.21}$ | $66.75_{\pm4.45}$ | $22.27_{\pm1.17}$ | $58.91_{\pm1.52}$ | $60.45_{\pm1.39}$ | $50.59_{\pm0.75}$ |
| GIN | $26.06_{\pm2.42}$ | $69.51_{\pm3.50}$ | $29.03_{\pm1.66}$ | $63.76_{\pm1.73}$ | $58.87_{\pm5.05}$ | $49.12_{\pm4.95}$ |
| GCC | $26.84_{\pm2.14}$ | $62.63_{\pm3.16}$ | $65.21_{\pm1.56}$ | $73.69_{\pm1.24}$ | $64.20_{\pm3.09}$ | $57.41_{\pm1.73}$ |
| DGI | $27.18_{\pm2.33}$ | $61.02_{\pm3.20}$ | $62.72_{\pm2.21}$ | $68.32_{\pm1.46}$ | $53.34_{\pm6.27}$ | $52.23_{\pm8.49}$ |
| GraphMAE | $27.68_{\pm2.13}$ | $65.78_{\pm4.20}$ | $28.36_{\pm1.01}$ | $77.25_{\pm1.07}$ | $65.04_{\pm9.98}$ | $57.81_{\pm3.91}$ |
| PRODIGY | $33.67_{\pm2.80}$ | $52.78_{\pm3.60}$ | $42.34_{\pm2.90}$ | $72.17_{\pm6.94}$ | $55.23_{\pm4.70}$ | $48.65_{\pm3.80}$ |
| GFT | $36.78_{\pm1.92}$ | $69.13_{\pm3.56}$ | $66.28_{\pm1.42}$ | $79.13_{\pm1.68}$ | $62.18_{\pm3.59}$ | $57.68_{\pm5.43}$ |
| RAGraph | $32.35_{\pm1.78}$ | $62.38_{\pm3.75}$ | $63.08_{\pm1.32}$ | $64.52_{\pm2.57}$ | $58.62_{\pm2.86}$ | $56.32_{\pm3.46}$ |
| SAMGPT | $34.42_{\pm2.25}$ | $60.87_{\pm3.64}$ | $75.12_{\pm1.63}$ | $77.63_{\pm2.71}$ | $59.14_{\pm2.60}$ | $57.63_{\pm2.87}$ |
| GCOPE | $\mathbf{39.18}_{\pm1.96}$ | $72.27_{\pm2.84}$ | $80.45_{\pm0.70}$ | $79.38_{\pm2.29}$ | $64.85_{\pm2.41}$ | $58.47_{\pm1.82}$ |
| MDGFM | $32.28_{\pm1.77}$ | $64.08_{\pm5.38}$ | $76.55_{\pm1.72}$ | $77.67_{\pm2.05}$ | $57.79_{\pm3.42}$ | $55.79_{\pm3.16}$ |
| GRAPHGLUE | $37.02_{\pm2.33}$ | $\mathbf{73.29}_{\pm0.70}$ | $\mathbf{85.05}_{\pm1.17}$ | $\mathbf{81.51}_{\pm2.31}$ | $\mathbf{65.32}_{\pm2.45}$ | $\mathbf{61.55}_{\pm2.66}$ |

Table 20: Performance of intra-domain transfer on various downstream tasks in the 1-shot setting, reported as mean ± std over 10 runs. The highest result is **bolded**, and the runner-up is underlined.

| Model | Node Classification | | | Link Classification | Graph Classification | |
|---|---|---|---|---|---|---|
| | Arxiv | Computers | Reddit | FB15k_237 | PROTEINS | HIV |
| GCN | $12.61_{\pm1.75}$ | $33.89_{\pm3.86}$ | $11.15_{\pm2.14}$ | $32.11_{\pm2.37}$ | $60.11_{\pm13.07}$ | $52.56_{\pm5.39}$ |
| GraphSAGE | $14.68_{\pm3.76}$ | $35.47_{\pm8.29}$ | $14.69_{\pm2.31}$ | $35.74_{\pm2.19}$ | $68.99_{\pm2.79}$ | $56.78_{\pm3.75}$ |
| GIN | $11.20_{\pm2.03}$ | $44.77_{\pm6.02}$ | $18.53_{\pm1.89}$ | $38.25_{\pm2.55}$ | $64.22_{\pm13.50}$ | $52.63_{\pm7.47}$ |
| GCC | $12.65_{\pm2.08}$ | $34.82_{\pm6.13}$ | $54.78_{\pm5.64}$ | $47.80_{\pm1.97}$ | $59.20_{\pm7.97}$ | $52.63_{\pm3.63}$ |
| DGI | $13.32_{\pm3.35}$ | $35.26_{\pm7.58}$ | $60.08_{\pm4.80}$ | $42.56_{\pm2.05}$ | $53.18_{\pm8.44}$ | $52.80_{\pm7.53}$ |
| GraphMAE | $12.61_{\pm1.75}$ | $33.89_{\pm3.86}$ | $11.15_{\pm2.14}$ | $51.34_{\pm1.87}$ | $60.11_{\pm13.07}$ | $52.56_{\pm5.39}$ |
| PRODIGY | $28.45_{\pm2.20}$ | $45.32_{\pm4.10}$ | $35.67_{\pm3.20}$ | $53.50_{\pm1.02}$ | $48.90_{\pm5.40}$ | $41.78_{\pm4.50}$ |
| GFT | $28.83_{\pm1.76}$ | $53.94_{\pm3.47}$ | $63.03_{\pm2.34}$ | $59.43_{\pm0.87}$ | $63.54_{\pm4.98}$ | $58.17_{\pm5.76}$ |
| RAGraph | $20.53_{\pm2.13}$ | $50.39_{\pm3.81}$ | $59.91_{\pm2.79}$ | $52.09_{\pm2.57}$ | $52.83_{\pm4.37}$ | $55.73_{\pm3.06}$ |
| SAMGPT | $25.88_{\pm3.58}$ | $55.31_{\pm6.67}$ | $63.05_{\pm3.75}$ | $58.75_{\pm2.16}$ | $64.59_{\pm2.89}$ | $52.38_{\pm2.71}$ |
| GCOPE | $27.41_{\pm4.77}$ | $58.24_{\pm7.48}$ | $65.07_{\pm3.76}$ | $58.33_{\pm1.79}$ | $68.55_{\pm3.17}$ | $\mathbf{60.67}_{\pm2.42}$ |
| MDGFM | $10.76_{\pm2.04}$ | $43.22_{\pm8.53}$ | $64.38_{\pm3.11}$ | $58.32_{\pm1.71}$ | $57.79_{\pm11.51}$ | $53.03_{\pm3.88}$ |
| GRAPHGLUE | $\mathbf{29.73}_{\pm2.56}$ | $\mathbf{61.03}_{\pm7.13}$ | $\mathbf{68.42}_{\pm4.68}$ | $\mathbf{60.89}_{\pm2.11}$ | $\mathbf{69.12}_{\pm4.19}$ | $58.53_{\pm8.20}$ |

Table 21: Performance of intra-domain transfer on various downstream tasks in the 5-shot setting, reported as mean ± std over 10 runs. The highest result is **bolded**, and the runner-up is underlined.

| Model | Node Classification | | | Link Classification | Graph Classification | |
|---|---|---|---|---|---|---|
| | Arxiv | Computers | Reddit | FB15k_237 | PROTEINS | HIV |
| GCN | $27.68_{\pm2.13}$ | $65.78_{\pm4.20}$ | $28.36_{\pm1.01}$ | $52.43_{\pm1.87}$ | $65.04_{\pm9.98}$ | $57.81_{\pm3.91}$ |
| GraphSAGE | $26.18_{\pm2.21}$ | $66.75_{\pm4.45}$ | $22.27_{\pm1.17}$ | $58.91_{\pm1.52}$ | $70.45_{\pm1.39}$ | $60.59_{\pm0.75}$ |
| GIN | $26.06_{\pm2.42}$ | $69.51_{\pm3.50}$ | $29.03_{\pm1.66}$ | $63.76_{\pm1.73}$ | $68.87_{\pm5.05}$ | $59.12_{\pm4.95}$ |
| GCC | $26.84_{\pm2.14}$ | $62.63_{\pm3.16}$ | $65.21_{\pm1.56}$ | $73.64_{\pm1.25}$ | $64.20_{\pm3.09}$ | $58.34_{\pm2.19}$ |
| DGI | $27.18_{\pm2.33}$ | $61.02_{\pm3.20}$ | $62.72_{\pm2.21}$ | $68.32_{\pm1.47}$ | $53.34_{\pm6.27}$ | $52.23_{\pm8.49}$ |
| GraphMAE | $27.68_{\pm2.13}$ | $65.78_{\pm4.20}$ | $28.36_{\pm1.01}$ | $77.25_{\pm1.07}$ | $65.04_{\pm9.98}$ | $57.81_{\pm3.91}$ |
| PRODIGY | $33.67_{\pm2.80}$ | $52.78_{\pm3.60}$ | $42.34_{\pm2.90}$ | $72.17_{\pm6.94}$ | $55.23_{\pm4.70}$ | $48.65_{\pm3.80}$ |
| GFT | $39.02_{\pm1.39}$ | $73.41_{\pm3.21}$ | $71.37_{\pm1.45}$ | $79.25_{\pm0.94}$ | $\mathbf{74.69}_{\pm2.84}$ | $61.03_{\pm4.83}$ |
| RAGraph | $35.74_{\pm1.46}$ | $61.98_{\pm2.79}$ | $66.30_{\pm0.75}$ | $67.86_{\pm1.69}$ | $62.52_{\pm3.83}$ | $59.23_{\pm2.80}$ |
| SAMGPT | $38.14_{\pm1.87}$ | $64.68_{\pm2.87}$ | $74.89_{\pm1.51}$ | $78.76_{\pm2.33}$ | $70.48_{\pm2.19}$ | $59.09_{\pm2.49}$ |
| GCOPE | $39.45_{\pm1.23}$ | $73.06_{\pm2.19}$ | $82.12_{\pm0.53}$ | $78.69_{\pm1.87}$ | $73.76_{\pm2.53}$ | $60.05_{\pm1.73}$ |
| MDGFM | $19.17_{\pm2.39}$ | $68.19_{\pm4.03}$ | $81.27_{\pm1.23}$ | $78.24_{\pm2.35}$ | $65.95_{\pm8.62}$ | $54.73_{\pm4.37}$ |
| GRAPHGLUE | $\mathbf{39.98}_{\pm1.67}$ | $\mathbf{74.15}_{\pm2.38}$ | $\mathbf{84.89}_{\pm0.68}$ | $\mathbf{79.52}_{\pm1.75}$ | $69.74_{\pm2.38}$ | $\mathbf{62.18}_{\pm2.50}$ |

Table 22: Ablation study of GRAPHGLUE's key components.

| | Variants | Arxiv | Computers | Reddit | FB15k_237 | PROTEINS | HIV |
|---|---|---|---|---|---|---|---|
| 1-shot | w/o $\mathcal{L}_{\text{curv}}$ | $22.33_{\pm 2.56}$ | $49.63_{\pm 5.11}$ | $64.38_{\pm 5.12}$ | $53.12_{\pm 3.74}$ | $51.21_{\pm 3.54}$ | $50.34_{\pm 3.87}$ |
| | w/o $\mathcal{L}_{\text{holo}}$ | $27.14_{\pm 3.62}$ | $56.39_{\pm 4.16}$ | $65.93_{\pm 4.33}$ | $54.85_{\pm 4.78}$ | $53.23_{\pm 4.48}$ | $54.83_{\pm 2.15}$ |
| | GRAPHGLUE | $\mathbf{28.88}_{\pm 5.22}$ | $\mathbf{59.50}_{\pm 7.05}$ | $\mathbf{67.12}_{\pm 3.39}$ | $\mathbf{59.75}_{\pm 5.27}$ | $\mathbf{55.87}_{\pm 4.85}$ | $\mathbf{60.22}_{\pm 3.09}$ |
| 5-shot | w/o $\mathcal{L}_{\text{curv}}$ | $29.17_{\pm 3.14}$ | $66.85_{\pm 3.58}$ | $74.13_{\pm 1.92}$ | $69.33_{\pm 3.98}$ | $58.77_{\pm 4.35}$ | $57.13_{\pm 3.33}$ |
| | w/o $\mathcal{L}_{\text{holo}}$ | $35.77_{\pm 2.64}$ | $67.16_{\pm 2.45}$ | $79.11_{\pm 3.47}$ | $74.02_{\pm 0.97}$ | $58.74_{\pm 2.18}$ | $54.12_{\pm 4.19}$ |
| | GRAPHGLUE | $\mathbf{37.02}_{\pm 2.33}$ | $\mathbf{73.29}_{\pm 0.70}$ | $\mathbf{85.05}_{\pm 1.17}$ | $\mathbf{81.51}_{\pm 2.31}$ | $\mathbf{65.32}_{\pm 2.45}$ | $\mathbf{61.55}_{\pm 2.66}$ |

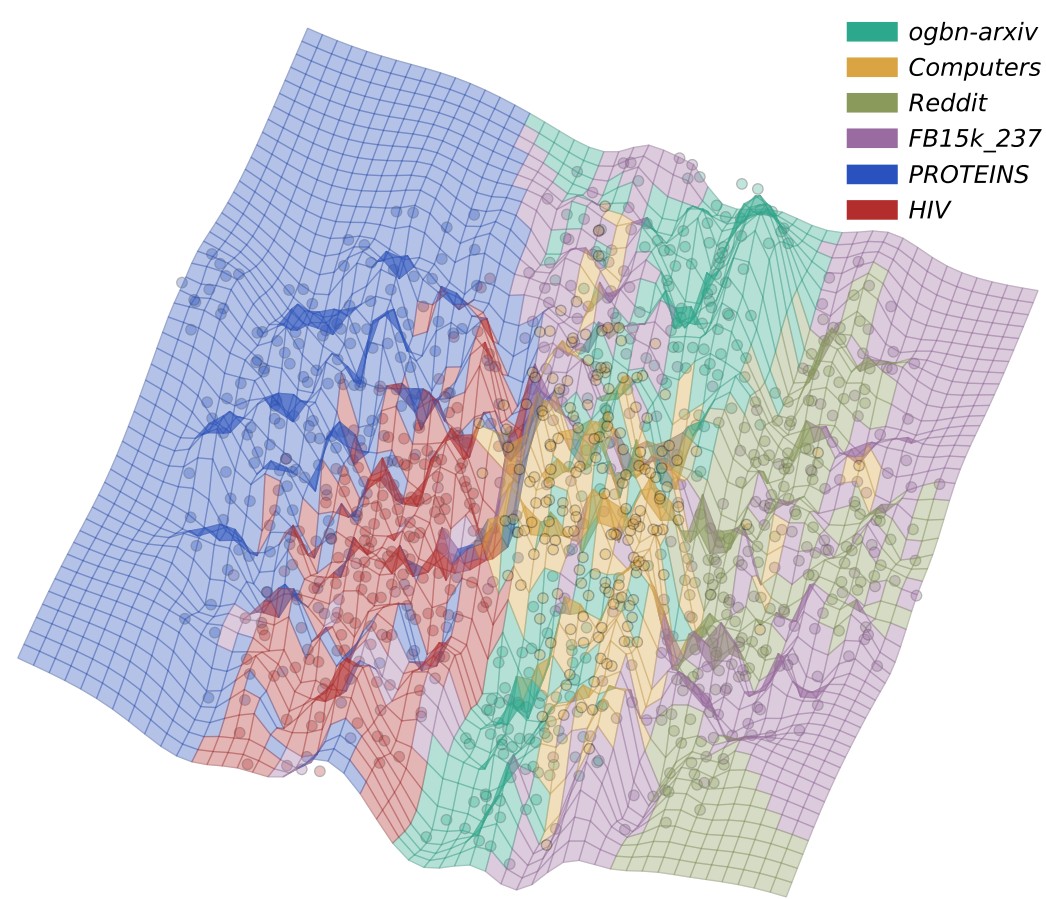

Figure 8: Visualization of the pre-trained manifold from 6 datasets.

## H    REPRODUCIBILITY STATEMENT

This part provides the reproducibility statement on claims, theory assumptions and proofs, empirical result reproducibility, empirical setting/details, empirical statistical significance, open access to data/code, computation resources, code of ethics, safeguards, licenses for existing assets, new assets, crowdsourcing and research with human subjects, declaration of LLM usage, and broader impacts.

1. **Claims.** Do the main claims made in the abstract and introduction accurately reflect the paper's contributions and scope?

   **Yes.** Main claims made in the abstract and introduction reflect the contributions in Sections 4, 5 and 6.

2. **Theory assumptions and proofs.** For each theoretical result, does the paper provide the full set of assumptions and a complete (and correct) proof?

   **Yes.** The theoretical results including the assumptions are clearly stated in Theorems in this paper, while the complete and correct proofs are provided in Appendix B.

3. **Empirical result reproducibility.** Does the paper fully disclose all the information needed to reproduce the main experimental results of the paper to the extent that it affects the main claims and/or conclusions of the paper (regardless of whether the code and data are provided or not)?

   **Yes.** Key information is introduced in the subsection of "Evaluation Protocals", and further details are disclosed in Appendix F entitled "Empirical Details".

4. **Empirical setting/details.** Does the paper specify all the training and test details (e.g., data splits, hyperparameters, how they were chosen, type of optimizer, etc.) necessary to understand the results?

   **Yes.** Specifications are provided in Appendix F entitled "Empirical Details", and the full details are included in the code.

5. **Empirical statistical significance.** Does the paper report error bars suitably and correctly defined or other appropriate information about the statistical significance of the experiments?

   **Yes.** In the experiment, each case undergoes 10 independent runs, and we report the mean with the error bar of standard derivations.

6. **Open access to data and code.** Does the paper provide open access to the data and code, with sufficient instructions to faithfully reproduce the main experimental results?

   **Yes.** Codes and data are available at the anonymous GitHub link with sufficient instructions.

7. **Computation resources.** For each experiment, does the paper provide sufficient information on the computer resources (type of compute workers, memory, time of execution) needed to reproduce the experiments?

   **Yes.** The computer resources for the evaluation are described in Appendix F entitled "Empirical Details".

8. **Code of ethics.** Does the research conducted in the paper conform, in every respect, with the ICLR Code of Ethics `https://iclr.cc/public/CodeOfEthics`?

   **Yes.** We confirm that the research conducted in the paper conform, in every respect, with the ICLR Code of Ethics.

9. **Safeguards.** Does the paper describe safeguards that have been put in place for responsible release of data or models that have a high risk for misuse (e.g., pretrained language models, image generators, or scraped datasets)?

   **Not available.**

10. **Licenses for existing assets.** Are the creators or original owners of assets (e.g., code, data, models), used in the paper, properly credited and are the license and terms of use explicitly mentioned and properly respected?

    **Yes.** The original papers that produced the code package or dataset are properly cited in this submission.

11. **New assets.** Are new assets introduced in the paper well documented and is the documentation provided alongside the assets?

    **Yes.** The documentation is provided alongside the Codes of the proposed model.

12. **Crowdsourcing and research with human subjects.** For crowdsourcing experiments and research with human subjects, does the paper include the full text of instructions given to participants and screenshots, if applicable, as well as details about compensation (if any)?

    **Not available.** This paper does not involve crowdsourcing nor research with human subjects.

13. **Institutional review board (IRB) approvals or equivalent for research with human subjects.** Does the paper describe potential risks incurred by study participants, whether such risks were disclosed to the subjects, and whether Institutional Review Board (IRB) approvals (or an equivalent approval/review based on the requirements of your country or institution) were obtained?

    **Not available.** This paper does not involve crowdsourcing nor research with human subjects.

14. **Declaration of LLM usage.** Does the paper describe the usage of LLMs (especially when it is an important, original, or non-standard component of the core methods in this research)?

    **Yes.** LLM is used to polish writing only, and we include a section of "Declaration of LLM Usage" in the Appendix.

15. **Broader impacts.** Does the paper discuss both potential positive societal impacts and negative societal impacts of the work performed?

    **Yes.** Both potential positive societal impacts and negative societal impacts are included in the section of "Broader Impact and Limitations" in the Appendix.

## I  ETHICS STATEMENT

We confirm that the research conducted in the paper conform, in every respect, with the ICLR Code of Ethics `https://iclr.cc/public/CodeOfEthics`.

## J  DECLARATION OF LLM USAGE

Large Language Model (LLM) is used to polish writing. Concretely, we refine the textual contents in Section 1 (Introduction) and Section 7 (Conclusion) with LLM.

## K  BROADER IMPACT

Our work brings together two previously separate domains – multi-domain graph pre-training and differential geometry. Our constructions taking in multi-domain graphs with a unified, smooth Riemannian manifold, thus enabling the solid tools of differential geometry to systematically understand the knowledge integration and transfer across graphs. Theoretically, we develop the neural manifold gluing that makes the differential geometry principles implementable through deep learning. In practice, the proposed pre-training model paves the way to build a powerful graph foundation model with better generality and quantifiable transferability.

Positive societal impacts lie in the transferability and generality of the proposed graph pre-training model, allowing for the analysis on more complicated real-world graphs. None of negative societal impacts we feel must be specifically highlighted.

