# OpenReview forum: "Multi-Domain Riemannian Graph Gluing for Building Graph Foundation Models"
_ICLR.cc/2026/Conference — ICLR 2026 Oral_

### Official Review · Reviewer_5PYX · 2025-10-30

**Soundness:** 3
**Presentation:** 3
**Contribution:** 3
**Rating:** 6
**Confidence:** 4

**Summary:**

The paper proposes a differential-geometry view of multi-domain graph pre-training: neural manifold gluing that merges graphs from diverse domains into a unified smooth Riemannian manifold. Locally, geometry is estimated via a (k, M)-sparse perturbation and an Adaptive Orthogonal Frame (AOF) to form tangent-space bases and local metrics. Globally, pieces are “glued” by edge tangent translation (isometries) with holonomy (triangle) triviality and further curvature-based smoothing to promote a smooth global metric. Experiments over six domains show strong few-shot transfer and a geometric scaling law: adding more pre-training datasets yields smoother manifolds and better transfer.

**Strengths:**

1. Conceptual originality & unification: Frames multi-domain GFM pre-training/adaptation in a single geometric framework with local-to-global gluing, tying transferability to metric compatibility, holonomy, and Ricci-related smoothness—offering interpretable levers rather than only heuristics.
2. Experiments span diverse graph domains and demonstrate solid performance and scalability. The observed geometric scaling law is an insightful and interpretable empirical finding that supports the theoretical motivation.
3. The proposed framework is well structured, moving clearly from local metric learning (AOF) to global manifold construction (holonomy and curvature smoothing) and then to practical pre-training and adaptation.

**Weaknesses:**

1. Although the method is conceptually rich, the explanation is dense, with long mathematical expressions and limited intuition. The main framework diagram is visually cluttered and could better emphasize the data flow between modules. Simplifying the narrative and improving the visuals would greatly enhance clarity.
2. The paper lacks a clear analysis of computational cost and training scalability — given that manifold operations such as matrix logarithm and curvature regularization can be expensive, a runtime or memory comparison would help readers assess practicality for large-scale deployment.
3. The method relies on geometric assumptions such as triangle-based holonomy and curvature smoothness, but many real graphs are sparse or irregular, with very few closed loops. It remains unclear how well the model behaves when these geometric constraints cannot be fully satisfied. Adding experiments or visualizations on sparse or highly heterogeneous graphs would make the theory–practice connection more convincing.

**Questions:**

1. Triangle coverage & sparsity: How does performance degrade when the underlying graph (or inter-domain scaffold) has few triangles, so triangle holonomy regularization is weak or inapplicable? Do you back off to cycle-based approximations or add synthetic motifs? (Additionally, some theorems are based on the assumption that every edge belongs to at least one triangle, e.g., Theorem 4.8)
2. How sensitive is the learned manifold geometry to the hyperparameters of the AOF module (e.g., k and perturbation scale)?
3. Could you provide more intuition or examples to help readers understand how the “manifold gluing” process works in practice?
4. From which previously introduced equations or lemmas is **equation (30)** derived?
5. See weaknesses.

---

> ### Author Response · Authors · 2025-11-21
> **Authors' Responses**
>
> We do appreciate your constructive comments and provide our responses as follows.
>
> ## W1. The main framework diagram could better emphasize the data flow between modules.
>
> We refine the main framework diagram (Figure 2) in the revised manuscript, and emphasize the data flow:
> First, we learn the local geometry with the proposed AOF;
> Second, we derive the Riemannian prototypes with EMA;
> Third, we glue the local pieces to obtain a manifold skeleton;
> Finally, a unified (smooth) manifold is constructed with the proposed losses.
>
> ## W2. A clear analysis of computational cost and training scalability (given that manifold operations can be expensive), and a runtime or memory comparison.
>
>  First of all, we clarify an important point--the complexity of manifold operations.
>  Since we construct an **orthogonal** frame, the matrix form $\mathbf{G}$ of the metric tensor is a diagonal matrix, which can be viewed as an $M$-dimensional vector in computation and storage. Thus, the computation of the matrix logarithm is indeed the same as operating on vectors.
>
> Second, we analyze both computational complexity and memory complexity of the key modules in *GraphGlue* as follows.
>
> |Module|Computational complexity| Memory complexity |
> |------|---------------|------------------|
> |$(k, M)$-Sparse Perturbation|$\mathcal{O}(k_sMB)$|$ \mathcal{O}(BM(k_s+d)) $ |
> |Adaptive Orthogonal Frame|$\mathcal{O}(B(\lvert V \rvert +\lvert E \rvert + M^2)d)$ |$\mathcal{O}(BMd)$ |
> |Matrix form of metric tensor | $\mathcal{O}(BMd)$ | $\mathcal{O}(BM)$ |
> | $\mathcal{L}_{holo}$ and $\mathcal{L}_{curv}$| $\mathcal{O}(T_s M)$| $\mathcal{O}(T_s)$ |
> | Riemannian prototypes and $\mathcal{L}_{proto}$ |$\mathcal{O}(KBd + K(d+M))$ | $\mathcal{O}(K(d+M))$ |
> | Riemannian MoE| $\mathcal{O}(KBd)$| $\mathcal{O}(KB)$|
>
> > **Notation**:
> > - $B$: batch size, the number of graph samples in a batch.
> > - $|V|, |E|$: average nodes/edges per graph in batch.
> > - $d$: hidden dimension, setting to 512 commonly.
> > - $M$: number of nodes $\mathbb{P}$ in $(k,M)$-sparse perturbation, also the dimension of manifold, setting to 32 commonly.
> > - $k_s$: number of selected top-$k_s$ nodes in sparse perturbation.
> > - $T_s$: number of sampled triangle paths, **not** all triangles. For more effectiveness, we sample pairs of adjacent edges to approximate closed triangle paths.
>
> The total computational cost in pretraining phase is $\mathcal{O}(B (\lvert V \rvert + \lvert E \rvert + M^2 + K) d + T_s M)$,
> and the adaption (per graph) costs $\mathcal{O}((\lvert V \rvert + \lvert E \rvert) d + K(d+M) +  T_sM)$.
> That is, in *GraphGlue* scales linearly with respect to the graph size. In our experiment, we pretrain the model on large-scale datasets, e.g., ogbn-arxiv and Reddit.
>
> Third, we compare the proposed GraphGlue to other GFM in pretraining and adaptation phases regarding the total computational cost.
>
> | Model | Pretraining | Adaptation (per graph sample) | Notes |
> |-------|--------------|------------|-------|
> | PRODIGY| $\mathcal{O}(B \lvert V \rvert^2 d)$ | $\mathcal{O}( (\lvert V \rvert + \lvert E \rvert) d + \lvert V \rvert^2)$ | In-context learning requires full attention over prompt+query nodes |
> | GFT | $\mathcal{O}(B (\lvert V \rvert + \lvert E \rvert)  d + B  T   h)$ | $\mathcal{O}((\lvert V \rvert + \lvert E \rvert) d + T   h)$ | $T$: number of trees, $h$: tree height; tree construction adds overhead |
> | RAGraph  | $\mathcal{O}(B   (\lvert V \rvert + \lvert E \rvert)   d + B   \lvert E_r \rvert   d)$ | $\mathcal{O}((\lvert V \rvert + \lvert E \rvert) d + \lvert E_r \rvert   d)$ | $\lvert E_r \rvert$: retrieved edges from external library |
> | SAMGPT | $\mathcal{O}(B   (\lvert V \rvert + \lvert E \rvert)   d + B   k_s   d)$ | $\mathcal{O}((\lvert V \rvert + \lvert E \rvert) d + k_p   d)$ | $k_s$: number of structure tokens; $k_p$: prompt tokens |
> | GCOPE  | $\mathcal{O}(B   (\lvert V \rvert + \lvert E \rvert)   d + B   K_c   d)$ | $\mathcal{O}((\lvert V \rvert + \lvert E \rvert)d + K_c   d)$ | $K_c$: number of virtual coordinators |
> | MDGFM  | $\mathcal{O}(B   (\lvert V \rvert + \lvert E \rvert)   d + B   \lvert V \rvert^2)$ | $\mathcal{O}( (\lvert V \rvert + \lvert E \rvert) d + \lvert V \rvert^2)$ | Graph Structure Learning (GSL) involves dense adjacency refinement |
> | **GraphGlue** | $\mathcal{O}(B (\lvert V \rvert + \lvert E \rvert + M^2 + K) d + T_s M)$ | $\mathcal{O}((\lvert V \rvert + \lvert E \rvert) d + K(d+M) +  T_sM)$ | $M=32$, $T_s \ll \lvert E \rvert$.|
>
> Indeed, we achieve a lower computational cost to the previous methods. We will include the complexity analysis and comparison in the final version, as shown in the revised submission.

---

> > ### Author Response · Authors · 2025-11-21
> > **Authors' Responses**
> >
> > ## Continue of W2
> > Furthermore, we compare the training cost to GCOPE(KDD24) and MDGFM(ICML25) in six datasets.
> > $[1,2,3, ..., 6]$ denotes that we incrementally include ogbn-arxiv, computers, FB15k_237, Reddit, PROTEINS, HIV in the pretraining dataset.
> > Under the setting of $512$ batch size, $[10,10]$ neighbor sampler size, $d=512$. Results on GPU memory cost (GB) are as follows.
> >
> > | #datasets |1 | 2| 3| 4|5|6|
> > |----------|----|---|---|---|---|---|
> > |GCOPE|18.39|19.11|21.12|OOM|OOM|OOM|
> > |MDGFM|19.67|21.67|29.35|OOM|OOM|OOM|
> > |GraphGlue|12.35 |15.07 |15.73 | 16.87| 28.67| 29.3|
> >
> > ## Q1/W3. Triangle coverage & sparsity.
> >
> > **How does performance degrade when the underlying graph (or inter-domain scaffold) has few triangles? Add experiments or visualizations on sparse or highly heterogeneous graphs would make the theory–practice connection more convincing.**
> >
> > We demonstrate the model behavior on the sparse graphs with few triangles.
> > Here, we list the sparsity of experimental datasets and two extra datasets, where we adopt the sparsity metric of $S=\frac{2\lvert E \rvert}{\lvert V \rvert(\lvert V \rvert-1)}$ following [1].
> > Intuitively, S measures the proportion of actual edges present in a graph relative to the maximum possible number of edges in an undirected simple graph with the same number of nodes, quantifying the sparsity of graph datasets.
> >
> > [1] Newman, M. E. J. (2010). Networks: An Introduction. Oxford University Press.
> >
> > |Dataset | #nodes| #edges| #sparsity|
> > |--------|--------|------|----------|
> > |Arxiv|169343| 2315598 | 0.0001|
> > |Computers|1375|491722|0.0026|
> > |FB15k_237|14541| 423056|0.0020|
> > |Reddit|232965|114615892|0.0021|
> > |Roman-empire|22662|65854|0.0001|
> > |Amazon-ratings| 24492|186100|0.0003|
> > |Texas|183|574|0.0172|
> > |Wisconsin| 251|916|0.0146|
> >
> > We additionally provide the performance on four heterophilic graph datasets (Roman-empire, Amazon-ratings, Texas and Wisconsin) of evident sparsity.
> >
> > **Results of 1-shot setting (pretrained on ogbn-arxiv, Reddit, Computers, FB15k_237, PROTEINS, HIV)**
> >
> > | Model       | Amazon-Ratings     | Roman-empire       | Texas              | Wisconsin          |
> > |-------------|--------------------|--------------------|--------------------|--------------------|
> > | GCOPE       | 28.65 ± 5.82       | 11.44 ± 1.91       | 33.19 ± 6.62       | 31.22 ± 6.85       |
> > | MDGFM       | 29.53 ± 3.45       | 14.51 ± 2.08       | 34.63 ± 10.70      | 35.11 ± 10.53      |
> > | **GraphGlue** | **31.16 ± 3.56**   | **16.23 ± 3.00**   | **35.16 ± 20.43**  | **37.95 ± 10.91**  |
> >
> > **Results of 5-shot setting (pretrained on ogbn-arxiv, Reddit, Computers, FB15k_237, PROTEINS, HIV)**
> >
> > | Model       | Amazon-Ratings     | Roman-empire       | Texas              | Wisconsin          |
> > |-------------|--------------------|--------------------|--------------------|--------------------|
> > | GCOPE       | 30.06 ± 5.11       | 16.00 ± 1.29       | 36.31 ± 10.14      | 38.21 ± 2.96       |
> > | MDGFM       | 30.42 ± 3.80       | 17.15 ± 1.66       | 48.33 ± 6.36       | 47.46 ± 4.86       |
> > | **GraphGlue** | **32.17 ± 2.91**   | **18.50 ± 1.07**   | **50.88 ± 11.93**  | **49.71 ± 8.00**   |
> >
> > --The empirical results above demonstrate that the pre-trained GraphGlue achieves better performance on the sparse graphs compared to the recent GCOPE (KDD24) and MDGFM (ICML25).
> >
> > **Results of 1-shot setting (pretrained on ogbn-arxiv, Reddit, Computers, FB15k_237, PROTEINS, HIV, Amazon-ratings and Roman-empire)**
> >
> > | Model       | Amazon-Ratings     | Roman-empire       | Texas              | Wisconsin          |
> > |-------------|--------------------|--------------------|--------------------|--------------------|
> > | GCOPE       | 29.03 ± 4.17       | 13.14 ± 2.36       | 33.82 ± 7.39       | 30.08 ± 5.13       |
> > | MDGFM       | 27.01 ± 2.98       | 14.11 ± 2.14       | 36.02 ± 9.54       | 33.28 ± 8.76       |
> > | **GraphGlue** | **34.12 ± 2.57**   | **18.19 ± 2.51**   | **38.65 ± 13.88**  | **40.17 ± 10.09**  |
> >
> > **Results on 5-shot setting (pretrained on ogbn-arxiv, Reddit, Computers, FB15k_237, PROTEINS, HIV, Amazon-ratings and Roman-empire)**
> >
> > | Model       | Amazon-Ratings     | Roman-empire       | Texas              | Wisconsin          |
> > |-------------|--------------------|--------------------|--------------------|--------------------|
> > | GCOPE       | 32.83 ± 3.26       | 16.98 ± 1.40       | 41.33 ± 9.85       | 43.74 ± 3.19       |
> > | MDGFM       | 32.54 ± 3.75       | 16.77 ± 1.92       | 48.10 ± 7.26       | 46.62 ± 4.16       |
> > | **GraphGlue** | **36.26 ± 3.09**   | **20.67 ± 1.34**   | **52.60 ± 6.07**   | **51.49 ± 7.97**   |
> >
> >
> > --The empirical results above demonstrate that GraphGlue exhibits performance gain  to the recent GCOPE and MDGFM, when  sparse graphs are included during pre-training.
> >
> > We will include the performance on sparse graphs in the final version, as shown in the revised submission.

---

> > > ### Author Response · Authors · 2025-11-21
> > > **Authors' Responses**
> > >
> > > ## Continue of Q1/W3
> > > **Does Theorem 4.8 assume that every edge belongs to at least one triangle? Do you back off to cycle-based approximations or add synthetic motifs?**
> > >
> > > Every edge belonging to at least one triangle is not the assumption of Theorem 4.8.
> > > This theorem states that, if every edge belongs to at least one triangle, triangles are already sufficient to construct the coherent manifold described in this work.
> > > It means that there is no need for exploring any higher-order motifs, but *the triangle coverage is not the necessary condition of manifold gluing*.
> > >
> > >
> > > We clarify that GraphGlue does not need to add synthetic motifs.
> > > Since GraphGlue aims to approximate a smooth manifold, the closed triple paths (strict triangles) benefit the approximation process.
> > > As we consider that sample closed triangle paths may be impossible in large-scale graphs or tree-like graphs, the triangle holonomy regularization gives a computationally efficient way to approximate a ``perfect gluing."
> > > In practice, we only sample two adjacent edges to approximate strict triangles (at the end of Appendix D.1), and the computation of $\mathcal{L}_{curv}$ only relies on two adjacent edges.

---

> > > > ### Author Response · Authors · 2025-11-21
> > > > **Authors' Responses**
> > > >
> > > > ## Q2. How sensitive is the learned manifold geometry to the hyperparameters of the AOF module (e.g., k and perturbation scale)?
> > > >
> > > > For AOF, we investigate the hyperparameter sensitivity on the neighborhood size $k$ and the number of nodes $M$ in $(k, M)$-sparse perturbation.
> > > >
> > > > **Results of 1-shot setting**
> > > >
> > > > |   k | Arxiv | Computers  | Reddit  | FB15k_237  | PROTEINS  | HIV   |
> > > > |-----|------------------|-----------------------------|-------------------|----------------------|---------------------|----------------|
> > > > |   2 | 18.64 ± 3.10     | 49.80 ± 12.41               | 66.04 ± 1.91      | 31.63 ± 6.19         | 57.80 ± 3.05        | 53.07 ± 3.02   |
> > > > |   5 | 17.16 ± 3.08     | 43.76 ± 10.78               | 48.57 ± 6.76      | 21.10 ± 2.47         | 59.49 ± 2.62        | 52.04 ± 3.01   |
> > > > |  10 | 15.29 ± 2.69     | 46.21 ± 11.10               | 61.03 ± 2.89      | 45.51 ± 15.83        | 58.10 ± 3.41        | 54.42 ± 3.16   |
> > > > |  15 | 29.73±2.56       | 61.03 ± 7.13                | 68.42 ± 4.68      | 60.89±2.11           | 69.12± 4.19         | 58.53±8.20     |
> > > > |  30 | 20.23 ± 2.83     | 55.39 ± 10.15               | 49.57 ± 4.80      | 38.85 ± 19.62        | 54.36 ± 5.80        | 54.49 ± 3.33   |
> > > > |  60 | 18.20 ± 2.63     | 51.88 ± 10.22               | 75.76 ± 3.00      | 31.83 ± 16.41        | 58.24 ± 3.34        | 51.64 ± 3.44   |
> > > >
> > > > |   M | Arxiv  | Computers | Reddit   | FB15k_237 | PROTEINS  | HIV  |
> > > > |-----|------------------|--------------------|-------------------|--------------------|-------------------|----------------|
> > > > |   4 | 19.89 ± 3.79     | 42.66 ± 11.17      | 60.57 ± 4.48      | 45.16 ± 5.85      | 56.99 ± 4.64      | 52.58 ± 3.68   |
> > > > |   8 | 22.64 ± 2.51     | 46.24 ± 9.12       | 61.73 ± 5.02      | 54.80 ± 9.57       | 58.77 ± 1.92      | 52.32 ± 2.94   |
> > > > |  16 | 27.84 ± 1.47     | 56.92 ± 14.86      | 62.73 ± 1.97      | 57.09 ± 7.44       | 55.34 ± 5.68      | 54.18 ± 3.84   |
> > > > |  32 | 29.73 ± 2.56     | 61.03 ± 7.13       | 68.42 ± 4.68      | 60.89 ± 2.11       | 69.12 ± 4.19      | 58.53 ± 8.20   |
> > > >
> > > > **Results of 5-shot setting**
> > > >
> > > > |   k | Arxiv  | Computers  | Reddit  | FB15k_237   | PROTEINS  | HIV|
> > > > |-----|------------------|-----------------------------|-------------------|----------------------|---------------------|----------------|
> > > > |   2 | 29.49 ± 3.14     | 70.82 ± 3.14                | 80.46 ± 1.21      | 36.24 ± 5.11         | 59.42 ± 2.03        | 56.10 ± 2.76   |
> > > > |   5 | 23.95 ± 7.88     | 67.74 ± 4.02                | 67.41 ± 8.74      | 39.26 ± 14.50        | 60.45 ± 3.05        | 54.76 ± 2.33   |
> > > > |  10 | 25.86 ± 7.17     | 70.59 ± 18.03               | 80.49 ± 0.59      | 48.01 ± 10.53        | 60.86 ± 3.08        | 55.69 ± 4.23   |
> > > > |  15 | 39.98±1.67       | 74.15± 2.38                 | 84.89 ± 0.68      | 79.52±1.75           | 73.94 ± 2.38        | 62.18±2.50     |
> > > > |  30 | 33.00 ± 1.63     | 57.65 ± 29.35               | 64.57 ± 6.84      | 42.88 ± 16.13        | 62.09 ± 2.45        | 54.63 ± 3.01   |
> > > > |  60 | 32.59 ± 1.57     | 68.08 ± 1.66                | 85.24 ± 0.42      | 45.42 ± 8.09         | 60.40 ± 1.35        | 54.01 ± 3.69   |
> > > >
> > > > |   M | Arxiv   | Computers | Reddit | FB15k_237  | PROTEINS | HIV  |
> > > > |-----|------------------|--------------------|-------------------|--------------------|-------------------|----------------|
> > > > |   4 | 32.62 ± 3.40     | 64.66 ± 13.87      | 70.86 ± 2.92      | 60.63 ± 4.80      | 57.38 ± 4.76      | 56.27 ± 1.62   |
> > > > |   8 | 34.92 ± 1.50     | 72.25 ± 1.99       | 78.82 ± 1.79      | 63.18 ± 14.37      | 59.70 ± 1.92      | 54.64 ± 2.35   |
> > > > |  16 | 37.78 ± 5.79     | 74.22 ± 13.95      | 74.92 ± 2.41      | 70.11 ± 11.89      | 60.62 ± 3.66      | 57.55 ± 2.65   |
> > > > |  32 | 39.98 ± 1.67     | 74.15 ± 2.38       | 84.89 ± 0.68      | 79.52 ± 1.75       | 73.94 ± 2.38      | 62.18 ± 2.50   |
> > > >
> > > > We will include the analysis of hyperparameter sensitivity in the Appendix, as shown in the revised  submission.
> > > >
> > > > ## Q3. Could you provide more intuition or examples to help readers understand how the “manifold gluing” process works in practice?
> > > >
> > > > Imagine each graph as a small, flat patch (like a piece of paper). The goal is to glue these pieces into one smooth surface, whose process is described as follows.
> > > >
> > > > First, we construct local geometry on each patch using $(k,M)$-sparse perturbation--like drawing a coordinate grid.
> > > >
> > > > Then, when two graphs share similar structures, we "glue" their grids together along overlapping regions, ensuring no stretching or twisting (via the isometry of Def. 4.4 and holonomy of Eq. 5).
> > > >
> > > > Finally, we smooth the entire surface so that curvature changes gradually--forming a unified manifold where knowledge flows naturally across domains.
> > > >
> > > > Also, we visualize a toy example of aforementioned process in Figure 8 in the revised manuscript.

---

> > > > > ### Author Response · Authors · 2025-11-21
> > > > > **Authors' Responses**
> > > > >
> > > > > ## Q4. From which previously introduced equations or lemmas is equation (30) derived?
> > > > >
> > > > > Thanks so much for your careful question.
> > > > >
> > > > > Equation (30),
> > > > > $$
> > > > > w_{mp}(t) = \big( K_{MM}(t)[m,:] - a^\top(t) \big) P,
> > > > > $$
> > > > > is not derived from a single named lemma, but is a constructive step within the **proof of Theorem 4.3** (Appendix B.1), based on the following components:
> > > > >
> > > > > - The heat diffusion solution on the perturbed graph gives the perturbation-only embedding as $F_{\text{pert}}(t) = \exp(-t\hat{L}) [0; P]$ (Eq. 18).
> > > > > - The virtual node embedding is  $f_{mp}(t)=[K_{MM}(t)P][m]$ (Eq. 23).
> > > > > - The perturbation-induced global mean is $z_p(t) = \frac{1}{N} 1_N^\top S K_{SM}(t) P = a^\top(t) P$
> > > > >
> > > > > where $ a(t) = \frac{1}{N} K_{SM}^\top(t) \mathbf{1}_S $ (Eq. 29).
> > > > >
> > > > > Thus, the centered tangent vector is $w_{mp}(t) = f_{mp}(t) - z_p(t)$, which directly yields Eq. (30).
> > > > >
> > > > > **Clarification on the \(1/N\) factor**:  The $1/N$ in $a(t)$ is **correct and necessary**—it comes from the **global mean pooling** operation $z(t) = \frac{1}{N} \sum_{i=1}^N z_i(t) $ (Eq. 26). Therefore, **Eq. (30) does not contain an extra $1/N$**; the normalization is consistent with the definition of the graph-level representation used throughout the paper.

---

> > > > > > ### Author Response · Authors · 2025-11-25
> > > > > >
> > > > > > We would like to kindly follow up on our responses and would greatly appreciate if the reviewer could briefly confirm whether the above clarifications and empirical results have addressed your concerns. We also welcome any further discussion regarding our submission! Thank you once again for your time and thoughtful review—we sincerely value your feedback, and your engagement is truly important to us!

---

### Official Review · Reviewer_9yab · 2025-10-31

**Soundness:** 3
**Presentation:** 3
**Contribution:** 3
**Rating:** 6
**Confidence:** 4

**Summary:**

This paper presents a novel differential-geometry view of multi-domain graph pretraining and domain adaptation, which the authors call a neural manifold gluing framework. Locally the model learns tangent-space bases and local metrics via an Adaptive Orthogonal Frame (AOF) and a (k, M)-sparse perturbation; globally the framework “glues” local pieces using edge tangent translation and holonomy/curvature-based smoothing to induce a smooth Riemannian metric across domains. The method includes practical engineering (EMA-based prototyping and a Riemannian MoE) to scale to large multi-graph settings. Experiments on multiple datasets and leave-one-domain-out few-shot transfer show consistent improvements over baselines and an empirical “geometric scaling law.”

**Strengths:**

1. Strong theoretical foundation and unification. The paper provides a mathematically solid perspective, casting multi-domain pretraining and domain adaptation as a manifold-gluing problem; this is novel and helps unify several previously disparate ideas about metric compatibility, holonomy, and transferability. Theorem-level results and carefully defined operators (e.g., edge tangent translation) make the theoretical contribution convincing.

2. Practical scaling via an EMA strategy. The use of EMA prototypes and the design choices to make the framework operate on large graphs are valuable. These engineering choices materially improve the applicability of multi-domain graph learning to realistic, large-scale settings.

3. Comprehensive experiments and diverse datasets. The authors evaluate across a variety of datasets and tasks (leave-one-domain-out few-shot settings, ablations, scaling-law experiments). The empirical section is thorough and supports the main claims.

**Weaknesses:**

1. The idea of “manifold gluing” may depend on the number of domains and on how many pairwise/collective gluing operations are needed. The paper briefly remarks that a QR-based subroutine reduces complexity, but it lacks a clear, end-to-end complexity and empirical runtime/memory analysis showing how cost scales with the number of domains, graph size, and manifold dimension.
2. The paper states (line ~1532) “For pretraining, we extract the 2-hop ego-graph with 10 neighbors each hop for single graph datasets and adopt a 2-layer GCN...” Please clarify: how are those 10 neighbors chosen? Are they random uniform samples among neighbors, the top-10 by degree/score, or chosen by some importance sampling / structural heuristic? This will inform whether the pretraining pipeline’s effectiveness depends on a particular sampling heuristic.
3. Many of the node-classification datasets used appear to be homophilic. The manifold gluing assumptions may rely implicitly on local coherence of labels/structure; it is unclear whether the proposed framework adapts to heterophilic graphs. We suggest that the authors either include experiments on heterophilic benchmarks (or a small toy study) or provide a discussion of expected behavior if heterophilic graphs are included in pretraining.

**Questions:**

See weaknesses.

---

> ### Author Response · Authors · 2025-11-21
> **Authors' Responses**
>
> We do appreciate your constructive comments and provide our responses as follows.
>
> ## W1.  A clear, end-to-end complexity and empirical runtime/memory analysis showing how cost scales with the number of domains, graph size, and manifold dimension.
>
> We provide both time and memory complexity of the key modules in GraphGlue as follows.
>
> |Module|Time complexity| Memory complexity |
> |------|---------------|------------------|
> |$(k, M)$-Sparse Perturbation|$\mathcal{O}(k_sMB)$|$ \mathcal{O}(BM(k_s+d)) $ |
> |Adaptive Orthogonal Frame|$\mathcal{O}(B(\lvert V \rvert +\lvert E \rvert + M^2)d)$ |$\mathcal{O}(BMd)$ |
> |Matrix form of metric tensor | $\mathcal{O}(BMd)$ | $\mathcal{O}(BM)$ |
> | $\mathcal{L}_{holo}$ and $\mathcal{L}_{curv}$| $\mathcal{O}(T_s M)$| $\mathcal{O}(T_s)$ |
> | Riemannian prototypes and $\mathcal{L}_{proto}$ |$\mathcal{O}(KBd + K(d+M))$ | $\mathcal{O}(K(d+M))$ |
> | Riemannian MoE| $\mathcal{O}(KBd)$| $\mathcal{O}(KB)$|
>
> > **Notation**:
> > - $B$: batch size, the number of graph samples in a batch.
> > - $|V|, |E|$: average nodes/edges per graph in batch.
> > - $d$: hidden dimension, setting to 512 commonly.
> > - $M$: number of nodes $\mathbb{P}$ in $(k,M)$-sparse perturbation, also the dimension of manifold, setting to 32 commonly.
> > - $k_s$: number of selected top-$k_s$ nodes in sparse perturbation.
> > - $T_s$: number of sampled triangle paths, **not** all triangles. For more effectiveness, we sample pairs of adjacent edges to approximate closed triangle paths.
>
> Thus, the total computational cost in pretraining phase is $\mathcal{O}(B (\lvert V \rvert + \lvert E \rvert + M^2 + K) d + T_s M)$,
> and the adaption (per graph) costs $\mathcal{O}((\lvert V \rvert + \lvert E \rvert) d + K(d+M) +  T_sM)$.
>
> Note that the proposed GraphGlue scales linearly with respect to the graph size, showing its applicability to large-scale graphs.
>
> Furthermore, we compare the memory cost to GCOPE and MDGFM in six datasets.
> $[1,2,3, ..., 6]$ denotes that we incrementally include ogbn-arxiv, computers, FB15k_237, Reddit, PROTEINS, HIV in the pretraining dataset.
> Under the setting of $512$ batch size, $[10,10]$ neighbor sampler size, $d=512$. Results on GPU memory cost (GB) are as follows.
>
> | #datasets |1 | 2| 3| 4|5|6|
> |----------|----|---|---|---|---|---|
> |GCOPE|18.39|19.11|21.12|OOM|OOM|OOM|
> |MDGFM|19.67|21.67|29.35|OOM|OOM|OOM|
> |GraphGlue|12.35 |15.07 |15.73 | 16.87| 28.67| 29.3|
>
> ## W2.  Clarification on Line 1532. How are those 10 neighbors chosen? Are they random uniform samples among neighbors, the top-10 by degree/score, or chosen by some importance sampling/structural heuristic?
>
> **On the neighbor sampling strategy.** In our pre-training pipeline, the 2-hop ego-graphs are extracted using **pytorch_geometric’s NeighborSampler**. By default, this sampler performs uniform random sampling with replacement for the neighbors at each hop.
>
> Specifically, for a given center node v , we sample 10 neighbors from its 1-hop neighborhood N(v) uniformly at random (with replacement), and then for each of those 10 nodes, we sample 10 neighbors from their respective neighborhoods to form the 2-hop structure. This approach does not rely on degree-based selection, importance sampling, or other structural heuristics. We chose this standard random walk-like strategy because it is widely used in GNN literature (e.g., GraphSAGE) for its efficiency and its ability to provide diverse local subgraph samples during batched training. Our framework’s effectiveness is therefore tied to this established sampling paradigm, rather than a novel heuristic.

---

> > ### Author Response · Authors · 2025-11-21
> > **Authors' Responses**
> >
> > ## W3. Include experiments on heterophilic benchmarks (or a small toy study), or provide a discussion of expected behavior if heterophilic graphs are included in pretraining.
> >
> > We demonstrate the performance of GraphGlue on several benchmarking heterophilic graphs (Amazon-ratings, Roman-empire, Texas and Wisconsin)
> >
> > **Results of 1-shot setting (pretrained on ogbn-arxiv, Reddit, Computers, FB15k_237, PROTEINS, HIV)**
> >
> > | Model       | Amazon-Ratings     | Roman-empire       | Texas              | Wisconsin          |
> > |-------------|--------------------|--------------------|--------------------|--------------------|
> > | GCOPE       | 28.65 ± 5.82       | 11.44 ± 1.91       | 33.19 ± 6.62       | 31.22 ± 6.85       |
> > | MDGFM       | 29.53 ± 3.45       | 14.51 ± 2.08       | 34.63 ± 10.70      | 35.11 ± 10.53      |
> > | **GraphGlue** | **31.16 ± 3.56**   | **16.23 ± 3.00**   | **35.16 ± 20.43**  | **37.95 ± 10.91**  |
> >
> > **Results of 5-shot setting (pretrained on ogbn-arxiv, Reddit, Computers, FB15k_237, PROTEINS, HIV)**
> >
> > | Model       | Amazon-Ratings     | Roman-empire       | Texas              | Wisconsin          |
> > |-------------|--------------------|--------------------|--------------------|--------------------|
> > | GCOPE       | 30.06 ± 5.11       | 16.00 ± 1.29       | 36.31 ± 10.14      | 38.21 ± 2.96       |
> > | MDGFM       | 30.42 ± 3.80       | 17.15 ± 1.66       | 48.33 ± 6.36       | 47.46 ± 4.86       |
> > | **GraphGlue** | **32.17 ± 2.91**   | **18.50 ± 1.07**   | **50.88 ± 11.93**  | **49.71 ± 8.00**   |
> >
> > --The empirical results above demonstrate that the pre-trained GraphGlue achieves better performance on the heterophilic graphs compared to the recent GCOPE (KDD24) and MDGFM (ICML25).
> >
> > **Results of 1-shot setting (pretrained on ogbn-arxiv, Reddit, Computers, FB15k_237, PROTEINS, HIV, Amazon-ratings and Roman-empire)**
> >
> > | Model       | Amazon-Ratings     | Roman-empire       | Texas              | Wisconsin          |
> > |-------------|--------------------|--------------------|--------------------|--------------------|
> > | GCOPE       | 29.03 ± 4.17       | 13.14 ± 2.36       | 33.82 ± 7.39       | 30.08 ± 5.13       |
> > | MDGFM       | 27.01 ± 2.98       | 14.11 ± 2.14       | 36.02 ± 9.54       | 33.28 ± 8.76       |
> > | **GraphGlue** | **34.12 ± 2.57**   | **18.19 ± 2.51**   | **38.65 ± 13.88**  | **40.17 ± 10.09**  |
> >
> > **Results on 5-shot setting (pretrained on ogbn-arxiv, Reddit, Computers, FB15k_237, PROTEINS, HIV, Amazon-ratings and Roman-empire)**
> >
> > | Model       | Amazon-Ratings     | Roman-empire       | Texas              | Wisconsin          |
> > |-------------|--------------------|--------------------|--------------------|--------------------|
> > | GCOPE       | 32.83 ± 3.26       | 16.98 ± 1.40       | 41.33 ± 9.85       | 43.74 ± 3.19       |
> > | MDGFM       | 32.54 ± 3.75       | 16.77 ± 1.92       | 48.10 ± 7.26       | 46.62 ± 4.16       |
> > | **GraphGlue** | **36.26 ± 3.09**   | **20.67 ± 1.34**   | **52.60 ± 6.07**   | **51.49 ± 7.97**   |
> >
> >
> > --The empirical results above demonstrate that GraphGlue exhibits performance gain to the recent GCOPE and MDGFM, when heterophilic graphs are included during pre-training.
> >
> > That is, *GraphGlue* shows superiority in the cases of testing on heterophilic graphs and pre-training with heterophilic graphs. It demonstrates its robustness to heterophily.
> >
> > Theoretically, GraphGlue works with the geometric consistency rather than label homophily. It aligns local tangent spaces and preserves intrinsic graph topology by isometric transport and enforcing holonomy triviality. Since these geometric cues are independent of label similarity between neighbors, the framework is inherently capable of handling heterophilic graphs. The EMA prototyping further ensures that domains with distinct characteristics (including heterophilic ones) are mapped to separate regions on the unified manifold, preventing interference.

---

> > > ### Author Response · Authors · 2025-11-25
> > >
> > > We would like to kindly follow up on our responses and would greatly appreciate if the reviewer could briefly confirm whether the above clarifications and empirical results have addressed your concerns. We also welcome any further discussion regarding our submission! Thank you once again for your time and thoughtful review—we sincerely value your feedback, and your engagement is truly important to us!

---

> > > ### Comment · Reviewer_9yab · 2025-11-26
> > >
> > > Thank you for the detailed and constructive rebuttal. The added experiments on both homophilic and heterophilic graphs, as well as the clearer complexity and memory analysis, substantially strengthen the submission and address my earlier concerns. These results improve my confidence in the method’s generality and scalability. The paper presents an important and timely contribution to multi-domain graph foundation models, offering a unifying geometric framework from a Riemannian manifold perspective. In light of the authors' responses, I am raising my overall score.

---

> > > > ### Author Response · Authors · 2025-11-26
> > > >
> > > > We thanks for your appreciation on our theoretical contributions, and we are so pleased that our responses have addressed your concerns. Thanks once again for your kind support!

---

### Official Review · Reviewer_RdxQ · 2025-11-01

**Soundness:** 4
**Presentation:** 3
**Contribution:** 4
**Rating:** 8
**Confidence:** 4

**Summary:**

This paper introduces GraphGlue, a novel framework for multi-domain graph pre-training that leverages principles from differential geometry to address the long-standing challenge of knowledge transfer across diverse graph domains. By conceptualizing graphs as pieces of a unified Riemannian manifold, the authors provide a systematic approach to ensure that these domains are seamlessly “glued” together, preserving both local and global geometric consistency. The framework employs an Adaptive Orthogonal Frame (AOF) to model local graph structure, then uses holonomy-based regularization to smoothly align these pieces into a coherent whole.

One of the key contributions of this work is the Geometric Transfer Metric (GTM), which quantifies the transfer difficulty between pre-trained models and target domains. The experiments show that GraphGlue achieves superior transferability and scalability across multiple graph domains, outperforming existing methods. The paper also introduces a new empirical insight: geometric scaling law, which suggests that increasing the number of domains during pre-training leads to a smoother manifold and better generalization.

**Strengths:**

1. The paper introduces a novel approach to multi-domain graph pre-training by treating the graphs as local pieces of a larger, unified Riemannian manifold. This fresh perspective allows for a more systematic understanding of knowledge transfer across domains, which is a critical issue in graph foundation models.
2. The concept of “neural manifold gluing” is well-formulated, using differential geometry to tie together multiple domains. The method employs an Adaptive Orthogonal Frame (AOF) to model local geometry and uses holonomy and curvature constraints to ensure smooth global alignment. This provides both theoretical depth and practical utility.
3. The experiments cover a range of graph domains and demonstrate that the proposed method outperforms previous models in terms of transferability. The geometric scaling law showing that increasing the number of domains improves transferability is both a novel and insightful contribution to the field.
4. Convincing theoretical analyses.

**Weaknesses:**

1. The method heavily relies on triangle-based holonomy for graph gluing. However, in sparse graphs or those with few cycles, this assumption may not hold, limiting the approach's applicability. Further analysis of the method’s behavior on sparse or acyclic graphs is needed.
2. The paper introduces the AOF for local geometry estimation, but does not provide sufficient analysis on how sensitive the results are to the choice of hyperparameters like perturbation strength and neighborhood size.
3. Some parts of the paper writing needs to be polished.

**Questions:**

1. See weaknesses.
2. Could you provide a clear logical chain explaining the connections and roles of the numerous theorems and lemmas? **This is my main concern**, and I hope the authors can primarily address this question within a limited number of characters.

---

> ### Author Response · Authors · 2025-11-21
> **Authors' Response**
>
> We do appreciate your constructive comments and provide our responses as follows.
>
> ## W1. Explanation on triangle-based holonomy and further analysis of the method’s behavior on sparse or acyclic graphs.
>
> First, we clarify that *GraphGlue* does not heavily rely on triangles.
> We summarize the role of triangles in the graph in Theorem 4.8 on Triangle Triviality,
> which states that triangles, rather than higher-order motifs, are already sufficient to construct the coherent manifold described in this work.
> It means that there is no need for exploring any higher-order motifs, but *the triangle coverage is not the necessary condition of manifold gluing*.
>
> Since GraphGlue aims to approximate a smooth manifold, the closed triple paths (strict triangles) benefit the approximation process.
> As we consider that sample closed triangle paths may be impossible in large-scale graphs or tree-like graphs, the triangle holonomy regularization gives a computationally efficient way to approximate a ``perfect gluing".
> In practice, we only sample two adjacent edges to approximate strict triangles (at the end of Appendix D.1), and the computation of $\mathcal{L}_{curv}$ only relies on two adjacent edges.
>
> Second, we demonstrate the model behavior on the sparse graphs with few triangles.
> Here, we list the sparsity of experimental datasets and two extra datasets, where we adopt the sparsity metric of  $S=\frac{2\lvert E \rvert}{\lvert V \rvert(\lvert V \rvert-1)}$ following [1].
> Intuitively, S measures the proportion of actual edges present in a graph relative to the maximum possible number of edges in an undirected simple graph with the same number of nodes, quantifying the sparsity of graph datasets.
>
> [1] Newman, M. E. J. (2010). Networks: An Introduction. Oxford University Press.
>
> |Dataset | #nodes| #edges| #sparsity|
> |--------|--------|------|----------|
> |Arxiv|169343| 2315598 | 0.0001|
> |Computers|1375|491722|0.0026|
> |FB15k_237|14541| 423056|0.0020|
> |Reddit|232965|114615892|0.0021|
> |Roman-empire|22662|65854|0.0001|
> |Amazon-ratings| 24492|186100|0.0003|
> |Texas|183|574|0.0172|
> |Wisconsin| 251|916|0.0146|
>
> We additionally provide the performance on four heterophilic graph datasets (Roman-empire, Amazon-ratings, Texas and Wisconsin) of evident sparsity.
>
> **Results of 1-shot setting (pretrained on ogbn-arxiv, Reddit, Computers, FB15k_237, PROTEINS, HIV)**
>
> | Model       | Amazon-Ratings     | Roman-empire       | Texas              | Wisconsin          |
> |-------------|--------------------|--------------------|--------------------|--------------------|
> | GCOPE       | 28.65 ± 5.82       | 11.44 ± 1.91       | 33.19 ± 6.62       | 31.22 ± 6.85       |
> | MDGFM       | 29.53 ± 3.45       | 14.51 ± 2.08       | 34.63 ± 10.70      | 35.11 ± 10.53      |
> | **GraphGlue** | **31.16 ± 3.56**   | **16.23 ± 3.00**   | **35.16 ± 20.43**  | **37.95 ± 10.91**  |
>
> **Results of 5-shot setting (pretrained on ogbn-arxiv, Reddit, Computers, FB15k_237, PROTEINS, HIV)**
>
> | Model       | Amazon-Ratings     | Roman-empire       | Texas              | Wisconsin          |
> |-------------|--------------------|--------------------|--------------------|--------------------|
> | GCOPE       | 30.06 ± 5.11       | 16.00 ± 1.29       | 36.31 ± 10.14      | 38.21 ± 2.96       |
> | MDGFM       | 30.42 ± 3.80       | 17.15 ± 1.66       | 48.33 ± 6.36       | 47.46 ± 4.86       |
> | **GraphGlue** | **32.17 ± 2.91**   | **18.50 ± 1.07**   | **50.88 ± 11.93**  | **49.71 ± 8.00**   |
>
> --The empirical results above demonstrate that the pre-trained GraphGlue achieves better performance on the sparse graphs compared to the recent GCOPE (KDD24) and MDGFM (ICML25).
>
> **Results of 1-shot setting (pretrained on ogbn-arxiv, Reddit, Computers, FB15k_237, PROTEINS, HIV, Amazon-ratings and Roman-empire)**
>
> | Model       | Amazon-Ratings     | Roman-empire       | Texas              | Wisconsin          |
> |-------------|--------------------|--------------------|--------------------|--------------------|
> | GCOPE       | 29.03 ± 4.17       | 13.14 ± 2.36       | 33.82 ± 7.39       | 30.08 ± 5.13       |
> | MDGFM       | 27.01 ± 2.98       | 14.11 ± 2.14       | 36.02 ± 9.54       | 33.28 ± 8.76       |
> | **GraphGlue** | **34.12 ± 2.57**   | **18.19 ± 2.51**   | **38.65 ± 13.88**  | **40.17 ± 10.09**  |

---

> ### Author Response · Authors · 2025-11-21
> **Authors' Responses**
>
> ## Continue of W1
> **Results on 5-shot setting (pretrained on ogbn-arxiv, Reddit, Computers, FB15k_237, PROTEINS, HIV, Amazon-ratings and Roman-empire)**
>
> | Model       | Amazon-Ratings     | Roman-empire       | Texas              | Wisconsin          |
> |-------------|--------------------|--------------------|--------------------|--------------------|
> | GCOPE       | 32.83 ± 3.26       | 16.98 ± 1.40       | 41.33 ± 9.85       | 43.74 ± 3.19       |
> | MDGFM       | 32.54 ± 3.75       | 16.77 ± 1.92       | 48.10 ± 7.26       | 46.62 ± 4.16       |
> | **GraphGlue** | **36.26 ± 3.09**   | **20.67 ± 1.34**   | **52.60 ± 6.07**   | **51.49 ± 7.97**   |
>
>
> The empirical results above demonstrate that GraphGlue exhibits performance gain  to the recent GCOPE and MDGFM, when  sparse graphs are included during pre-training.
>
> ## W2. On AOF. How sensitive the results are to the choice of hyperparameters like perturbation strength and neighborhood size.
>
> For AOF, we investigate the hyperparameter sensitivity on the neighborhood size $k$ and the number of nodes $M$ in $(k, M)$-sparse perturbation.
>
> **Results of 1-shot setting**
>
> |   k | Arxiv | Computers  | Reddit  | FB15k_237  | PROTEINS  | HIV   |
> |-----|------------------|-----------------------------|-------------------|----------------------|---------------------|----------------|
> |   2 | 18.64 ± 3.10     | 49.80 ± 12.41               | 66.04 ± 1.91      | 31.63 ± 6.19         | 57.80 ± 3.05        | 53.07 ± 3.02   |
> |   5 | 17.16 ± 3.08     | 43.76 ± 10.78               | 48.57 ± 6.76      | 21.10 ± 2.47         | 59.49 ± 2.62        | 52.04 ± 3.01   |
> |  10 | 15.29 ± 2.69     | 46.21 ± 11.10               | 61.03 ± 2.89      | 45.51 ± 15.83        | 58.10 ± 3.41        | 54.42 ± 3.16   |
> |  15 | 29.73±2.56       | 61.03 ± 7.13                | 68.42 ± 4.68      | 60.89±2.11           | 69.12± 4.19         | 58.53±8.20     |
> |  30 | 20.23 ± 2.83     | 55.39 ± 10.15               | 49.57 ± 4.80      | 38.85 ± 19.62        | 54.36 ± 5.80        | 54.49 ± 3.33   |
> |  60 | 18.20 ± 2.63     | 51.88 ± 10.22               | 75.76 ± 3.00      | 31.83 ± 16.41        | 58.24 ± 3.34        | 51.64 ± 3.44   |
>
> |   M | Arxiv  | Computers | Reddit   | FB15k_237 | PROTEINS  | HIV  |
> |-----|------------------|--------------------|-------------------|--------------------|-------------------|----------------|
> |   4 | 19.89 ± 3.79     | 42.66 ± 11.17      | 60.57 ± 4.48      | 45.16 ± 5.85      | 56.99 ± 4.64      | 52.58 ± 3.68   |
> |   8 | 22.64 ± 2.51     | 46.24 ± 9.12       | 61.73 ± 5.02      | 54.80 ± 9.57       | 58.77 ± 1.92      | 52.32 ± 2.94   |
> |  16 | 27.84 ± 1.47     | 56.92 ± 14.86      | 62.73 ± 1.97      | 57.09 ± 7.44       | 55.34 ± 5.68      | 54.18 ± 3.84   |
> |  32 | 29.73 ± 2.56     | 61.03 ± 7.13       | 68.42 ± 4.68      | 60.89 ± 2.11       | 69.12 ± 4.19      | 58.53 ± 8.20   |
>
> **Results of 5-shot setting**
>
> |   k | Arxiv  | Computers  | Reddit  | FB15k_237   | PROTEINS  | HIV|
> |-----|------------------|-----------------------------|-------------------|----------------------|---------------------|----------------|
> |   2 | 29.49 ± 3.14     | 70.82 ± 3.14                | 80.46 ± 1.21      | 36.24 ± 5.11         | 59.42 ± 2.03        | 56.10 ± 2.76   |
> |   5 | 23.95 ± 7.88     | 67.74 ± 4.02                | 67.41 ± 8.74      | 39.26 ± 14.50        | 60.45 ± 3.05        | 54.76 ± 2.33   |
> |  10 | 25.86 ± 7.17     | 70.59 ± 18.03               | 80.49 ± 0.59      | 48.01 ± 10.53        | 60.86 ± 3.08        | 55.69 ± 4.23   |
> |  15 | 39.98±1.67       | 74.15± 2.38                 | 84.89 ± 0.68      | 79.52±1.75           | 73.94 ± 2.38        | 62.18±2.50     |
> |  30 | 33.00 ± 1.63     | 57.65 ± 29.35               | 64.57 ± 6.84      | 42.88 ± 16.13        | 62.09 ± 2.45        | 54.63 ± 3.01   |
> |  60 | 32.59 ± 1.57     | 68.08 ± 1.66                | 85.24 ± 0.42      | 45.42 ± 8.09         | 60.40 ± 1.35        | 54.01 ± 3.69   |
>
> |   M | Arxiv   | Computers | Reddit | FB15k_237  | PROTEINS | HIV  |
> |-----|------------------|--------------------|-------------------|--------------------|-------------------|----------------|
> |   4 | 32.62 ± 3.40     | 64.66 ± 13.87      | 70.86 ± 2.92      | 60.63 ± 4.80      | 57.38 ± 4.76      | 56.27 ± 1.62   |
> |   8 | 34.92 ± 1.50     | 72.25 ± 1.99       | 78.82 ± 1.79      | 63.18 ± 14.37      | 59.70 ± 1.92      | 54.64 ± 2.35   |
> |  16 | 37.78 ± 5.79     | 74.22 ± 13.95      | 74.92 ± 2.41      | 70.11 ± 11.89      | 60.62 ± 3.66      | 57.55 ± 2.65   |
> |  32 | 39.98 ± 1.67     | 74.15 ± 2.38       | 84.89 ± 0.68      | 79.52 ± 1.75       | 73.94 ± 2.38      | 62.18 ± 2.50   |

---

> ### Author Response · Authors · 2025-11-21
> **Authors' Responses**
>
> ## W3. A logical chain explaining the connections and roles of the numerous theorems and lemmas.
>
> In a nutshell, we first learn the local geometry and then "glue'' the local pieces together.
> The theorems progressively "glue'' the discrete graph domains into a coherent, smooth, and transferable continuous structure, giving the metric to quantify the transferability in the meanwhile.
> Hence, the main logic chain is shown as follows,
>
> Learning local geometry (4.3) $\rightarrow$ Gluing local pieces (4.5–4.8) $\rightarrow$ Smoothing the global structure (4.9–4.11) $\rightarrow$ Forming a unified manifold (4.11).
>
> Each component of  *GraphGlue* is established with strict theoretical gaurantees. Now, we specify the logic chain for each definition/theorem.
>
> - Definition 4.1 and 4.2: we give the construction of tangent space, which is the key to constructing a differential manifold. The orthogonality of the adaptive frame makes the computation of the metric or translation friendly, since the metric is diagonal and can be treated as an $M$-dimensional vector.
> - Theorem 4.3: Bounding tangent vector length ensures stable local geometry via $(k,M)$-perturbation.
> - Definition 4.4 and Theorem 4.5 & 4.6: Edge-wise isometric transport ($\mathbf{P}^{(i,j)}$) guarantees metric compatibility $\rightarrow$ existence of a global continuous metric, is a key tool of manifold gluing.
> - Theorem 4.8: Trivial holonomy on triangles $\rightarrow$ trivial holonomy on all cycles $\rightarrow$ path-independent gluing (no holonomy conflict).
> - Theorem 4.9: Volume ratio ($\text{det} \mathbf{G}_i/\text{det} \mathbf{G}_j$) approximates Ricci curvature sign ⇒ enables curvature-aware smoothing.
> - Definition 4.10 + Theorem 4.11: Enforcing log-det $\infty$-smoothness + trivial holonomy ⇒ $C^2$-smooth global Riemannian manifold.

---

> > ### Comment · Reviewer_RdxQ · 2025-11-21
> > **A theoretically solid piece of work.**
> >
> > I no longer have any concerns regarding this paper. Given that **most current GFM research focuses primarily on empirical results while the theoretical foundations remain relatively weak**, I find the authors’ attempt to develop a theoretical framework for GFMs from a manifold perspective to be **highly meaningful and valuable**. Based on the above considerations, I strongly recommend this paper and intend to raise my score. I hope the authors will further verify the lemmas and equations presented in the manuscript.

---

> > > ### Author Response · Authors · 2025-11-23
> > >
> > > Dear Reviewer,
> > >
> > > We deeply appreciate your thoughtful suggestions and kind support! Thanks so much!
> > >
> > >
> > >
> > > Sincerely,
> > >
> > > Authors of GraphGlue

---

### Official Review · Reviewer_w9Kq · 2025-11-07

**Soundness:** 2
**Presentation:** 3
**Contribution:** 3
**Rating:** 4
**Confidence:** 2

**Summary:**

This paper focuses on how knowledge is theoretically integrated and transferred during multi-domain graph pre-training, which is a fundamental and underexplored problem in the field of graph foundation models (GFMs). Authors propose a novel perspective from differential geometry, with the core idea of gluing diverse graph datasets into a unified Riemannian manifold. Moreover, authors empirically validate GRAPHGLUE’s geometric scaling law and show that larger quantities of datasets could improve the transferability of models by producing a smoother manifold.

**Strengths:**

- This paper is good in originality and introduces a principled and powerful theoretical framework from differential geometry. This Neural Manifold Gluing concept provides a new, systematic, and theoretically sound language to model knowledge integration in graphs, which is a conceptual advance for the GFM field.
- The experiments provide strong support for the theory. The Geometric Scaling Law experiment shows that 1-shot accuracy improves and transfer loss decreases as more datasets are added, which is a powerful validation of the smoother manifold hypothesis. Furthermore, the case study in Figure 5 shows that GRAPHGLUE benefits from adding semantically distinct domains in mitigating negative transfer is a strong practical result.

**Weaknesses:**

- The proposed theory assumes that all source and target domains can be glued into a single smooth manifold. However, it is unclear how the model would perform if a new domain is fundamentally geometrically incompatible. For example, what if a new domain possesses a markedly different intrinsic dimensionality? Therefore, further discussion on the limitations and potential failure cases would be valuable for practical use.
- Several key design choices in the GRAPHGLUE framework are not ablated. In particular, what is the impact of the EMA prototyping and the L_proto loss? How critical is the Riemannian MoE during adaptation compared to a simpler prompting scheme? I expect a more comprehensive ablation that provides deeper comprehension regarding the impact of individual components within the full framework.
- The proposed framework involves several non-trivial operations. For instance, (k, M)-sparse perturbation, maintaining an Adaptive Orthogonal Frame (AOF), and calculating gluing losses (L_holo, L_curv). I am curious about the computational cost and memory complexity of the overall pre-training and adaptation process.

**Questions:**

Please see the weakness.

---

> ### Author Response · Authors · 2025-11-21
> **Authors' Responses**
>
> We do appreciate your constructive comments and provide our responses as follows.
>
> ## W1.  Further discussion on how the model would perform if a new domain is fundamentally geometrically incompatible.
>
> In our case study, we examine the performance in an incompatible new domain. Specifically, we incrementally expand a Reddit-only pre-training with the distinct PROTEINS and HIV dataset.
> (Note that Reddit is a social network, while PROTEINS and HIV are biological datasets. They exhibit evident differences in both semantics and geometry).
>
> Here, we present the results on a diversity of datasets. It demonstrates that the inclusion of diverse geometry during pre-training robustly improve the performance on the testing datasets.
> Also, taking PROTEINS dataset for instance, Reddit-only pre-trained GraphGlue (51.34) achieve better results than GCN (50.1 as shown in Table 1 of our submission) on an evidently incompatible PROTEINS.
>
> **Results of 1-shot setting on Reddit dataset**
> | Pretrain Datasets         | Reddit | PROTEINS | HIV  |Computers|
> |---------------------------|--------|----------|------|---------|
> | Reddit                    | 61.53  | 54.21    | 51.34|55.62 |
> | Reddit + PROTEINS         | 62.71  | 65.88    | 53.17|56.83|
> | Reddit + PROTEINS + HIV   | 64.33  | 67.05    | 57.92| 58.11|
>
> Indeed, GraphGlue naturally has the ability to handle geometrically incompatible domains:
> - In the pretraining phase, manifold unification is to merge different local pieces as a coherent whole (w.r.t. to a certain Riemannian metric), and it does not mean a manifold with only one single connected component. If a batch of graphs are intrinsically different from one another, similar graphs are densely connected or overlapped, while semantically distinct graphs are far away from each other. Note that our theory is applicable to a unified manifold with disconnected components.
> - In the adaptive phase, if the target dataset is intrinsically different from the source datasets, GTM, in Eq. (12), will quantify the degree of geometric disagreement. As we optimize $\mathcal{L}_{glue}$ (Eq. (11)), the adapter will not force target dataset gluing to the pretrained manifold but will keep the original geometric structure to form a local branch that sparsely connects to the pretrained manifold.
>
> Hence, GraphGlue is capable of merging geometrically incompatible graphs into the pretrained manifold as a coherent whole.
>
> ## W2.  Comprehensive ablation on the proposed components of  EMA, prototype loss and Riemannian MoE.
>
> We conduct a further ablation study to verify the effectiveness of the aforementioned components. Specifically, we introduce 3 variants of GraphGlue, described as follows:
>
> ''w/o EMA '' means that we replace EMA with the common average of a batch of embeddings.
>
> ''w/o $\mathcal{L}_{proto}$'' means  pretraining without prototype loss.
>
> ''w/o Riemannian MoE'' means that during adaption, Riemannian MoE module is replaced by a typical prompting scheme.
>
>  Both results on 1-shot setting and 5-shot setting demonstrate the effectiveness of the proposed components.
>
> **Performances of 1-shot Setting on 6 Datasets**：
>
> | Method             | Arxiv           | Computers | Reddit          | FB15k_237       | PROTEINS        | HIV             |
> |--------------------|-----------------|------------------|-----------------|-----------------|-----------------|-----------------|
> | w/o EMA            | 15.46 ± 1.41    | 30.84 ± 9.50     | 7.43 ± 2.37     | 35.90 ± 17.40   | 58.48 ± 2.56    | 52.62 ± 2.41    |
> | w/o L_proto        | 9.57 ± 4.56     | 31.24 ± 10.36    | 37.90 ± 9.34    | 43.59 ± 8.13    | 58.49 ± 2.60    | 54.10 ± 2.76    |
> | w/o Riemannian MoE | 16.90 ± 3.54    | 44.98 ± 12.65    | 51.02 ± 8.53    | 42.23 ± 10.83   | 59.52 ± 2.07    | 51.51 ± 2.50    |
> | **GraphGlue**      | **29.73 ± 2.56**| **61.03 ± 7.13** | **68.42 ± 4.68**| **60.89 ± 2.11**| **69.12 ± 4.19**| **58.53 ± 8.20**|
>
> **Performances of 5-shot Setting on 6 Datasets**：
> | Method             | Arxiv           | Computers | Reddit          | FB15k_237       | PROTEINS        | HIV             |
> |--------------------|-----------------|------------------|-----------------|-----------------|-----------------|-----------------|
> | w/o EMA            | 16.08 ± 2.13    | 34.90 ± 7.77     | 11.53 ± 2.26    | 40.21 ± 12.07   | 59.85 ± 4.53    | 53.87 ± 2.43    |
> | w/o L_proto        | 15.26 ± 9.91    | 36.63 ± 11.84    | 46.13 ± 14.14   | 64.56 ± 16.42   | 61.64 ± 6.28    | 55.50 ± 2.70    |
> | w/o Riemannian MoE | 28.04 ± 8.75    | 46.34 ± 25.40    | 72.68 ± 5.52    | 56.07 ± 9.32    | 57.49 ± 3.31    | 53.22 ± 3.13    |
> | **GraphGlue**      | **39.98 ± 1.67**| **74.15 ± 2.38** | **84.89 ± 0.68**| **79.52 ± 1.75**| **73.94 ± 2.38**| **62.18 ± 2.50**|
>
> We will include the ablation study on these components in the final version, as shown in the revised submission.

---

> > ### Author Response · Authors · 2025-11-21
> > **Authors' Responses**
> >
> > ## W3. The computational cost and memory complexity of the overall pre-training and adaptation process.
> >
> > First, we analyze both computational complexity and memory complexity of the key modules in *GraphGlue* as follows.
> >
> > |Module|Computational complexity| Memory complexity |
> > |------|---------------|------------------|
> > |$(k, M)$-Sparse Perturbation|$\mathcal{O}(k_sMB)$|$ \mathcal{O}(BM(k_s+d)) $ |
> > |Adaptive Orthogonal Frame|$\mathcal{O}(B(\lvert V \rvert +\lvert E \rvert + M^2)d)$ |$\mathcal{O}(BMd)$ |
> > |Matrix form of metric tensor | $\mathcal{O}(BMd)$ | $\mathcal{O}(BM)$ |
> > | $\mathcal{L}_{holo}$ and $\mathcal{L}_{curv}$| $\mathcal{O}(T_s M)$| $\mathcal{O}(T_s)$ |
> > | Riemannian prototypes and $\mathcal{L}_{proto}$ |$\mathcal{O}(KBd + K(d+M))$ | $\mathcal{O}(K(d+M))$ |
> > | Riemannian MoE| $\mathcal{O}(KBd)$| $\mathcal{O}(KB)$|
> >
> > > **Notation**:
> > > - $B$: batch size, the number of graph samples in a batch.
> > > - $|V|, |E|$: average nodes/edges per graph in batch.
> > > - $d$: hidden dimension, setting to 512 commonly.
> > > - $M$: number of nodes $\mathbb{P}$ in $(k,M)$-sparse perturbation, also the dimension of manifold, setting to 32 commonly.
> > > - $k_s$: number of selected top-$k_s$ nodes in sparse perturbation.
> > > - $T_s$: number of sampled triangle paths, **not** all triangles. For more effectiveness, we sample pairs of adjacent edges to approximate closed triangle paths.
> >
> > The total computational cost in the pretraining phase is $\mathcal{O}(B (\lvert V \rvert + \lvert E \rvert + M^2 + K) d + T_s M)$,
> > and the adaption (per graph) costs $\mathcal{O}((\lvert V \rvert + \lvert E \rvert) d + K(d+M) +  T_sM)$.
> > That is, in *GraphGlue* scales linearly with respect to the graph size. In our experiment, we pretrain the model on large-scale datasets, e.g., ogbn-arxiv and Reddit.
> >
> > Second, we compare the proposed GraphGlue to other GFM in the pretraining and adaptation phases regarding the total computational cost.
> >
> > | Model | Pretraining | Adaptation (per graph sample) | Notes |
> > |-------|--------------|------------|-------|
> > | PRODIGY| $\mathcal{O}(B \lvert V \rvert^2 d)$ | $\mathcal{O}( (\lvert V \rvert + \lvert E \rvert) d + \lvert V \rvert^2)$ | In-context learning requires full attention over prompt+query nodes |
> > | GFT | $\mathcal{O}(B (\lvert V \rvert + \lvert E \rvert)  d + B  T   h)$ | $\mathcal{O}((\lvert V \rvert + \lvert E \rvert) d + T   h)$ | $T$: number of trees, $h$: tree height; tree construction adds overhead |
> > | RAGraph  | $\mathcal{O}(B   (\lvert V \rvert + \lvert E \rvert)   d + B   \lvert E_r \rvert   d)$ | $\mathcal{O}((\lvert V \rvert + \lvert E \rvert) d + \lvert E_r \rvert   d)$ | $\lvert E_r \rvert$: retrieved edges from external library |
> > | SAMGPT | $\mathcal{O}(B   (\lvert V \rvert + \lvert E \rvert)   d + B   k_s   d)$ | $\mathcal{O}((\lvert V \rvert + \lvert E \rvert) d + k_p   d)$ | $k_s$: number of structure tokens; $k_p$: prompt tokens |
> > | GCOPE  | $\mathcal{O}(B   (\lvert V \rvert + \lvert E \rvert)   d + B   K_c   d)$ | $\mathcal{O}((\lvert V \rvert + \lvert E \rvert)d + K_c   d)$ | $K_c$: number of virtual coordinators |
> > | MDGFM  | $\mathcal{O}(B   (\lvert V \rvert + \lvert E \rvert)   d + B   \lvert V \rvert^2)$ | $\mathcal{O}( (\lvert V \rvert + \lvert E \rvert) d + \lvert V \rvert^2)$ | Graph Structure Learning (GSL) involves dense adjacency refinement |
> > | **GraphGlue** | $\mathcal{O}(B (\lvert V \rvert + \lvert E \rvert + M^2 + K) d + T_s M)$ | $\mathcal{O}((\lvert V \rvert + \lvert E \rvert) d + K(d+M) +  T_sM)$ | $M=32$, $T_s \ll \lvert E \rvert$.|
> >
> > Indeed, we achieve a lower computational cost than the previous methods. We will include the complexity analysis and comparison in the final version, as shown in the revised submission.

---

> > > ### Author Response · Authors · 2025-11-25
> > >
> > > We would like to kindly follow up on our responses and would greatly appreciate if the reviewer could briefly confirm whether the above clarifications and empirical results have addressed your concerns. We also welcome any further discussion regarding our submission! Thank you once again for your time and thoughtful review—we sincerely value your feedback, and your engagement is truly important to us!

---

> > > ### Comment · Reviewer_w9Kq · 2025-11-26
> > >
> > > I appreciate the authors' rebuttal in discussing the incorporation of new domains, comprehensive ablations, and complexity analysis. I no longer have any further concerns and therefore have decided to raise my score.

---

> > > > ### Author Response · Authors · 2025-11-26
> > > >
> > > > We are so happy that our responses have addressed your concerns. Thank again for your feedback, and we deeply appreciate your support!

---

### Author Response · Authors · 2025-11-29
**Summary of Author-Reviewer Discussion**

We thank again for the constructive comments and the reviewers’ engagement!

- For reviewer w9kq, we have addressed the concerns of the performance on geometrically incompatible graphs/diverse geometries, comprehensive ablation study, and computational cost/memory complexity. **Reviewer w9kq has raised the Rating from 4 to 6 on Nov. 26**.
- For reviewer RdxQ, we have addressed the concerns of the role of triangle-based holonomy, the performance on sparse/acyclic graphs, hyperparameter sensitivity of AOF, and the logic chain of the numerous theorems. **Reviewer RdxQ strongly recommends our theoretically solid piece of work since most current GFM research focuses primarily on empirical results while the theoretical foundations remain relatively weak, and raises the rating from 8 to 10 on Nov. 21**.
- For reviewer 9yab, we have addressed the concerns of complexity and empirical runtime/memory analysis, implementation details of neighbor choice, and the performance on heterophilic benchmarks with heterophilic graph pre-training. **Reviewer 9yab has raised the Rating from 6 to 8 on Nov. 26**.
- For the concerns of reviewer 5PYX, we provide improved main framework diagram, computational cost and training scalability, runtime/memory comparison, triangle coverage & sparsity, hyperparameter sensitivity of AOF, and intuition of graph gluing, derivation of Eq. 30. Most concerns are overlapped with other reviewers, and have been addressed.

---

### Meta-Review · Area_Chair_QPqi · 2026-01-04

**Summary:**

This paper proposes a principled and theoretically grounded framework for multi-domain graph pre-training via neural manifold gluing, offering a coherent geometric perspective on transferability in graph foundation models. Reviewers initially raised concerns regarding theoretical clarity, scalability, complexity analysis, behavior on sparse/heterophilic graphs, and empirical completeness. Through a detailed and well-structured rebuttal, the authors provided substantial clarifications, additional theoretical explanations, comprehensive ablations, expanded experimental results, and thorough computational and memory complexity analyses. The discussion significantly strengthened the paper, resolving most concerns and reinforcing both the technical soundness and practical relevance of the proposed approach.

**Reviewer Concerns:**

Most of the reviewers’ concerns have been adequately addressed in the rebuttal.
In particular, the authors clarified the logical chain and roles of the theoretical results, provided deeper intuition for manifold gluing and triangle-based holonomy, and justified key modeling assumptions. Concerns about performance on geometrically incompatible, sparse, and heterophilic graphs were addressed with new experiments and analyses. Questions regarding hyperparameter sensitivity, neighbor sampling, and implementation details were answered with concrete explanations and empirical evidence. Importantly, the authors added clear end-to-end computational and memory complexity analyses and comparisons with related methods, alleviating scalability concerns.

No major unresolved issues remain that would substantially undermine the paper’s contributions.

**Reviewer Scores:**

Based on the author–reviewer discussion, multiple reviewers have already increased their scores following the rebuttal. Reviewer w9kq raised the score from 4 to 6 after the authors addressed concerns on geometrically incompatible graphs, ablation studies, and computational complexity. Reviewer RdxQ increased the score from 8 to 10, explicitly endorsing the work as a theoretically solid contribution. Reviewer 9yab raised the score from 6 to 8 after clarifications on complexity analysis, implementation details, and heterophilic graph performance. Reviewer 5PYX’s concerns largely overlapped with others and were similarly addressed. Overall, the discussion indicates a clear positive shift in reviewer sentiment, with scores converging toward acceptance.

---

### Decision · Program_Chairs · 2026-01-26

Accept (Oral)